

# Large structures simulation for landscape evolution models

Julien Coatléven[1] and Benoit Chauveau[1]

[1]IFP Énergies nouvelles, 1 et 4 avenue de Bois-Préau, 92852 Rueil-Malmaison, France

**Correspondence:** Julien Coatléven (julien.coatleven@ifpen.fr)

**Abstract.** Because of the chaotic behavior of the coupling between water flow and sediment erosion and transport, without any special treatment the practical results of landscape evolution models (LEM) are likely to be dominated by numerical errors. This paper describes two areas of improvement that we believe are necessary for the successful simulation of landscape evolution models. The first one concerns the expression of the water flux that was initially rebuilt in Coatléven (2020) in a mathematically consistent way for the cell-to-cell multiple flow direction algorithms, thanks to a reinterpretation as a well chosen discretization of the Gauckler-Manning-Strickler continuous equation. Building on those results, we introduce here a general framework allowing to derive consistent expressions of the water flux for the most commonly used multiple/single flow direction (MFD/SFD) water flow routines, including node-to-node versions. If having a consistent water flux is crucial to avoid any mesh size dependence in a LEM and controlling the consistency error, the expected non-linear self amplification mechanisms of the water and sediment coupling can still lead to simulations blurred by numerical errors. Those numerical instabilities being highly reminiscent of turbulence induced instabilities in computational fluid dynamics (CFD), in the second part of our paper we present a "large structure simulation" (LSS) approach for LEM, mimicking the large-eddy simulations (LES) used for turbulent CFD. The LSS allows to control numerical errors while preserving the major physical based geomorphic patterns.

## 1 Introduction

Since the pioneering work of Gilbert in the XIX century (Gilbert (1880)), the meaning of the term "landscape evolution model" (LEM) has evolved until reaching in the late XX century its modern definition. It is now considered has a numerical application of a mathematical system that seeks to simulate a part of the physical processes controlling the landscape dynamic. The capability of LEMs to provide an integrated simulation in which several processes are addressed make them particularly relevant to tackle a large variety of contexts, such as the imprint of the upper mantle on the history of a sedimentary basin over thousands to millions of years (e.g., Braun et al. (2013); Granjeon (2014)), the control exerted by glacial dynamics (Egholm et al. (2012); Hergarten (2021)), or the impact of climate change at variable time-scales on drainage basins and soils (e.g., Tucker and Slingerland (1997); Coulthard et al. (2000); Braun et al. (2015); Srivastava et al. (2022)). The success of those numerical approaches depends on their ability to correctly handle the positive non-linear feedback between the water flow and



the sediment erosion and deposition in a decent computational time. This non-linear coupling between water and sediments is indeed expected to potentially induce complex water flow networks even on initially small topographic variations, allowing in return the emergence of complex geomorphic landforms. However, in the absence of reference analytic solutions, it is hard to decipher if the obtained landform results from physical processes or from the self-amplification of initially small numerical errors. The objective of the present paper is to propose a general approach to considerably diminish the risk of producing

inconsistent numerical results. The first ingredient is obviously to make sure to remove any anomalous consistency error in the numerical schemes, while the second one consists in introducing a method to control the evolution of the numerical errors. We believe that any LEMs developer will take advantage in following the recommendations resulting from this two topics, whatever the space and time scales considered.

There is a wide variety of mathematical models describing the flow of water, depending on the prominent space and time scales

of the considered problem. The most complete model is the Navier-Stokes model which allows for very precise but prohibitively costly simulations. The shallow-water approximation is sometimes used to solve rivers system (e.g. Audusse et al. (2004)) or to simulate glacial dynamics Egholm et al. (2011). Despite a reduced computational cost compared to the Navier-Stokes model, this model has not been often explicitly deployed in LEMs. Probably one of the reasons is that computationally efficient water flow routing algorithms have been developed during the last decades. Those algorithms are built assuming that the water flow

follows the direction of steepest descent (e.g. O'Callaghan and Mark (1984); Freeman (1989); Fairfield and Leymarie (1991); Quinn et al. (1991); Holmgren (1994); Quinn et al. (1995); Qin et al. (2007)), and are able to simulate relatively complex water flow networks despite this inherent simplicity. Multiple flow direction (MFD) and single flow direction (SFD) algorithms are among the most known water-flow routing families implemented in reference LEMs such as in SIBERIA(Willgoose et al. (1989, 1991); Willgoose (2005)), CAESAR-Lisflood (Bates et al. (2010); Coulthard et al. (2013)), FastScape (Braun and

Willett (2013)), eSCAPE (Salles (2018)), CIDRE (Carretier et al. (2020)), EROS (Davy et al. (2017)) or BadLand (Salles et al. (2017)), or in stratigraphic models such as DionisosFlow (Granjeon (2014)). This list being not exhaustive, the reader is referred to Tucker and Hancock (2010); Van der Beek (2013); Valters (2016); Armitage (2019); Nones (2020) for a complete review. The main differences between them is in their representation of the discretized domain (cell-to-cell or node-to-node interaction following the terminology in Armitage (2019)) and by the empirical choice made to distribute water among the

mesh elements.

The empirical foundations of the MFD/SFD water flow routing and their lack of mathematical framework make them very difficult to validate. A first behavior known since a long time is not very encouraging: the water flow distribution $\mathcal{Q}_w$ is mesh dependent. This is probably the most documented problem of the LEM community since more than twenty years (e.g. Schoorl et al. (2000); Pelletier (2010); Armitage (2019)) and one that still disturbs current models. Smart solutions have been published

to minimize this effect without making it completely vanish (Perron et al. (2009); Pelletier (2010)), while alternatively it was proposed to redefine drainage area at the continuous level Gallant and Hutchinson (2011); Bonetti et al. (2018) to allow a consistent discretization of $\mathcal{Q}_w$. From a mathematical point of view, it has been quite recently understood by Coatléven (2020) that the water discharge obtained from the cell-to-cell MFD/SFD corresponds in fact to a non-consistent approximation of the





water flux of a Gauckler-Manning-Strickler model, a simplification of the shallow water model. This allowed Coatléven (2020)
to correct the MFD/SFD to obtain a consistent and thus mesh independent approximation for $\mathcal{Q}_w$, in the usual numerical
analysis sense of convergence when the mesh size go to zero. The first purpose of this paper is consequently to recall this
result and then to explicitly show how all the classical MFD/SFD algorithms, even the node-to-node versions, can be in fact
interpreted and thus corrected in the same way slightly generalizing the results of Coatléven (2020) and finally solving the grid
dependency issue.

Because of the self-amplification mechanisms at the core of the equation system and that are assumed to play a major role in
valleys formation and their spacing (see Scheingross et al. (2020); Bonetti et al. (2020); Perron et al. (2009); Hooshyar and
Porporato (2021b)), solving the consistency issue in these LEMs is thus absolutely necessary to avoid creating anomalous
numerical errors but yet but not sufficient to guarantee that results are not dominated by numerical errors. Unless some special
numerical treatment is added, the expected self-amplification processes can indeed also amplify legitimate numerical round-
off or solver errors up to the point that they potentially completely blur the numerical solution. This "butterfly effect" is very
reminiscent of the turbulence issue arising in the field of computational fluid dynamics (CFD). This observation is not new
and was studied in details for instance in Bonetti et al. (2020); Hooshyar et al. (2020). The modern solution found by the
CFD community to achieve reproducible and meaningful simulations is to replace direct numerical simulation (DNS) of the
Navier-Stokes equations by large eddy simulation (LES, Berselli et al. (2005)). The objective of LES is to obtain a correct
approximation of local spatial averages of turbulent flows, recovering the correct dynamics only for the organized structures
of the flow (the eddies) which are larger than some target length scale $\alpha$. Thus, LES chooses to abandon the idea of resolving
all the scales involved in the true physical processes, as there is no hope to use a mesh fine enough to correctly resolve the
smallest scales. In practice this is done by filtering the solution to distinguish between the behavior of the flow above and
below $\alpha$, and obtaining local averages that are smoother and as mesh independent as possible. To our knowledge, the first
attempt at using an LES approach for simulating landscape evolution albeit without explicitly mentioning LES is Perron et al.
(2009), where a Laplacian smoothing (equivalent to a mesh related box filter in the LES terminology) was applied on the
topography. More recently Hooshyar and Porporato (2021a); Porporato (2022) resorted to a mono-directional domain size
related box filter to obtain robust results on channelization statistics and scaling signatures: in other words they substitute the
elevation and the specific drainage area by their mean values in the axial direction of their rectangular simulated domain. In
their conclusion they suggest that the use of more general LES approaches seems a viable avenue for more complex landscape
evolution simulations. In line with this observation, we also believe that the success of the attempts of Perron et al. (2009);
Hooshyar and Porporato (2021a); Porporato (2022), the numerous analogies between the instabilities arising in landscape
evolution models and turbulence reported in Smith and Bretherton (1972); Scheingross et al. (2020); Bonetti et al. (2020);
Hooshyar and Porporato (2021b) as well as the numerical experiments strongly advocate for the use of some LES technology
to overcome the numerical issues arising in the non-linear coupling of sediment evolution and water flow. Those are the reasons
why we have considered deriving a "large structures simulation" (LSS) approach for landscape evolution. We will see that the
numerical results of LSS seem remarkably reproducible. Notice that contrary to Hooshyar and Porporato (2021a); Porporato
(2022) and more in line to what is done in the CFD community, we will fix a length scale that will correspond to the size of





the smallest structures we want to resolve in the problem, quite independently of the domain size. We also consider a more advanced differential filter, namely the Leray-$\alpha$ filter (Cheskidov et al. (2005); Guermond et al. (2003)) that is not related to any specific geometric configuration. In this sense, our work can be considered as a generalization of Hooshyar and Porporato (2021a); Porporato (2022). We will show that the results obtained from LSS are actually free of the non-physical heterogeneity that appeared spontaneously from numerical errors. Notice that those numerical artifacts were often misleading as they induced more "realistic looking" solutions than the correct ones obtained by the LSS, in the sense that the obtained topography was

more complex. Starting from the LSS approach, it then becomes relevant to inject physically controlled heterogeneity in order to bring out the complexity in the results.

The two items addressed in this paper are complementary and should benefit to every LEMs. The paper will be organized accordingly. In the first part we introduce the notations to describe a sedimentary system. The second part tackles the issue of the consistency in the MFD/SFD algorithms. After recalling the results of Coatléven (2020) on the classical cell-to-cell MFD

algorithms, we detail the extension to the most classical MFD/SFD node-to-node algorithms of the literature. The resulting relations between the Gauckler-Manning-Strickler model and the water flow routing are summarized in a synthesis table, which should help the developer to implement a consistent version of the flow routing used in its landscape model. The third section of the paper starts by illustrating on an easy to analyze synthetic sedimentary system the issue related to the self-reinforcement between the water flow and the sediment dynamics. We then introduce the LES inspired filtering strategy and apply this "large

structure simulation" (LSS) approach on the illustrative test case as well as on a more complex model.

## 2    Model and notation

Following Smith and Bretherton (1972), we assume that a sedimentary system can be idealized through the following assumptions: (H1) the basin topography can be represented as a mathematical surface, (H2) the principle of the conservation of mass applies to this surface, (H3) the sediment flux at any point of the surface is a function of the local slope and the local discharge of water. In other words, using an Eulerian approach (H1) implies that we consider a fixed geographical region over the time

period $]0,T[$ mathematically modeled by means of a domain $\Omega \in \mathbb{R}^2$, a function $b : \Omega \times ]0,T[ \longrightarrow \mathbb{R}$ describing the basement i.e. the lower part of the basin in the $z$ direction, and a function $h_s : \Omega \times ]0,T[ \longrightarrow \mathbb{R}$ describing the thickness of the sediments (see Fig. 1). Thus, our basin $\mathcal{B} : ]0,T[ \longrightarrow \mathbb{R}^3$ can be described for almost every (a.e.) $t \in ]0,T[$ by:

$$\mathcal{B}(t) = \left\{ (x,y,z) \in \mathbb{R}^3 \ \middle| \ (x,y) \in \Omega \text{ and } b(x,y,t) \leq z \leq b(x,y,t) + h_s(x,y,t) \right\}. \tag{1}$$


The evolution of the basement $b$ is mostly governed by two processes: tectonics (both thermal and structural) and flexure. In the present paper we assume that the evolution of $b$ is a data, and we focus on computing the evolution of the function $h_s$. For



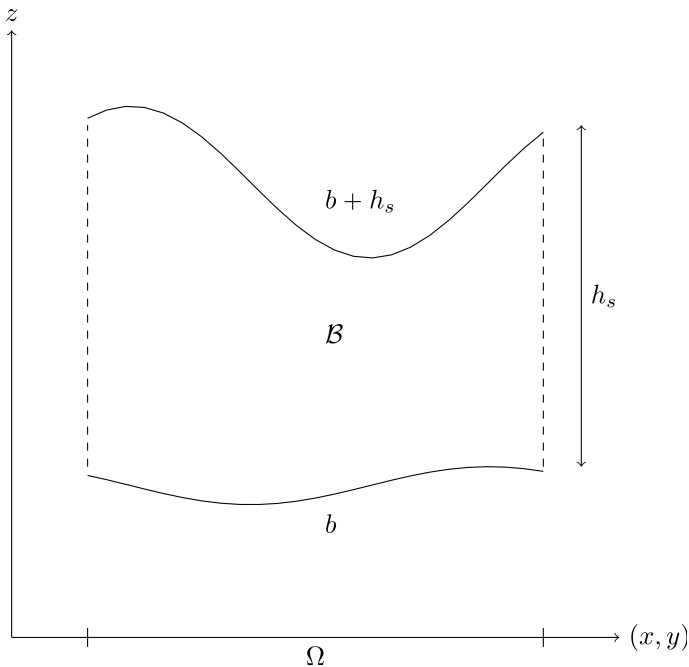

**Figure 1.** Representation of the two main surfaces considered in a landscape evolution model in the $(z,\Omega)$ parameter space, where $z$ is the elevation and $\Omega$ the spatial domain. the basement b surface represents the bottom part of the simulated block, on which sediments are deposited. The topographic surface is $b + h_s$ where $h_s$ is the sediment thickness. The simulated sedimentary content is denoted $\mathcal{B}$.

the sake of clarity, we give the expression of the mass conservation (H2) equations, neglecting porosity for simplicity:

$$
\left|
\begin{array}{ll}
\dfrac{\partial h_s}{\partial t} + div\left(\boldsymbol{J}_s\right) = S_s & \text{in } \Omega \times ]t_0, T[, \\[2ex]
-\boldsymbol{J}_s \cdot \boldsymbol{n} = B_s & \text{on } \partial\Omega_{\mathcal{N}} \times ]t_0, T[, \\[2ex]
h_s = 0 & \text{on } \partial\Omega_{\mathcal{D}} \times ]t_0, T[, \\[2ex]
h_s(t = t_0) = h_{s,0} & \text{in } \Omega,
\end{array}
\right.
\tag{2}
$$

where $S_s$ and $B_s$ are sediment source terms (coming from an in-situ sediment production, from soil erosion, or from sediment supplies defined in the domain boundaries) and $\boldsymbol{J}_s$ is the sediment flux. The domain boundary $\partial\Omega$ is divided between $\partial\Omega_{\mathcal{N}}$ where Neumann boundary conditions are imposed and $\partial\Omega_{\mathcal{D}}$ where we enforce homogeneous Dirichlet boundary conditions. Choosing a model corresponds to choosing a specific expression for the sediment flux and the source terms. A common feature of almost all LEMs is that the sediment flux model $\boldsymbol{J}_s$ and/or the source term $S_s$ depend non-linearly on the local discharge of

water $\mathcal{Q}_w$, very often through a power law like $\mathcal{Q}_w^{r_s}||\nabla(h_s + b)||^{p_s+1}$. Self-amplification mechanisms are known to appear as soon as $r_s > 1$ (Smith and Bretherton (1972)).





Let us precise that in the following the xy coordinates will be expressed in kilometers (km), while sediment height $h_s$ and basement $b$ will be expressed in meters (m).

## 3 From mesh dependent multiple flow direction algorithms to consistent approximations of continuous
Gauckler-Manning-Strickler models

Landscape evolution models usually defines the "local discharge of water" $\mathcal{Q}_w$ directly from the so-called drainage or catchment area $CA$ (also sometimes called contributing area). Roughly speaking $CA$ corresponds at a given outlet to the measure of the horizontal projection of the surface area from which the water contributing to this outlet is coming from (Maxwell (1870); Leopold et al. (1964); Bonetti et al. (2018)). Despite being a very intuitive notion, it has evaded for a long time a precise math-
ematical definition. Classical multiple flow direction (MFD) algorithms are intended to provide a practical way at computing $CA$ for a mesh cell. As is well documented (Desmet and Govers (1996); Pelletier (2010, 2013); Porporato (2022)) the discrete catchment areas obtained from those algorithms strongly depends on the cell size, geometry and orientation with respect to the flow. Several attempts can be found in the literature to reduce this mesh dependency, defining $\mathcal{Q}_w = (CA/w)$ where $w$ is a normalization factor equal for instance to the cartesian cell side length or diagonal length (cf Desmet and Govers (1996)) or
to an estimate of the width of the flow (Pelletier (2010)) defining the so-called specific or unit catchment area (SCA/UCA). A more modern mathematical definition of the specific catchment area $a$ was proposed in Gallant and Hutchinson (2011); Bonetti et al. (2018), consisting in solving an abstract uniform flow equation:

$$
\left|
\begin{array}{ll}
-div\left(a\dfrac{\nabla(h_s+b)}{||\nabla(h_s+b)||}\right)=1 & \text{in } \Omega, \\[4mm]
-a\dfrac{\nabla(h_s+b)}{||\nabla(h_s+b)||}\cdot\boldsymbol{n}=0 & \text{on } \partial\Omega_{in},
\end{array}
\right.
\tag{3}
$$

where $\partial\Omega_{in}=\{\boldsymbol{x}\in\partial\Omega \mid \nabla(h_s+b)\cdot\boldsymbol{n}>0\}$ is the part of the boundary that is in going and $\boldsymbol{n}$ denotes the outward normal to
$\Omega$. Setting $\mathcal{Q}_w=a$, this allows to reduce the mesh dependency to the usual consistency errors of numerical schemes.

A mathematical model encompassing the most classical cell-to-cell MFD algorithms was described in Coatléven (2020) where it is established that those algorithms coincide with a solver for a well chosen discretization of the following stationary water mass conservation with Gauckler-Manning-Strickler flux model for surface runoff:

$$
\left|
\begin{array}{ll}
-div\left(k_m h_w \eta_w(h_w)s_{ref}^{-p_w}||\nabla(h_s+b)||^{p_w}\nabla(h_s+b)\right)=S_w & \text{in } \Omega, \\[2mm]
-k_m h_w \eta_w(h_w)s_{ref}^{-p_w}||\nabla(h_s+b)||^{p_w}\nabla(h_s+b)\cdot n=0 & \text{on } \partial\Omega_{in},
\end{array}
\right.
\tag{4}
$$

where $h_w$ is the water height, $s_{ref}=$ 1 m.km$^{-1}$ the reference slope, $p_w$ a model parameter and $\eta_w$ the water mobility function. For simplicity we assume here that the mobility function has no dimension, and that the source $S_w$ is given in m$^3$.s$^{-1}$km$^{-2}$ such that its integral over a 2d area measured in km$^2$ coincides with a discharge in m$^3$.s$^{-1}$. The coefficient $k_m$ can be though of as the Strickler coefficient or the inverse of the Gauckler-Manning coefficient up to a change of unit (strictly speaking, this identification is trully valid for channels and if the mobility function $\eta_w$ is equal to a dimensionless hydraulic radius). For





this choice of source, $k_m$ has the unit m.s$^{-1}$ of a speed. Steady state analysis (Graf and Altinakar (2000); Birnir et al. (2001)) for channels suggests to use values $\eta_w(h_w) = (h_w/h_{ref})^{1/2}$ and $p_w = -1/2$, while the classical Gauckler-Manning-Strickler formula would coincide with $\eta_w(h_w) = (R_h(h_w)/h_{ref})^{2/3}$ with $R_h(h_w)$ the hydraulic radius and again $p_w = -1/2$. However the hypothesis underlying those results is tailored to channel flows, which is probably not valid over the wide range of flow configurations occurring at the large time and space scales of landscape evolution models. For this reason, we prefer to not fix precise values for $\eta_w$ and $p_w$ and think of them as modeling parameters that can be tuned for each considered problem. The analysis of Coatléven (2020) allows to give a general definition of the catchment area: for an open set $\mathcal{O} \subset \mathbb{R}^2$, the catchment area for the outlet of $\mathcal{O}$ is defined by:

$$CA(\mathcal{O}) = \widetilde{q}_{\mathcal{O}}^{ex} = \int_{\partial\mathcal{O}} h_w \eta_w(h_w) \left( -k_m s_{ref}^{-p_w} ||\nabla(h_s+b)||^{p_w} \nabla(h_s+b) \cdot \boldsymbol{n} \right)^+ ,$$

where $h_w$ is the solution of (4) with $S_w = 1$ and $v^+ = \max(0,v)$. We see that $CA(\mathcal{O})$ strongly depends on the geometry of $\mathcal{O}$ and its orientation with respect to the flow. In particular if we take for $\mathcal{O}$ a cell of the mesh we understand why the MFD algorithms produce mesh dependent catchment areas. In line with the attempts of Desmet and Govers (1996) or Pelletier (2010) to define a unit catchment area (UCA) by rescaling the CA, it is clear that the correct scaling would be to set $w$ to the length of the portion of $\partial\mathcal{O}$ such that $\left( -k_m s_{ref}^{-p_w} ||\nabla(h_s+b)||^{p_w} \nabla(h_s+b) \cdot \boldsymbol{n} \right)^+ > 0$, which depending on the orientation of the flow will sometimes match the choices of Desmet and Govers (1996) or Pelletier (2010, 2013) explaining their partial success. Thus a corrected definition of the unit catchment int the spirit of Desmet and Govers (1996); Pelletier (2010, 2013) area would be to use:

$$UCA(\mathcal{O}) = \frac{1}{\displaystyle\int_{\partial\mathcal{O}} \chi_{-k_m s_{ref}^{-p_w} ||\nabla(h_s+b)||^{p_w} \nabla(h_s+b) \cdot \boldsymbol{n} > 0}} \int_{\partial\mathcal{O}} h_w \eta_w(h_w) \left( -k_m s_{ref}^{-p_w} ||\nabla(h_s+b)||^{p_w} \nabla(h_s+b) \cdot \boldsymbol{n} \right)^+ ,$$

where $\chi$ is the indicator function (i.e. the function with value 1 when the condition is satisfied and 0 otherwise). This scales as an approximation of the continuous water flux magnitude $q_w = |k_m h_w \eta_w(h_w)| s_{ref}^{-p_w} ||\nabla(h_s+b)||^{p_w+1}$ (in m$^3$s$^{-1}$km$^{-1}$) but is not equal to it, and still retains some dependency in the geometry of $\mathcal{O}$ and its orientation with respect to the flow. In this context it is more natural to use directly $\mathcal{Q}_w = q_w$ such that the erosion in (2) depends on the local water flux magnitude and the slope. Comparing (4) with (3), we see that (3) corresponds to the particular case where one chooses $k_m = 1$ and $p_w = -1$ leading to $a = h_w \eta_w(h_w)$. Thus as was already explained in Bonetti et al. (2018), using $\mathcal{Q}_w = a$ relates the erosion to (a power of) the water height and the slope. Both choices have pros and cons, however the choice $\mathcal{Q}_w = q_w$ seems more general to us.

The results of Coatléven (2020) explain why such a strong mesh dependency resisting mesh refinement was observed in the geological literature for the CA obtained from cell-to-cell MFD algorithms. It also explains how to compute a correct approximation of $q_w$ from the obtained CA which is the main objective of this section. We consequently start by recalling the results of Coatléven (2020) for cell-to-cell MFD in a slightly more general setting and to compute the sought approximation $q_K$ of $q_w$ in cell $K$. As node-to-node MFD algorithms are the core of many legacy codes, to offer a more straightforward application of the results of Coatléven (2020) for such implementations we next detail how the most classical node-to-node





MFD algorithms also enter this framework, suffer from the same deficiencies and how to correct them in the same way than for the cell-to-cell case that was already explored in Coatléven (2020). Notice that systems (3) and (4) are well-posed from the mathematical point of view if:

$$-\Delta\left(h_s + b\right) > 0 \quad \text{or} \ > 0, \tag{5}$$

(or quite equivalently $-div\left(k_m s_{ref}^{-p_w}||\nabla(h_s + b)||^{p_w}\nabla(h_s + b)\right) > 0$), i.e. roughly speaking if there are no water accumulation areas or flat areas (see Coatléven (2020); Bardos (1970); Veiga (1987); DiPerna and Lions (1989); Fernández-Cara et al. (2002); Girault and Tartar (2010)). This essentially implies that the flow model (4) is well justified for drainage basin only. If it used with topographies that do not fulfill (5), a modeling error can appear, ruining our efforts to achieve consistency. This assumption limits the domain of application of the model, and can be considered as the price to pay for a low computational cost strategy.

Model (4) being in fact a simplification of the shallow water equation, if extending the computation time is allowed alternative models also derived from the shallow water equation can be considered to overcome this limitation. This will be discussed in section 5.2.

### 3.1  Mesh description

Let $\Omega$ be a bounded polyhedral connected domain of $\mathbb{R}^2$, whose boundary is denoted $\partial\Omega = \overline{\Omega} \setminus \Omega$. We recall the usual finite
volume notations describing a mesh $\mathcal{M} = (\mathcal{T}, \mathcal{F})$ of $\Omega$. The set of the cells of the mesh $\mathcal{T}$ is a finite family of connected open disjoint polygonal subsets of $\Omega$, such that $\overline{\Omega} = \cup_{K \in \mathcal{T}} \overline{K}$. For any $K \in \mathcal{T}$, we denote by $|K|$ the measure of $|K|$, by $\partial K = \overline{K} \setminus K$ the boundary of $K$, by $\rho_K$ its diameter and by $\boldsymbol{x}_K$ its barycenter. The set of faces of the mesh $\mathcal{F}$ is a finite family of disjoint subsets of $\mathbb{R}^2$ included in $\overline{\Omega}$ such that, for all $\sigma \in \mathcal{F}$, its measure is denoted $|\sigma|$, its diameter $h_\sigma$ and its barycenter $\boldsymbol{x}_\sigma$ . For any $K \in \mathcal{T}$, the faces of cells $K$ corresponds to the subset $\mathcal{F}_K$ of $\mathcal{F}$ such that $\partial K = \cup_{\sigma \in \mathcal{F}_K} \overline{\sigma}$. Then, for any face $\sigma \in \mathcal{F}$, we
denote by $\mathcal{T}_\sigma = \{K \in \mathcal{T} \mid \sigma \in \mathcal{F}_K\}$ the cells of which $\sigma$ is a face. Next, for all cell $K \in \mathcal{T}$ and all face $\sigma \in \mathcal{F}_K$ of cell $K$, we denote by $\boldsymbol{n}_{K,\sigma}$ the unit normal vector to $\sigma$ outward to $K$, and $d_{K,\sigma} = |\boldsymbol{x}_\sigma - \boldsymbol{x}_K|$. The set of boundary faces is denoted $\mathcal{F}_{ext}$, while interior faces are denoted $\mathcal{F}_{int}$. Finally for any $\sigma \in \mathcal{F}_{int}$, whenever the context is clear we will denote by $K$ and $L$ the two cells forming $\mathcal{T}_\sigma = \{K, L\}$, as well as $d_{KL} = |\boldsymbol{x}_K - \boldsymbol{x}_L|$. This for instance allows when looping over the faces $\sigma$ of cell $K$ to denote by $L$ the other face of $\sigma$ without resorting to a too heavy notation. To avoid any confusion with water and sediment
heights, $\epsilon = \max_{K \in \mathcal{T}} \rho_K$ will denote the mesh size. For any continuous quantity $u$, its discrete counterpart will be denoted $u_\mathcal{T} = ((u_K)_{K \in \mathcal{T}}, (u_\sigma)_{\sigma \in \mathcal{F}_{ext}})$ where for any $K \in \mathcal{T}$ $u_K$ is the constant approximation of $u$ in cell $K$ while for any $\sigma \in \mathcal{F}_{ext}$ $u_\sigma$ is the constant approximation of $u$ over face $\sigma$.

In the following we will assume that the mesh is orthogonal, i.e. there exists a family of centroids $(\overline{\boldsymbol{x}}_K)_{K \in \mathcal{T}}$ such that:

$$\overline{\boldsymbol{x}}_K \in \overline{K} \quad \forall K \in \mathcal{T} \quad \text{and} \quad \frac{\overline{\boldsymbol{x}}_L - \overline{\boldsymbol{x}}_K}{|\overline{\boldsymbol{x}}_L - \overline{\boldsymbol{x}}_K|} = \boldsymbol{n}_{K,\sigma} \quad \text{for} \quad \sigma \in \mathcal{F}_{int}, \sigma = \{K, L\}$$

and let us denote $\overline{\boldsymbol{x}}_\sigma$ the orthogonal projection of $\overline{\boldsymbol{x}}_K$ to the hyperplane containing $\sigma$ for any $\sigma \in \mathcal{F}_K$ and any $K \in \mathcal{T}$ with $\overline{d}_{K,\sigma} = |\overline{\boldsymbol{x}}_K - \overline{\boldsymbol{x}}_\sigma|$, as well as $\overline{d}_{KL} = |\overline{\boldsymbol{x}}_K - \overline{\boldsymbol{x}}_L|$. Then, one can use a two-point finite volume scheme to discretize diffusion operators with scalar diffusion coefficients (no tensors).



## 3.2 The cell-to-cell multiple flow direction algorithm and its link with Gauckler-Manning-Strickler models

As mentioned above, the results of this subsection are mostly reproduced from Coatléven (2020). As a consequence, no true originality is claimed here however we believe that the node-to-node version will be easier to understand after this reminder.

The starting point of a finite volume discretization is to integrate equation (4) over each cell $K$:

$$-\int_K div\left(k_m h_w \eta_w(h_w) s_{ref}^{-p_w} ||\nabla(h_s+b)||^{p_w} \nabla(h_s+b)\right) = \int_K S_w.$$

Denoting $S_{w,K} = \frac{1}{|K|}\int_K S_w$ and using Stokes' formula, this leads to:

$$-\sum_{\sigma \in \mathcal{F}_K} \int_\sigma k_m h_w \eta_w(h_w) s_{ref}^{-p_w} ||\nabla(h_s+b)||^{p_w} \nabla(h_s+b) \cdot \boldsymbol{n}_{K,\sigma} = |K|S_{w,K}.$$

Choosing a finite volume scheme then simply amounts to choosing how to approximate each term appearing in the face integrals. The most natural and classical finite volume scheme consists in choosing constant approximate values $k_{m,\sigma}$ and $\boldsymbol{G}_{s,\sigma}$ for $k_m$ and $||\nabla(h_s+b)||^{pw}$ along each face $\sigma$ and to use an upwind scheme $h_{w,\sigma}^{up}$ for the true unknown $h_w \eta_w(h_w)$:

$$-\int_\sigma k_m h_w \eta_w(h_w) s_{ref}^{-p_w} ||\nabla(h_s+b)||^{p_w} \nabla(h_s+b) \cdot \boldsymbol{n} \approx -k_{m,\sigma} s_{ref}^{-p_w} ||\boldsymbol{G}_{s,\sigma}||^{p_w} h_{w,\sigma}^{up} \int_\sigma \nabla(h_s+b) \cdot \boldsymbol{n}.$$

Finally, thanks to our hypothesis on mesh orthogonality we can use the two-point flux approximation to compute $\int_\sigma \nabla(h_s+b)\cdot \boldsymbol{n}$. The TPFA consists in noticing that for a linear function $h_s+b$, the gradient being constant and satisfying $\nabla(h_s+b)\cdot\boldsymbol{n}_{K,\sigma} = \frac{1}{d_{KL}}((h_s+b)(\boldsymbol{x}_L)-(h_s+b)(\boldsymbol{x}_K))$, the following formula:

$$-\int_\sigma \nabla(h_s+b)\cdot\boldsymbol{n} = -\nabla(h_s+b)\cdot\int_\sigma \boldsymbol{n} = \frac{|\sigma|}{d_{KL}}((h_s+b)(\boldsymbol{x}_K)-(h_s+b)(\boldsymbol{x}_L)),$$

is exact since $\frac{1}{d_{KL}}(\boldsymbol{x}_L-\boldsymbol{x}_K)=\boldsymbol{n}_{K,\sigma}$ and will thus be a first order approximation of the flux. More precisely, denoting $h_{w,K}$ for any $K \in \mathcal{T}$ the discrete water height value associated to cell $K$, if one further assumes that $h_{s,\sigma}+b_\sigma = h_{s,K}+b_K$ for any $\sigma \in \mathcal{F}_{ext}$ and $K \in \mathcal{T}_\sigma$ which is generally what is done in practical applications of the MFD algorithm, for any $K \in \mathcal{T}$ the proposed finite volume scheme rewrites:

$$\sum_{\sigma \in \mathcal{F}_K \cap \mathcal{F}_{int}} \tau_{KL} h_{w,\sigma}^{up}(h_{s,K}+b_K-h_{s,L}-b_L) = |K|S_{w,K},$$

where the upwind value is given by $h_{up}^{w;\sigma} = h_{w,K}\eta_w(h_{w,K})$ if $h_{s,K}+b_K \geq h_{s,L}+b_L$ and $h_{up}^{w;\sigma}=h_{w,L}\eta_w(h_{w,L})$ if $h_{s,K}+b_K < h_{s,L}+b_L$, the transmissivity $\tau_{KL}$ is given by:

$$\tau_{KL} = \frac{|\sigma|k_{m,\sigma}}{d_{KL}s_{ref}^{-p_w}}||\boldsymbol{G}_{s,\sigma}||^{p_w},$$

and where $\boldsymbol{G}_{s,\sigma}=\frac{1}{2}(\boldsymbol{G}_{s,K}+\boldsymbol{G}_{s,L})$ and $\boldsymbol{G}_{s,K}$ is a discrete reconstruction of the gradient of $h_s+b$ in cell $K$. To derive it, we use:

$$\mathbb{I}_d = \sum_{\sigma \in \mathcal{F}_K} |\sigma|(\boldsymbol{x}_\sigma-\boldsymbol{x}_K)\boldsymbol{n}_{K,\sigma}, \tag{6}$$



leading to

$$\boldsymbol{G}_{s,K} = \sum_{\sigma \in \mathcal{F}_K} |\sigma| \boldsymbol{G}_{s,K} \cdot \boldsymbol{n}_{K,\sigma} (\boldsymbol{x}_\sigma - \boldsymbol{x}_K),$$

and thus on the orthogonal meshes we consider here as by consistency $|\sigma| \boldsymbol{G}_{s,K} \cdot \boldsymbol{n}_{K,\sigma} \approx \int_\sigma \nabla(h_s + b) \cdot \boldsymbol{n}_{K,\sigma}$, $\boldsymbol{G}_{s,K}$ is naturally given by:

$$\boldsymbol{G}_{s,K} = \frac{1}{|K|} \sum_{\sigma \in \mathcal{F}_K \cap \mathcal{F}_{int}} \frac{|\sigma|}{\overline{d}_{KL}} (h_{s,L} + b_L - h_{s,K} - b_K)(\boldsymbol{x}_\sigma - \boldsymbol{x}_K)$$

$$+ \frac{1}{|K|} \sum_{\sigma \in \mathcal{F}_K \cap \mathcal{F}_{ext}} \frac{|\sigma|}{\overline{d}_{K\sigma}} (h_{s,\sigma} + b_\sigma - h_{s,K} - b_K)(\boldsymbol{x}_\sigma - \boldsymbol{x}_K).$$

From the mathematical point of view, a natural choice for the face value $k_{m,\sigma}$ is the harmonic mean:

$$k_{m,\sigma} = \frac{\overline{d}_{KL} k_{m,K} k_{m,L}}{k_{m,K} \overline{d}_{L,\sigma} + k_{m,L} \overline{d}_{K,\sigma}} \quad \text{with for instance} \quad k_{m,K} = \frac{1}{|K|} \int_K k_m \ \forall K \in \mathcal{T},$$

but many other choices are possible. Let us now recall the elementary proof given in Coatléven (2020): gathering the faces by upwinding kind, we get:

$$\sum_{\sigma \in \mathcal{F}_K \cap \mathcal{F}_{int}, h_{s,K} + b_K \geq h_{s,L} + b_L} \tau_{KL} h_{w,K} \eta_w(h_{w,K})(h_{s,K} + b_K - h_{s,L} - b_L) -$$

$$\sum_{\sigma \in \mathcal{F}_K \cap \mathcal{F}_{int}, b_K < b_L} \tau_{KL} h_{w,L} \eta_w(h_{w,L})(h_{s,L} + b_L - h_{s,K} - b_K) = |K| S_{w,K}. \tag{7}$$

Setting

$$s_K = \sum_{\sigma \in \mathcal{F}_K \cap \mathcal{F}_{int}, h_{s,K} + b_K \geq h_{s,L} + b_L} \tau_{KL}(h_{s,K} + b_K - h_{s,L} - b_L),$$

and noticing that $s_L > 0$ as soon as there exists $\sigma \in \mathcal{F}_L \cap \mathcal{F}_{int}$ such that $b_L > b_K$, we see that equation (7) can be rewritten:

$$s_K h_{w,K} \eta_w(h_{w,K}) - \sum_{\sigma \in \mathcal{F}_K \cap \mathcal{F}_{int}, b_K < b_L} \tau_{KL} h_{w,L} \eta_w(h_{w,L})(h_{s,L} + b_L - h_{s,K} - b_K) = |K| S_{w,K}.$$

Defining the water outflux by $\widetilde{q}_K = s_K h_{w,K} \eta_w(h_{w,K})$, we thus obtain:

$$\widetilde{q}_K - \sum_{\sigma \in \mathcal{F}_K \cap \mathcal{F}_{int}, h_{s,K} + b_K < h_{s,L} + b_L} \tau_{KL} \frac{\widetilde{q}_L}{s_L}(h_{s,L} + b_L - h_{s,K} - b_K) = |K| S_{w,K}. \tag{8}$$

The cell-to-cell MFD algorithm admits a reformulation as a linear system first mentioned by Richardson et al. (2014) although without exhibiting an explicit formula. In Coatléven (2020), it is established that linear system underlying the cell-to-cell MFD algorithm illustrated on Fig. 2 is equivalent to solving (8) for $k_m = 1$ and $p_w = 0$ using a lower triangular solver and a cell




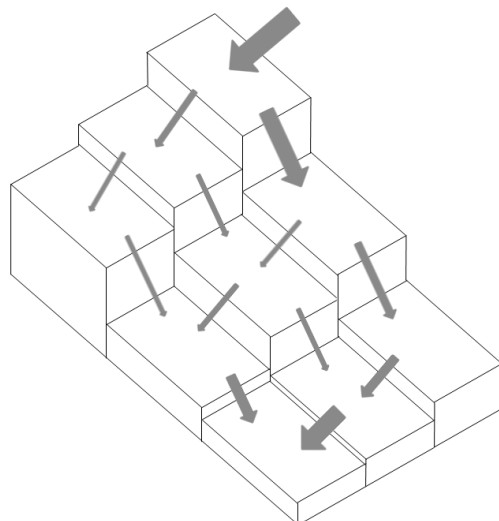

**Figure 2.** Basic principle of the simplest cell-to-cell MFD algorithm: water is distributed to lower neighbouring cells proportionally to the slope (reproduced from Coatléven (2020))

ordering based on decreasing topography. Indeed the algorithm illustrated Fig. 2 consists in distributing the total outflow $\widetilde{q}_K$ of cell $K$ along the neighbouring cells of $K$ with lower altitude altitude proportionally to the ratio $s_{KL}/s_K$ of the discrete

slope $s_{KL}$ between the high cell $K$ and the low cell $L$ regarding the total positive slope $s_K$ of the high cell $K$. It is then easy to observe that the formula (8) corresponds to reversing this idea, expressing how the inflow received by the low cell $K$ is computed from the outflow of its higher neighbours.

From this equivalence between the classical MFD and the two-point flux approximation (TPFA) of the classical Gauckler-Manning-Strickler model, a crucial observation of Coatléven (2020) is that the usual unknown $\widetilde{q}_K$ of the MFD algorithm that

corresponds to the CA of cell $K$ in the case $S_w = 1$ is not the good quantity to represent the water flux magnitude. Indeed, from $\widetilde{q}_K = s_K h_{w,K} \eta_w(h_{w,K})$ and the consistency of the two-point formula we see that it approximates as announced:

$$\widetilde{q}_K \approx \widetilde{q}_K^{ex} = \sum_{\sigma \in \mathcal{F}_K} \int_\sigma h_w \eta_w(h_w) \left( -k_m s_{ref}^{-p_w} ||\nabla(h_s + b)||^{p_w} \nabla(h_s + b) \cdot \boldsymbol{n}_{K,\sigma} \right)^+ .$$

As explained in Coatléven (2020) the quantity $\widetilde{q}_K$ approximates the outflux of a cell which thanks to the equivalence with a discretization of a Gauckler-Manning-Strickler model we can easily identify as a mesh dependent quantity. Thus, the only

convergence that can be expected for $\widetilde{q}_K$ is to zero. As explained in the introductory part of this section we could normalize it by the portion of $\partial K$ along which the flow is outgoing but this is highly impractical and still prone to some mesh dependency depending on the cell orientation with respect to the flux. To effectively compute an accurate discrete water flux magnitude $q_K$ for each cell $K \in \mathcal{T}$, from Coatléven (2020) we know that we can reconstruct cellwise the water flux vector using (6) by





setting:


$$\boldsymbol{Q}_K = \sum_{\sigma \in \mathcal{F}_K \cap \mathcal{F}_{int}, h_{s,K}+b_K > h_{s,L}+b_L} \frac{\tau_{KL}\widetilde{q}_K}{|K|s_K}(h_{s,K}+b_K-h_{s,L}-b_L)(\boldsymbol{x}_\sigma - \boldsymbol{x}_K)-$$

$$\sum_{\sigma \in \mathcal{F}_K \cap \mathcal{F}_{int}, h_{s,K}+b_K < h_{s,L}+b_L} \frac{\tau_{KL}\widetilde{q}_L}{|K|s_L}(h_{s,L}+b_L-h_{s,K}-b_K)(\boldsymbol{x}_\sigma - \boldsymbol{x}_K), \tag{9}$$

and simply deduce a consistent water flux magnitude by setting $q_K = ||\boldsymbol{Q}_K||$. This consistent water flux magnitude is mesh independent in the usual numerical analysis sense: it converges to the continuous flux when the mesh size $\varepsilon$ goes to zero,

contrary to $\widetilde{q}_K$. The use of $\widetilde{q}_K$ or its normalized versions instead of $q_K$ in the geological literature is the main reason why such a strong mesh dependency was observed, without any significant improvement with mesh refinement. Instead, the convergence of the consistent water flux magnitude $q_K$ was rigorously established and illustrated in Coatléven (2020), up to providing error estimates. Thus, it is important to use $q_K$ instead of $\widetilde{q}_K$ when coupling with sediment evolution models i.e. using $\mathcal{Q}_w = q_w$ in (17) and not $\widetilde{q}_K$. From the flow routing literature perspective and by virtue of (9), $q_K$ can certainly be considered as a

post-processing consistency correction of $\widetilde{q}_K$, easy to implement in legacy softwares.

The MFD formulation allows in turn some interesting observations for the Gauckler-Manning-Strickler model: it is indeed clear that the choice of the water mobility function $\eta_w$ has no influence on the water flux strength $q_w$, as it appears nowhere in (8) and (9). In the same way, only the contrasts of the coefficient $k_m$ will impact $q_w$, as only ratios $\tau_{KL}/s_K$ are appearing in (8) and (9).

### 3.3 The classical node-to-node MFD/SFD algorithms interpreted as discrete Gauckler-Manning-Strickler solvers

In this subsection, we will explain how to reinterpret the most classical node-to-node flow routing algorithms as attempts to discretize a continuous Gauckler-Manning-Strickler model. Such an explicit interpretation seems to be absent from the literature, so at least to the author's knowledge the results of this subsection are quite new. To this end, for simplicity we restrict ourselves in this section to uniform cartesian meshes, and we adopt the usual cartesian index $(i,j)$ notation for designating

its nodes (see Fig. 3) as well as $\Delta x$ and $\Delta y$ for the cartesian cell side lengths. This is by no means a restriction but simply a more convenient way to explain how to link node-to-node flow routing with Gauckler-Manning-Strickler models. In order to reinterpret the node-to-node flow routing algorithms as finite volume schemes, we must associate a volume to each node. The easiest way to do so is to consider the dual mesh, formed by joining the centers of the cells of the primal mesh (see again Fig. 3, where the dual mesh corresponds to the dashed lines). On the dual mesh, the node $(i,j)$ of the primal mesh becomes the

center of the dual cell $K_{i,j}$.

In Fig. 4, we propose a decomposition of the boundary of the dual cartesian cell $K_{i,j}$ centered on the primal node $(i,j)$ into 12 faces $(\sigma_l)_{1 \leq l \leq 12}$. The faces $\sigma_{j\pm 1}$ are of length $\gamma_x \Delta x$, with of course the faces $\sigma_{j\pm 1}^{i\pm 1}$ of length $\frac{1-\gamma_x}{2}\Delta x$. In the same way, faces $\sigma_{i\pm 1}$ are of length $\gamma_y \Delta y$ and the faces $\sigma_{i\pm 1}^{j\pm 1}$ of length $\frac{1-\gamma_y}{2}\Delta y$. Using those notations, we integrate (4) over the dual cell $K_{i,j}$



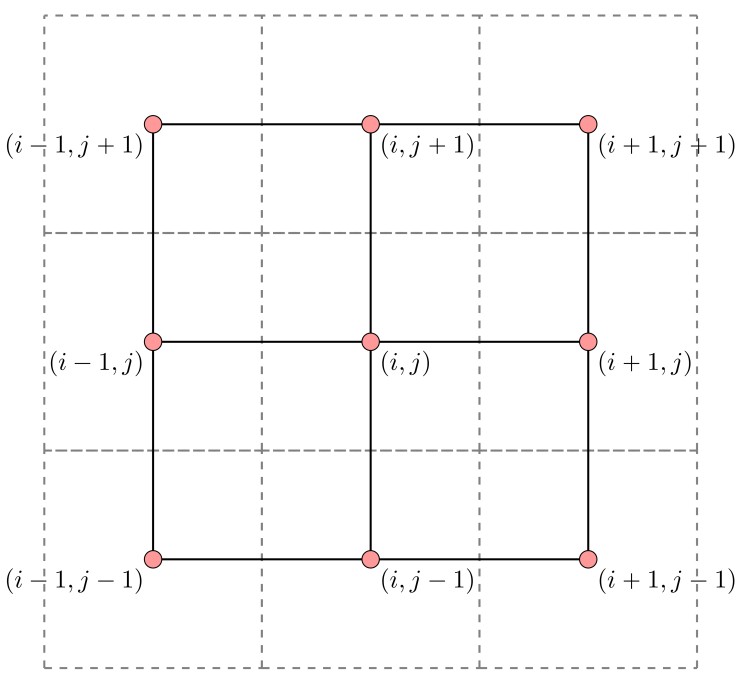

**Figure 3.** The cartesian mesh (plain lines) and its dual (dashed lines)

to get:

$$-\sum_{l=1}^{12}\int_{\sigma_l} k_m h_w \eta_w(h_w) s_{ref}^{-p_w} ||\nabla(h_s+b)||^{p_w} \nabla(h_s+b)\cdot\boldsymbol{n}_{K_{i,j}} = |K_{i,j}|S_{w,K}.$$

On the four faces $\sigma_{i-1}$, $\sigma_{i+1}$, $\sigma_{j-1}$ and $\sigma_{j+1}$, we use the same finite volume discretization than before:

$$\int_{\sigma_{j-1}} k_m h_w \eta_w(h_w) s_{ref}^{-p_w} ||\nabla(h_s+b)||^{p_w} \nabla(h_s+b)\cdot\boldsymbol{n}_{K_{i,j}}$$

$$\approx \frac{\gamma_x \Delta x}{s_{ref}^{p_w} \Delta y} k_{m,\sigma_{j-1}} ||\boldsymbol{G}_{s,\sigma_{j-1}}||^{p_w} h_{w,\sigma_{j-1}}^{up}(h_{s,i,j-1}+b_{i,j-1}-h_{s,i,j}-b_{i,j}),$$

and

$$\int_{\sigma_{i-1}} k_m h_w \eta_w(h_w) s_{ref}^{-p_w} ||\nabla(h_s+b)||^{p_w} \nabla(h_s+b)\cdot\boldsymbol{n}_{K_{i,j}}$$

$$\approx \frac{\gamma_y \Delta y}{s_{ref}^{p_w} \Delta x} k_{m,\sigma_{i-1}} ||\boldsymbol{G}_{s,\sigma_{i-1}}||^{p_w} h_{w,\sigma_{i-1}}^{up}(h_{s,i-1,j}+b_{i-1,j}-h_{s,i,j}-b_{i,j}),$$



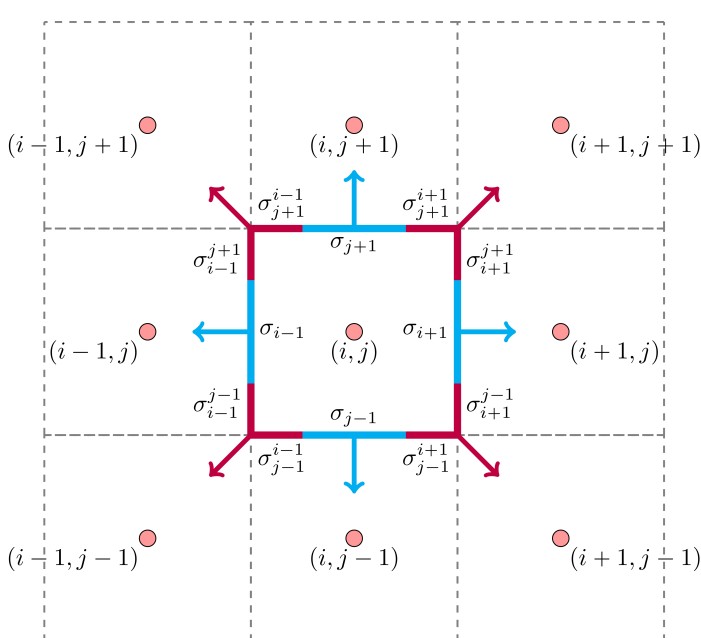

**Figure 4.** Decomposition and notations for the dual cartesian cell boundaries

and

$$\int\limits_{\sigma_{j+1}} k_m h_w \eta_w(h_w) s_{ref}^{-p_w} ||\nabla(h_s + b)||^{p_w} \nabla(h_s + b) \cdot \boldsymbol{n}_{K_{i,j}}$$

$$\approx \frac{\gamma_x \Delta x}{s_{ref}^{p_w} \Delta y} k_{m,\sigma_{j+1}} ||\boldsymbol{G}_{s,\sigma_{j+1}}||^{p_w} h_{w,\sigma_{j+1}}^{up}(h_{s,i,j+1} + b_{i,j+1} - h_{s,i,j} - b_{i,j}),$$

and

$$\int\limits_{\sigma_{i+1}} k_m h_w \eta_w(h_w) s_{ref}^{-p_w} ||\nabla(h_s + b)||^{p_w} \nabla(h_s + b) \cdot \boldsymbol{n}_{K_{i,j}}$$

$$\approx \frac{\gamma_y \Delta y}{s_{ref}^{p_w} \Delta x} k_{m,\sigma_{i+1}} ||\boldsymbol{G}_{s,\sigma_{i-1}}||^{p_w} h_{w,\sigma_{i+1}}^{up}(h_{s,i+1,j} + b_{i+1,j} - h_{s,i,j} - b_{i,j}),$$

while for the remaining height cells, we gather the faces to form the corners illustrated in Fig. (4). More precisely, we denote:

$$\overline{\sigma}_{i-1,j-1} = \overline{\sigma}_{i-1}^{j-1} \cup \overline{\sigma}_{j-1}^{i-1},$$


$$\overline{\sigma}_{i-1,j+1} = \overline{\sigma}_{i-1}^{j+1} \cup \overline{\sigma}_{j+1}^{i-1},$$

$$\overline{\sigma}_{i+1,j-1} = \overline{\sigma}_{i+1}^{j-1} \cup \overline{\sigma}_{j-1}^{i+1},$$



$\overline{\sigma}_{i+1,j+1} = \overline{\sigma}_{i+1}^{j+1} \cup \overline{\sigma}_{j+1}^{i+1},$

those four corners, $\sigma_{i\pm1,j\pm1}$ thus being the corner corresponding to the neighbouring cell $K_{i\pm1,j\pm1}$. On those corners, we perform the same discretization than before considering the whole corner as if it was a single face: in other words we use constant values $k_{m,\sigma}$ and $||\boldsymbol{G}_{s,\sigma}||^{p_w}$ for $k_m$ and $||\nabla(h_s + b)||^{p_w}$ along the corner, an upwind scheme for the unknown $h_w\eta_w(h_w)$ and the a two-point flux formula for in the average normal direction to the corner. Denoting $(\nabla(h_s + b))_{\sigma_{i\pm1,j\pm1}}$ the equivalent

constant gradient exact for linear function underlying the TFPA along the corner, this leads to the following approximation:

$$\int_{\sigma_{i\pm1,j\pm1}} k_m h_w \eta_w(h_w) s_{ref}^{-p_w} ||\nabla(h_s + b)||^{p_w} \nabla(h_s + b) \cdot \boldsymbol{n}_{K_{i,j}} \approx$$

$$k_{m,\sigma_{i\pm1,j\pm1}} s_{ref}^{-p_w} ||\boldsymbol{G}_{s,\sigma_{i\pm1,j\pm1}}||^{p_w} h_{w,\sigma_{i\pm1,j\pm1}}^{up} (\nabla(h_s + b))_{\sigma_{i\pm1,j\pm1}} \cdot \int_{\sigma_{i\pm1,j\pm1}} \boldsymbol{n}_{K_{i,j}}.$$

By construction, we have:

$$\int_{\sigma_{i\pm1,j\pm1}} \boldsymbol{n}_{K_{i,j}} = \pm\frac{(1-\gamma_y)}{2}\Delta y \boldsymbol{e_x} \pm \frac{(1-\gamma_x)}{2}\Delta x \boldsymbol{e_y}.$$

Denoting

$$|\sigma_{i\pm1,j\pm1}| = \frac{(1-\gamma_x)}{2}\Delta x + \frac{(1-\gamma_y)}{2}\Delta y = \delta,$$

we seek $\gamma_x$ and $\gamma_y$ such that:

$$\frac{(1-\gamma_x)}{2\delta}\Delta x = \frac{\Delta y}{(\Delta x^2 + \Delta y^2)^{1/2}} \quad \text{and} \quad \frac{(1-\gamma_x)}{2\delta}\Delta y = \frac{\Delta x}{(\Delta x^2 + \Delta y^2)^{1/2}},$$

leading to:

$$\gamma_x = 1 - \frac{2\delta\Delta y/\Delta x}{(\Delta x^2 + \Delta y^2)^{1/2}} \quad \text{and } \gamma_y = 1 - \frac{2\delta\Delta x/\Delta y}{(\Delta x^2 + \Delta y^2)^{1/2}}, \tag{10}$$

which can be achieved with $\gamma_x \geq 0$ and $\gamma_y \geq 0$ provided $\delta$ satisfies:

$$0 \leq \delta \leq \frac{1}{2}\min\left(\frac{\Delta x}{\Delta y}, \frac{\Delta y}{\Delta x}\right)(\Delta x^2 + \Delta y^2)^{1/2}. \tag{11}$$

With this choice for $\gamma_x$ and $\gamma_y$, for all $\delta$ satisfying (11) we get that

$$\int_{\sigma_{i\pm1,j\pm1}} \boldsymbol{n}_{K_{i,j}} = \frac{\pm\Delta x}{(\Delta x^2 + \Delta y^2)^{1/2}}\boldsymbol{e_x} + \frac{\pm\Delta y}{(\Delta x^2 + \Delta y^2)^{1/2}}\boldsymbol{e_y},$$

and thus the average normal at the corner $\sigma_{i\pm1,j\pm1}$ is precisely pointing from $\boldsymbol{x}_{K_{i,j}}$ to $\boldsymbol{x}_{K_{i\pm1,j\pm1}}$. Thus it is natural to use the two point flux formula:

$$(\nabla(h_s + b))_{\sigma_{i\pm1,j\pm1}} \cdot \int_{\sigma_{i\pm1,j\pm1}} \boldsymbol{n}_{K_{i,j}} \approx \frac{\delta}{(\Delta x^2 + \Delta y^2)^{1/2}}(h_{s,i\pm1,j\pm1} + b_{i\pm1,j\pm1} - h_{s,i} - b_i).$$





The upwinding is done exactly as before, following the sign of the difference in elevation $h_s + b$ between the two value forming the TPFA. This gives for the non-corners:

$$
h^{up}_{w,\sigma_{i\pm1}} = \left|
\begin{array}{ll}
h_{w,i,j}\eta_w(h_{w,i,j}) & \text{if } h_{s,i,j} + b_{i,j} \geq h_{s,i\pm1,j} + b_{i\pm1,j}, \\[8pt]
h_{w,i\pm1,j}\eta_w(h_{w,i\pm1,j}) & \text{if } h_{s,i,j} + b_{i,j} < h_{s,i\pm1,j} + b_{i\pm1,j},
\end{array}
\right.
$$

$$
h^{up}_{w,\sigma_{j\pm1}} = \left|
\begin{array}{ll}
h_{w,i,j}\eta_w(h_{w,i,j}) & \text{if } h_{s,i,j} + b_{i,j} \geq h_{s,i,j\pm1} + b_{i,j\pm1}, \\[8pt]
h_{w,i,j\pm1}\eta_w(h_{w,i,j\pm1}) & \text{if } h_{s,i,j} + b_{i,j} < h_{s,i,j\pm1} + b_{i,j\pm1},
\end{array}
\right.
$$

and for the corners:

$$
h^{up}_{w,\sigma_{i\pm1,j\pm1}} = \left|
\begin{array}{ll}
h_{w,i,j}\eta_w(h_{w,i,j}) & \text{if } h_{s,i,j} + b_{i,j} \geq h_{s,i\pm1,j\pm1} + b_{i\pm1,j\pm1}, \\[8pt]
h_{w,i\pm1,j\pm1}\eta_w(h_{w,i\pm1,j\pm1}) & \text{if } h_{s,i,j} + b_{i,j} < h_{s,i\pm1,j\pm1} + b_{i\pm1,j\pm1}.
\end{array}
\right.
$$

To get more compact notations, let us denote

$$
\mathcal{N}(i,j) = \{(m,n) \in \{i-1,i,i+1\} \times \{j-1,j,j+1\} \mid (m,n) \leq (i,j)\},
$$

the neighbours of node $(i,j)$, and define the transmissivities:

$$
\tau^{m,n}_{i,j} = \left|
\begin{array}{ll}
\dfrac{\gamma_x \Delta x}{s^{p_w}_{ref}\Delta y}k_{m,\sigma_{j\pm1}}||\boldsymbol{G}_{s,\sigma_{j\pm1}}||^{p_w} & \text{if } (m,n) = (i,j-1) \text{ or } (i,j+1), \\[12pt]
\dfrac{\gamma_y \Delta y}{s^{p_w}_{ref}\Delta x}k_{m,\sigma_{i\pm1}}||\boldsymbol{G}_{s,\sigma_{i\pm1}}||^{p_w} & \text{if } (m,n) = (i-1,j) \text{ or } (i+1,j), \\[12pt]
\dfrac{\delta}{s^{p_w}_{ref}(\Delta x^2 + \Delta y^2)^{1/2}}k_{m,\sigma_{i\pm1,j\pm1}}||\boldsymbol{G}_{s,\sigma_{i\pm1,j\pm1}}||^{p_w} & \text{otherwise,}
\end{array}
\right.
$$

assuming for simplicity that the gradients $\boldsymbol{G}_{s,\sigma}$ are obtained on the dual mesh in the same way as in the cell-to-cell case (of course, a reconstruction formula using also the diagonal neighbours is possible). Using those notations, we get gathering by upwind kind as in the case of the cell-to-cell flow routing the following expression for the proposed finite volume scheme on the dual mesh:

$$
h_{w,i,j}\eta_w(h_{w,i,j}) \left( \sum_{(m,n)\in\mathcal{N}(i,j),\, h_{s,i,j}+b_{i,j}>h_{s,m,n}+b_{m,n}} \tau^{m,n}_{i,j}(h_{s,i,j} + b_{i,j} - h_{s,m,n} - b_{m,n}) \right)
$$

$$
- \left( \sum_{(m,n)\in\mathcal{N}(i,j),\, h_{s,i,j}+b_{i,j}<h_{s,m,n}+b_{m,n}} \tau^{m,n}_{i,j} h_{w,m,n}\eta_w(h_{w,m,n})(h_{s,m,n} + b_{m,n} - h_{s,i,j} - b_{i,j}) \right) = |K_{i,j}|S_{w,i,j}.
$$

Proceeding as in the cell-to-cell case, denoting:

$$
s_{i,j} = \sum_{(m,n)\in\mathcal{N}(i,j),\, h_{s,i,j}+b_{i,j}>h_{s,m,n}+b_{m,n}} \tau^{m,n}_{i,j}(h_{s,i,j} + b_{i,j} - h_{s,m,n} - b_{m,n}) \quad \text{and} \quad \widetilde{q}_{i,j} = h_{w,i,j}\eta_w(h_{w,i,j})s_{i,j}
$$





, we finally get:

$$\widetilde{q}_{i,j} - \left( \sum_{(m,n)\in\mathcal{N}(i,j),\, h_{s,i,j}+b_{i,j}<h_{s,m,n}+b_{m,n}} \tau_{i,j}^{m,n}\frac{\widetilde{q}_{m,n}}{s_{m,n}}(h_{s,m,n}+b_{m,n}-h_{s,i,j}-b_{i,j}) \right) = |K_{i,j}|S_{w,i,j}. \tag{12}$$

The flow sharing formula common to all flow routing algorithms of the literature identifies in this context with the ratios:

$$\frac{1}{s_{m,n}}\tau_{i,j}^{m,n}(h_{s,m,n}+b_{m,n}-h_{s,i,j}-b_{i,j}),$$

for $(m,n)\in\mathcal{N}(i,j), h_{s,i,j}+b_{i,j} < h_{s,m,n}+b_{m,n}$, which expresses how node $(i,j)$ receives water from other nodes. Reversing the point of view, it rewrites in probably more familiar fashion by expressing how node $(i,j)$ distributes water to its neighbours

through the flow sharing formula (noticing that $\tau_{i,j}^{m,n} = \tau_{m,n}^{i,j}$):

$$\frac{\tau_{i,j}^{m,n}\max(0, h_{s,i,j}+b_{i,j}-h_{s,m,n}-b_{m,n})}{\displaystyle\sum_{m',n'\in\mathcal{N}(i,j)} \tau_{i,j}^{m',n'}\max(0, h_{s,i,j}+b_{i,j}-h_{s,m',n'}-b_{m',n'})}. \tag{13}$$

Notice that several attempts of the literature at improving the behavior of the flow routing consider powers $q$ of the two point slope instead of the slope in the flow sharing formula, which with our notations rewrites:

$$\frac{\tau_{i,j}^{m,n}\max(0, h_{s,i,j}+b_{i,j}-h_{s,m,n}-b_{m,n})^q}{\displaystyle\sum_{m',n'\in\mathcal{N}(i,j)} \tau_{i,j}^{m',n'}\max(0, h_{s,i,j}+b_{i,j}-h_{s,m',n'}-b_{m',n'})^q}. \tag{14}$$

Another important consequence of the formal identification of cell-to-cell flow routing algorithms with a numerical scheme for the stationary Gauckler-Manning-Strickler model is the fact that if one wants to incorporate powers of the slope in the flow distribution procedure, then one should not use powers of the directional slope $\frac{1}{\overline{d}_{KL}}(h_{s,L}+b_L-h_{s,K}-b_K)$ but rather use powers of $||\boldsymbol{G}_{s,\sigma}||$ to remain consistent with a continuous model incorporating powers of $||\nabla(h_s+b)||$. Otherwise, the consistency of the flow routing algorithm will be lost again. In Quinn et al. (1995) it is even suggested to choose different

values of $q$ for different grid sizes, emphasizing this non-consistency. However, the sought flow concentration effect can be achieved in a consistent manner through the use of $p_w$: the full gradient and not only the directional gradient being used this way, this does not endanger consistency and a value independent of the mesh should be chosen according to physical considerations. An option that we do not consider here is to make the value of $p_w$ spatially variable, as was suggested in Qin et al. (2007) but still on the non-consistent formulation (14).

Although (14) clearly leads to some non consistency, this expression is useful to derive a classification of the most prominent flow routing algorithms of the literature. To exactly match the definitions of most node-to-node flow routing schemes of the literature, we now consider the special case of square cartesian cells for which $\Delta x = \Delta y = \Delta_{xy}$. In this case we get from (10) that $\gamma_x = \gamma_y = 1 - (2\delta)/(\sqrt{2}\Delta_{xy})$. It remains to choose a value for $\delta$. The most natural choice is choose to enforce $\delta = \gamma_x\Delta x = \gamma_y\Delta y$ and thus balance the contribution to each neighbour. This immediately leads to:

$$\delta = \frac{\sqrt{2}}{2+\sqrt{2}}\Delta_{xy} \quad \text{and} \quad \gamma_x = \gamma_y = \frac{\sqrt{2}}{2+\sqrt{2}},$$





implying that:

$$\frac{\delta}{(\Delta x^2 + \Delta y^2)^{1/2}} = \frac{1}{2 + \sqrt{2}} \quad \text{and} \quad \frac{\gamma_x \Delta x}{\Delta y} = \frac{\sqrt{2}}{2 + \sqrt{2}} \quad \text{and} \quad \frac{\gamma_y \Delta y}{\Delta x} = \frac{\sqrt{2}}{2 + \sqrt{2}},$$

thus the diagonal transmissivities differ from the non-diagonal ones by the factor $1/\sqrt{2}$ which corresponds to the D8, Rho8 and most MFD algorithms. To recover the FD8/TOPMODEL noticing that the $L_1$ and $L_2$ non diagonal and diagonal "face measures" of this MFD algorithm satisfy $L_1 = \Delta_{xy}/2$ and $L_2 = \frac{\sqrt{2}}{4}\Delta_{xy}$, we recover the same weighting within our notations by setting

$$\delta = \frac{\sqrt{2}}{4}\Delta_{xy} \quad \text{and} \quad \gamma_x = \gamma_y = \frac{1}{2},$$

which is compatible with (10) as in this case:

$$\frac{(1 - \gamma_x)}{2\delta}\Delta x = \frac{(1 - \gamma_y)}{2\delta}\Delta y = \frac{1}{\sqrt{2}} = \frac{\Delta x}{(\Delta x^2 + \Delta y^2)^{1/2}} = \frac{\Delta y}{(\Delta x^2 + \Delta y^2)^{1/2}}.$$

Finally denoting:

$$\Delta \mathcal{H}_{i,j}^{m,n} = \max(0, h_{s,i,j} + b_{i,j} - h_{s,m,n} - b_{m,n}),$$

in table 1 we recast the most classical MFD algorithms using our notations, with $p_w = 0$ for all the presented methods. For the Rho8 method (Fairfield and Leymarie (1991)), the $\rho_8$ parameter is a random number generated for each face, while for the MFD-md (Qin et al. (2007)), the parameter $e$ is the maximum downslope gradient and $f(e) = 8.9\min(e, 1) + 1.1$.

This reinterpretation calls for several comments. The main one is that the node-to-node situation is no better than the cell-to-cell one: $\widetilde{q}_{i,j}$ will be as non consistent, non convergent and thus strongly mesh dependent than its cell-to-cell counterpart. The node-to-node routing is indeed simply a cell-to-cell routing on a dual mesh, with a more involved cell boundary decomposition. The quantity $\widetilde{q}_{i,j}$ should not be used to couple with sediment evolution, one should instead reconstruct a consistent water flux vector $\boldsymbol{Q}_{i,j}$ for instance by setting:

$$\boldsymbol{Q}_{i,j} = \sum_{(m,n) \in \mathcal{N}(i,j), h_{s,i,j} + b_{i,j} > h_{s,m,n} + b_{m,n}} \frac{\tau_{i,j}^{m,n}\widetilde{q}_{i,j}}{|K_{i,j}|s_{i,j}}(h_{s,i,j} + b_{i,j} - h_{s,m,n} - b_{m,n})(\boldsymbol{x}_{i,j}^{m,n} - \boldsymbol{x}_{K_{i,j}}) -$$

$$\sum_{(m,n) \in \mathcal{N}(i,j), h_{s,i,j} + b_{i,j} < h_{s,m,n} + b_{m,n}} \frac{\tau_{i,j}^{m,n}\widetilde{q}_{m,n}}{|K_{i,j}|s_{m,n}}(h_{s,m,n} + b_{m,n} - h_{s,i,j} - b_{i,j})(\boldsymbol{x}_{i,j}^{m,n} - \boldsymbol{x}_{K_{i,j}}) \quad (15)$$

where:

$$\boldsymbol{x}_{i,j}^{m,n} = \begin{vmatrix} \frac{1}{2}(\boldsymbol{x}_{K_{i,j}} + \boldsymbol{x}_{K_{m,n}}) & \text{if } (m,n) \in \{(i, j-1), (i, j+1), (i-1, j), (i+1, j)\} \\ \frac{1}{|\sigma_m^n| + |\sigma_n^m|}\left(|\sigma_m^n|\boldsymbol{x}_{\sigma_m^n} + |\sigma_n^m|\boldsymbol{x}_{\sigma_n^m}\right) & \text{otherwise} \end{vmatrix}$$

and then use $q_{i,j} = ||\boldsymbol{Q}_{i,j}||$ which again can be considered as an easy to implement post-processing consistency correction step. The second one is that it is clear that contrary to what is done in some flow routing algorithms of the literature, the chosen value



**Table 1.** A possible classification of MFD algorithms using (14)

| Method | $\delta/\Delta_{xy}$ | $\gamma_x = \gamma_y$ | $q$ | $k_{m,\sigma_{m,n}}$ | |
|---|---|---|---|---|---|
| D8 (O'Callaghan et al. 1984 O'Callaghan and Mark (1984)) | $\frac{\sqrt{2}}{2+\sqrt{2}}$ | $\frac{\sqrt{2}}{2+\sqrt{2}}$ | 1 | 1 | $\sigma_{m,n}$ has largest $\Delta\mathcal{H}_{i,j}^{m,n}$ |
| | | | | 0 | otherwise |
| MFD (Freeman 1989 Freeman (1989)) | $\frac{\sqrt{2}}{2+\sqrt{2}}$ | $\frac{\sqrt{2}}{2+\sqrt{2}}$ | 1 | 1 | |
| MFD (Freeman 1991 Freeman (1991)) | $\frac{\sqrt{2}}{2+\sqrt{2}}$ | $\frac{\sqrt{2}}{2+\sqrt{2}}$ | 1.1 | 1 | |
| Rho8 (Fairfield 1991 Fairfield and Leymarie (1991)) | $\frac{\sqrt{2}}{2+\sqrt{2}}$ | $\frac{\sqrt{2}}{2+\sqrt{2}}$ | 1 | 1 | $\sigma_{m,n}$ has largest $\rho_8\Delta\mathcal{H}_{i,j}^{m,n}$ |
| | | | | 0 | otherwise |
| FD8 (Quinn et al. 1991 Quinn et al. (1991)) | $\frac{\sqrt{2}}{4}$ | $\frac{1}{2}$ | 1 | 1 | |
| MFD (Holmgren 1994 Holmgren (1994)) | $\frac{\sqrt{2}}{2+\sqrt{2}}$ | $\frac{\sqrt{2}}{2+\sqrt{2}}$ | $\in [1,\infty[$ | 1 | |
| TOPMODEL (Quinn et al. 1995 Quinn et al. (1995)) | $\frac{\sqrt{2}}{4}$ | $\frac{1}{2}$ | $\in [1-100]$ | 1 | |
| MFD-md (Qin et al. 2007 Qin et al. (2007)) | $\frac{\sqrt{2}}{4}$ | $\frac{1}{2}$ | $f(e)$ | 1 | |

for $k_{m,\sigma}$ should be a discretization of an inverse of a continuous roughness with a more or less physical interpretation. Apart from the unavoidable sampling induced by the mesh, it should be as mesh independent as possible and in particular should not depend on cell orientations. The single flow direction D8 and Rho8 methods reinterpreted this way introduce a coefficient $k_{m,\sigma}$ that is clearly mesh dependent and not the discretization of a continuous coefficient. This will consequently increase the mesh dependency of the overall method.

The two point flux approximation (TPFA) is of course not the only possible approximation for the terms $(\nabla(h_s + b))_{\sigma_{i\pm1,j\pm1}} \cdot \int_{\sigma_{i\pm1,j\pm1}} \boldsymbol{n}$. In particular, if one reconstructs an approximation $\hat{\boldsymbol{G}}_{s,\sigma}$ of the full topographic gradient along each face $\sigma$, then it can be used to compute an approximation of the flux. We denote it $\hat{\boldsymbol{G}}_{s,\sigma}$ to distinguish it from the reconstruction $\boldsymbol{G}_{s,\sigma}$ used to approximate the non-linear dependency in the slope, as the two can be different. In this case, (14) becomes:

$$\frac{|\sigma_{m,n}|(\Delta\mathcal{H}_{i,j}^{m,n})^q}{\sum\limits_{m',n'\in\mathcal{N}(i,j)} |\sigma_{m',n'}|(\Delta\mathcal{H}_{i,j}^{m',n'})^q} \quad \text{and} \quad \Delta\mathcal{H}_{i,j}^{m,n} = \max\left(0, \hat{\boldsymbol{G}}_{s,\sigma_{m,n}} \cdot \int_{\sigma_{m,n}} \boldsymbol{n}_{K_{i,j}}\right). \tag{16}$$

Then, more flow routing algorithms of the literature can be rewritten this way. In particular, choosing $\gamma_x = \gamma_y = 0$ or 1 we can easily recover the flux decomposition method (Desmet et al. 1996 Desmet and Govers (1996)) and a variation of the MD$\infty$ method (Seibert et al. 2007 Seibert and McGlynn (2007)). The flux decomposition method chooses a single value for $\hat{\boldsymbol{G}}_{s,K_{i,j}}$



for each cell, and then loop over cells and set $\hat{\boldsymbol{G}}_{s,\sigma} = \hat{\boldsymbol{G}}_{s,K_{i,j}}$ for the faces of the current cell that have not already been handled through a previous cell in the loop. The MD$\infty$ methods computes $\hat{\boldsymbol{G}}_{s,\sigma}$ for each face using a triangular reconstruction of the slope: to be precise, with our notations $\hat{\boldsymbol{G}}_{s,\sigma}$ is for face $\sigma_{m,n}$ half the sum of the two triangular gradients computed in Seibert and McGlynn (2007) that can contribute to $\sigma_{m,n}$. We say that this is a variation of Seibert and McGlynn (2007) as it is unclear whether they use the normal component of the gradient as we do here or the full norm of the gradient in their flow sharing formula.

**Table 2.** A classification of some flow routing algorithms using (16)

| Method | $\delta/\Delta_{xy}$ | $\gamma_x = \gamma_y$ | $q$ | $k_{m,\sigma_l}$ |
|---|---|---|---|---|
| Flux decomposition (Desmet et al. 1996 Desmet and Govers (1996)) | 0 | 1 | 1 | 1 |
| MD$\infty$ (Seibert et al. 2007, Seibert and McGlynn (2007)) | $\frac{\sqrt{2}}{2+\sqrt{2}}$ | $\frac{\sqrt{2}}{2+\sqrt{2}}$ | 1 | 1 |


Other flow routing algorithms than do not seem to easily enter this framework are also available in the literature. We mention in particular the ANSWERS (Beasley et al. (1980)), DEMON (Costa-Cabral and Burge (1994)) and Lea's method (Lea (1992)), that are all based on a local planar approximation of the topography and use either a multiple or single direction flow sharing formula based on purely geometric considerations. The D$\infty$ method (Tarboton 1997 Tarboton (1997)) strongly looks like the

SFD method at first sight, however because the flow sharing formula used when the steepest direction is not aligned with mesh direction is based on angular considerations similar to those of ANSWERS and DEMON, it is not immediately obvious how to relate the D$\infty$ method to a continuous model. Finally, let us mention that many variations around the classical algorithms have been explored since their first publications leading for instance to some generalization to triangular meshes Banninger (2007); Zhou et al. (2011). We refer the reader to Erskine et al. (2006); Wilson et al. (2008); Orlandini and Moretti (2009) and

references therein for a broader review on flow routing algorithms and their numerical behavior.

## 4   Large structures simulation for numerical instabilities free landscape evolution models

In this section, after illustrating the numerical problems arising when non-linearly coupling water flow and sediment evolution on an easy to analyze synthetic test case, we explain how to transpose the ideas underlying the concept of large eddy simulation from the computational fluid dynamics community to our landscape evolution model. In our opinion, this is a key ingredient

for achieving reproducible LEM simulations. All the simulations shown in the following sections are performed using the ArcaDES platform (Coatléven (2020)) (although ArcaDES is mentioned for the first time in a scientific paper, it is used since 2015 in the stratigraphic numerical forward model DionisosFlow™ initially developed by Granjeon (1996)).





## 4.1 Model description

At first let us mention that all the observations, conclusions and recommendations coming from this section are not linked to
any specific sediment evolution model and should in principle apply to any coherent sediment model satisfying (H1), (H2) and
(H3). In the present paper we have chosen to focus on the sediment model that has already been discussed in detail in Granjeon
(1996); Eymard et al. (2004, 2005); Peton et al. (2020):

$$\boldsymbol{J}_s = -\eta_s(h_s)s_{ref}^{-p_s}||\nabla(h_s+b)||^{p_s}\left(\left(\frac{q_w}{q_{ref}}\right)^{r_s}\nabla\psi_w(h_s+b)+\nabla\psi_g(h_s+b)\right) \quad \text{in } \Omega\times]t_0,T[, \tag{17}$$

where $r_s > 0$ and $p_s > 0$ are model parameters, $q_w$ is the water flux obtained from (4), $q_{ref}$ and $s_{ref}$ are dimensional factors,
and $\eta_s$ is a dimensionless sediment mobility function such that:

$$0 \leq \eta_s(u) \leq 1 \quad \text{and} \quad \eta_s(0) = 0, \tag{18}$$

whose main role is to ensure that the sediment height $h_s$ remains positive. In the following we use:

$$\eta_s(u) = \begin{vmatrix} 1 - \dfrac{h_*}{u + h_*} & \text{if } u \geq 0, \\ \\ 0 & \text{otherwise} \end{vmatrix} \tag{19}$$

with $h_*$=1 cm. We consider here the most common form for functions $\psi_w$ and $\psi_g$ corresponding to:

$$\psi_w(u) = \int_0^u k_w(v)dv \quad \text{and} \quad \psi_g(u) = \int_0^u k_g(v)dv, \tag{20}$$

where $k_w$ and $k_g$ are diffusion coefficients such that:

$$0 \leq k_g^- \leq k_g(u) \leq k_g^+ < +\infty \quad \text{and} \quad 0 \leq k_w^- \leq k_w(u) \leq k_w^+ < +\infty, \tag{21}$$

in such a way that:

$$\nabla\psi_w(h_s+b) = k_w(h_s+b)\nabla(h_s+b) \quad \text{and} \quad \nabla\psi_g(h_s+b) = k_g(h_s+b)\nabla(h_s+b), \tag{22}$$

so that the sediment flux follows the topographic slope $\nabla(h_s+b)$. This sediment flux model is implemented in our modeling
platform ArcaDES (Coatléven (2020)) (although ArcaDES is mentioned for the first time in a scientific paper, it is used since
2015 in the stratigraphic numerical forward model DionisosFlow™ initially developed by Granjeon (1996)). Both soil erosion
and sediment deposition are considered. As ArcaDES is tailored for large time and space scales simulations, this is the reason
why we have chosen to express the xy coordinates in kilometers (km), time in million years (My), sediment height $h_s$ and
basement $b$ in meters (m). Thus the unit of sediment sources will be meters per million years (m.My⁻¹). Since we have chosen
to use $\mathcal{Q}_w = q_w$ with $q_w$ the water flux from (4), the unit for the water discharge $q_w$ is m³.s⁻¹.km⁻¹ and thus we naturally set
$q_{ref}$= 1 m³.s⁻¹.km⁻¹. The natural unit of coefficients $k_g$ and $k_w$ is km².My⁻¹, with the reference slope again set to $s_{ref}$= 1
m.km⁻¹.



## 4.2 Numerical issues with non linear coupling of overland flow and sediment erosion and transport

From Smith and Bretherton (1972), we know that for sublinear to linear coupling, i.e. $r_s \leq 1$ no chaotic behavior is expected, as is confirmed by numerical experiments. However as soon as $r_s > 1$ a self-amplification mechanism is expected leading to highly non linear behaviors and complex topographies. This is precisely the domain we explore in this section.

As mentioned in the introduction, in the absence of reference analytic solution it is in general hard to decide whether a numerical solution of (2) is correct or not. To partially circumvent this difficulty, we consider a simple synthetic topographic surface

defined by three constant slope planes. The numerical domain is rectangular with the dimensions $Lx = 400$ km in the x axis and $Ly = 300$ km in the y axis (see Fig. 5-a,5-b). The mesh size is $\Delta_{xy} = 2$ km. The gravity diffusion coefficient $k_g$ is equal to 100 km$^2$.My$^{-1}$ in the whole domain while $k_w = 10$ km$^2$.My$^{-1}$ for $h_s + b \geq 0$ and $k_w = 0.1$ km$^2$.My$^{-1}$ for $h_s + b < 0$, corresponding to a modulation of the water induced transport in a fictitious marine domain. Water is supplied by three constant water-flux sources located at the domain boundary (black arrows in Fig. 5-a), so we call this "three rivers" test case. Each water source is 12 km large and supplies 1200 m$^3$s$^{-1}$ of water.

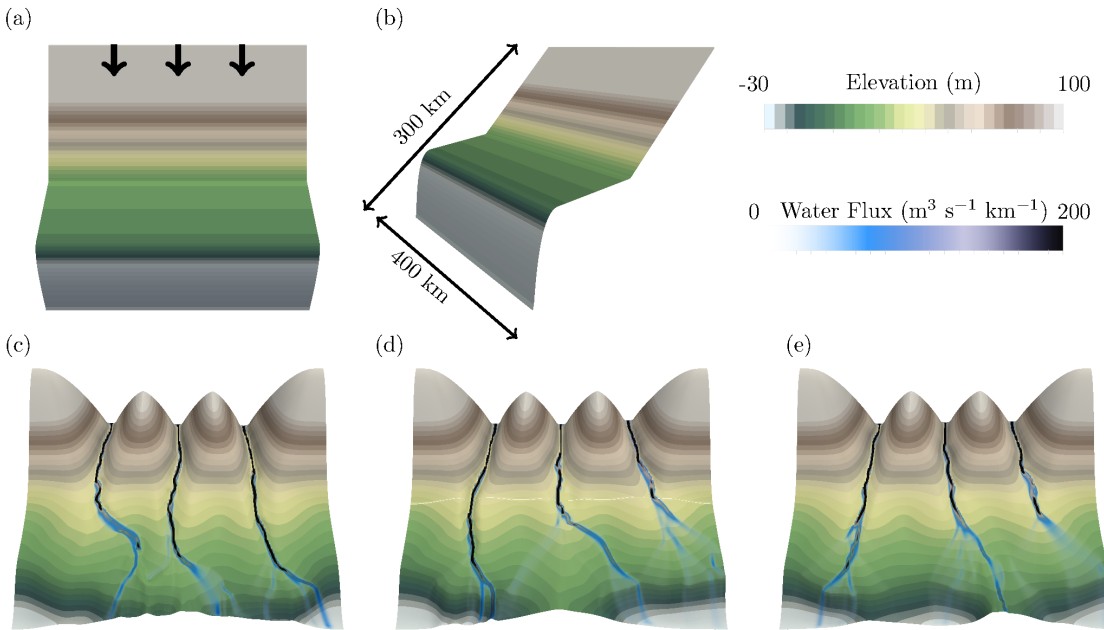

**Figure 5.** The "three rivers" test case with $\Delta_{xy}$=2 km. a-b : Initial topography, black arrows represent the position of the water inflows. Bottom row : topography and water flux after 6 My obtained under different numerical settings. c: sequential GMRES, d: parallel GMRES, e: sequential BiCGStab


An essential remark is that the whole configuration is symmetrical with respect to the vertical plane $x = Lx/2$. In principle, the equation system consisting of (2) and (4), here used with $r_s = 2$, $p_s = 1$, $p_w$=0 and $k_m$=1 m.s$^{-1}$ should maintain this



symmetry. Therefore symmetry will be our main tool to evaluate solution quality. Using the finite volume scheme depicted in section A which for water flow corresponds to using the consistent water flux obtained from (9), we perform a set of three

identical simulations in terms of physical parameters but using different numerical settings in order to illustrate the impacts of numerical errors. We perform a sequential computation using GMRES as linear solver for all systems, its parallel equivalent on 4 processors and another sequential simulation using BiCGStab as linear solver for all systems. The linear solvers are part of the well-known and reference PETSc library (Balay et al. (1998)) to avoid any potential mistake in their implementation, while the parallelism relies on the Arcane framework (Grospellier and Lelandais (2009)). Final topographies and water flux

are shown on the bottom row of Fig. 5. Figure 5-c corresponds to sequential GMRES, Fig. 5-d to parallel GMRES and Fig. 5-e to sequential BiCGStab.

All the results from these simulations should be almost identical and in any case symmetrical with respect to the vertical plane $x = Lx/2$ in absence of any spatial heterogeneity in the input data. Clearly, symmetry is lost in the three cases and what is even more striking is that we get three very different results. The only difference between the three cases being the numerical

solvers, this indicates that this has originated from numerical errors. As we are using a decoupled time scheme between water flow and sediment evolution (see section A), one may argue that those instabilities are arising from some violated coupling constraint on the time step. Should this be the case, reducing the time step enough would ultimately lead to clean solutions. However, we have observed the exact opposite: the smaller the time step is, the larger are the obtained instabilities. The fact that reducing the time step makes things even worse is thus another clear sign that our problems are the result of amplified error

accumulation. Finally if the same experiments are performed with $r_s \leq 1$ then this time the symmetry is maintained and all three solutions are almost identical however small the time step might be, which clearly indicates that the non-linearity of the coupling for $r_s > 1$ is responsible of the observed chaotic behavior. Our interpretation is that small numerical perturbations are rapidly amplified by the model up to the point that they become of the same order of magnitude that the originally dominant part of the solution, and do influence flow branching. From the modeling perspective the model behaves as expected in the

sense that small perturbations are amplified and strongly impact the final result. However in our simulations no heterogeneity is present in data thus this phenomenon should not spontaneously occur: the non-physical numerical errors are amplified up to the point that the numerical solution is no longer a reasonable approximation. This clearly also implies that the numerical schemes must be as precise as possible to reduce the numerical noise. In particular it is mandatory to use our consistent MFD discretization of (4) rather than the non consistent flow routing algorithms of the literature.

**4.3    Principles and physical interpretation of filtering**

Recall that the main idea of LES is to filter the solution to distinguish between the behavior of the flow above and below the target length scale, to obtain local averages that are smoother and as mesh independent as possible. This target length scale controls the size of the smallest structures that we will be able to resolve in the problem, quite independently of the domain size. The main practical consequence is that our mesh will have to resolve this length scale, i.e. the mesh size $\varepsilon$ will have to be

smaller than the chosen length scale.



LES filters/models are probably as numerous as the various authors working on the subject (Berselli et al. (2005)), thus we will very brief on the subject and refer the reader to a quite recent review Zhiyin (2015). The very first LES model is called the Leray-$\alpha$ model. It was used by Leray in 1934 to establish existence of weak solutions to the Navier-Stokes equations (Leray (1934)). Originally, the filtering in Leray (1934) as well as in many classical LES models was achieved by using a convolution
operator $\mathcal{F}$ defined by:

$$\mathcal{F}(u)(\boldsymbol{x}) = \int_{\mathbb{R}^d} u(\boldsymbol{y}) g_\delta(\boldsymbol{x} - \boldsymbol{y}) d\boldsymbol{y}, \quad \text{where} \quad g_\delta(\boldsymbol{x}) = \frac{1}{\delta^d} g\left(\frac{\boldsymbol{x}}{\delta}\right),$$

where the filter kernel $g$ satisfies:

$$0 \leq g(\boldsymbol{x}) \leq 1, \qquad g(\boldsymbol{0}) = 1, \qquad \int_{\mathbb{R}^d} g(\boldsymbol{x}) d\boldsymbol{x} = 1.$$

Several kernels are used in the literature, such as a low-pass filter, a box-filter or the very natural Gaussian filter $g(\boldsymbol{x}) = \pi^{-d/2} e^{-|\boldsymbol{x}|^2}$.

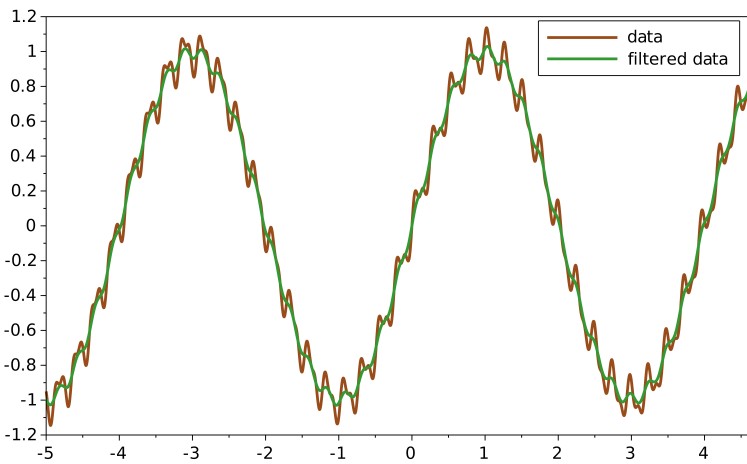

**Figure 6.** Illustration of the effect of the convolution by a Gaussian function


In figure 6 we illustrate the smoothing effect of a Gaussian kernel on an oscillating data: as expected, it preserves the high amplitude and low frequency oscillation while filtering out the high frequency and low amplitude oscillations. Such filters might therefore be ideal for our application to landscape evolution models: the small topographic perturbations will be cleaned out such that the flow routing will not be affected by it. Although convolution operators produce averages with the desired
properties, they are impractical on bounded domains. The modern way of defining the Leray-$\alpha$ filter for bounded domains





consists in using the differential filter $\mathcal{F}_\alpha$ defined by (Cheskidov et al. (2005); Guermond et al. (2003)):

$$
\left|
\begin{array}{ll}
-\alpha^2 \Delta \mathcal{F}_\alpha(u) + \mathcal{F}_\alpha(u) = u & \text{in } \Omega, \\[2mm]
\nabla \mathcal{F}_\alpha(u) \cdot \boldsymbol{n} = 0 & \text{on } \partial\Omega_\mathcal{N}, \\[2mm]
\mathcal{F}_\alpha(u) = 0 & \text{on } \partial\Omega_\mathcal{D}.
\end{array}
\right.
\tag{23}
$$

The filtered result $\mathcal{F}_\alpha(u)$ basically amounts to a convolution of $u$ by the underlying Green's function (23), i.e. the filter applied to the Dirac distribution. Using a finite volume scheme $\mathcal{F}_\alpha$ we can this time easily obtain a discrete version $\mathcal{F}_{\alpha,h}$ which is one

of the main reasons why we have chosen to use this filter, along with its theoretical and practical success for CFD. Notice that contrary to Cheskidov et al. (2005); Guermond et al. (2003), we use homogeneous Neumann and Dirichlet boundary conditions instead of periodic boundary conditions to simplify the treatment of the boundary. The main drawback of this choice is that our filter does not commute with differential operators. Resorting to only Dirichlet boundary conditions would have solved this issue, however from our numerical experiments we found that this can create boundary effects unless the chosen Dirichlet

boundary condition is adapted to the filtered quantity. The Neumann choice avoids those difficulties without creating any practical issues, which has motivated our choice. For quantities such as the water flux for which Neumann everywhere is a more natural boundary condition, we introduce the alternative filter $\mathcal{F}_\alpha^\mathcal{N}$ with only Neumann boundary conditions:

$$
\left|
\begin{array}{ll}
-\alpha^2 \Delta \mathcal{F}_\alpha^\mathcal{N}(u) + \mathcal{F}_\alpha^\mathcal{N}(u) = u & \text{in } \Omega, \\[2mm]
\nabla \mathcal{F}_\alpha^\mathcal{N}(u) \cdot \boldsymbol{n} = 0 & \text{on } \partial\Omega.
\end{array}
\right.
\tag{24}
$$

### 4.4 Leray filtering applied to our landscape evolution model

From the numerical observations that the model governing the simultaneous evolution of sediment and water seems as intractable to solution as the Navier-Stokes system is, following the idea of LES we abandon the idea of resolving all the scales involved in the landscape evolution problem and will only try to simulate the large sedimentary and water structures. In practice, this means that the sediment flux used in the mass conservation equations:

$$
\left|
\begin{array}{ll}
\dfrac{\partial h_s}{\partial t} + div\,(\boldsymbol{J}_s) = S_s & \text{in } \Omega \times ]t_0, T[, \\[3mm]
-\boldsymbol{J}_s \cdot \boldsymbol{n} = B_s & \text{on } \partial\Omega_\mathcal{N} \times ]t_0, T[, \\[3mm]
h_s = 0 & \text{on } \partial\Omega_\mathcal{D} \times ]t_0, T[, \\[3mm]
h_s(t = t_0) = h_{s,0} & \text{in } \Omega,
\end{array}
\right.
$$

will now be given by:

$$
\boldsymbol{J}_s = -\eta_s(h_s) s_{ref}^{-p_s} ||\nabla(h_s + b)||^{p_s} \left( \left( \frac{\mathcal{F}_\alpha^\mathcal{N}(q_w)}{q_{ref}} \right)^{r_s} \nabla\psi_w(h_s + b) + \nabla\psi_g(h_s + b) \right) \quad \text{in } \Omega \times ]t_0, T[,
\tag{25}
$$



where we use the filtered water flux magnitude $\mathcal{F}_\alpha^{\mathcal{N}}(q_w)$ instead of directly using the water flux $q_w$. In the same way, in the water equations, we will now use the filtered topography $\mathcal{F}_\alpha(h_s + b)$ instead of the topography $h_s + b$, leading to:

$$\left| \begin{array}{ll} -div\left(k_m h_w \eta_w(h_w) s_{ref}^{-p_w} ||\nabla(\mathcal{F}_\alpha(h_s + b))||^{p_w} \nabla(\mathcal{F}_\alpha(h_s + b))\right) = S_w & \text{in } \Omega, \\ -k_m h_w \eta_w(h_w) s_{ref}^{-p_w} ||\nabla(\mathcal{F}_\alpha(h_s + b))||^{p_w} \nabla(\mathcal{F}_\alpha(h_s + b)) \cdot n = B_w & \text{on } \partial\Omega, \end{array} \right. \tag{26}$$

with the associated water flux:

$$q_w = ||k_m h_w \eta_w(h_w) s_{ref}^{-p_w} ||\nabla(\mathcal{F}_\alpha(h_s + b))||^{p_w} \nabla(\mathcal{F}_\alpha(h_s + b))||. \tag{27}$$

Our "reproducible" large structures simulation for landscape evolution thus consists in solving (2)-(25)-(26)-(27).

### 4.5   Numerical results with filtering

We reproduce the very same experiment that was performed at the beginning of this section on the "three rivers" test case, with
sequential GMRES, parallel GMRES and sequential BiCGStab, but using a filter $\alpha = 2.2$ km. Contrary to Fig. 5, the symmetry is maintained and we obtain almost identical results for the three configurations 7. The expected impact of the filter on the simulated water flow and topography is a smoothing effect, which is what is observed when comparing for example the width of the three valleys. However, the differences remain marginal in this case.

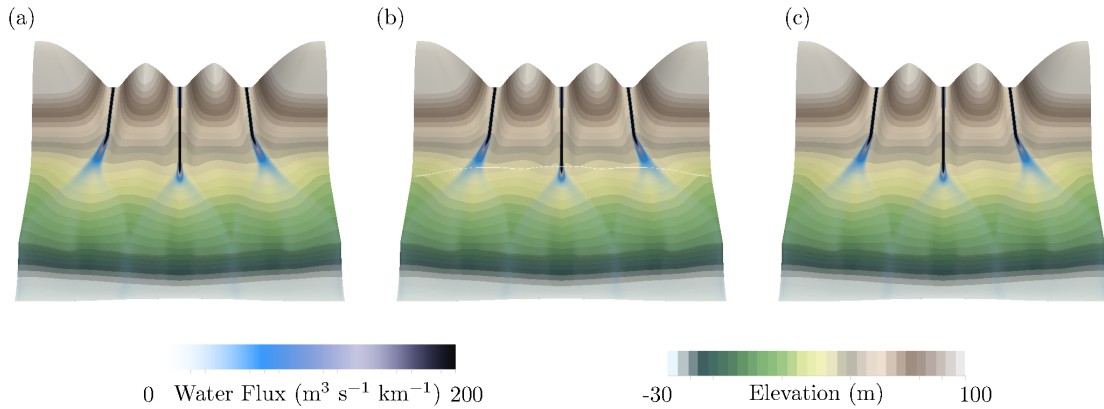

**Figure 7.** The "three rivers" test case with filter $\alpha$= 2.2 km and $\Delta_{xy}$=2 km. Topography and water flux after 6 My. a: sequential GMRES, b: parallel GMRES, c: sequential BiCGStab

Following LES principles, the filter scale $\alpha$ corresponds to the spatial resolution of our continuous model, which must naturally be resolved by the grid resolution, meaning we should have at the very least $\Delta_{xy} < \alpha$ for cartesian grids (and more generally $\epsilon < \alpha$ for a general mesh recalling that $\epsilon = \sqrt{2}\Delta_{xy}$ for cartesian meshes). To assess the legitimacy of this condition, still on our "three rivers" test case we first fix the grid size to $\Delta_{xy}$=2 km and observe the behavior of the solution for various values of





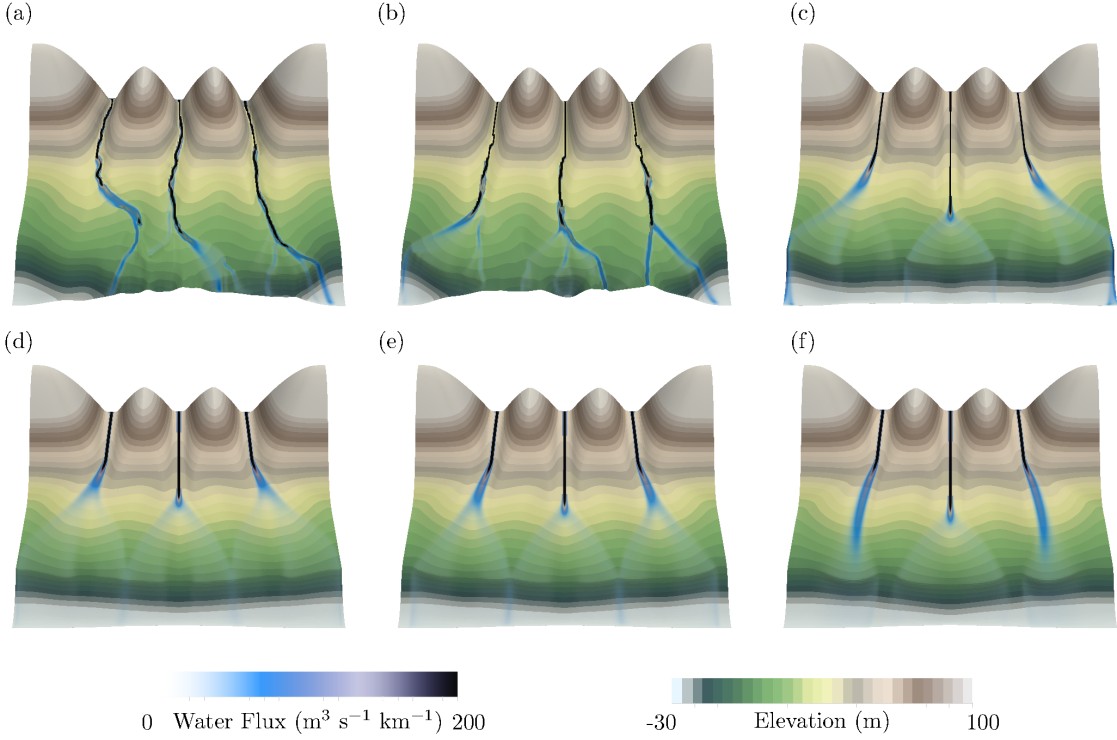

$$0 \quad \text{Water Flux (m}^3 \text{ s}^{-1} \text{ km}^{-1}) \quad 200 \qquad -30 \quad \text{Elevation (m)} \quad 100$$

**Figure 8.** The "three rivers" test case with $\Delta_{xy}$=2 km. Final topography and water flux after 6 My obtained with different values of the filter parameter $\alpha$. a: no filter, b: $\alpha$= 0.2 km, c: $\alpha$=1 km, d: $\alpha$= 2.2 km, e: $\alpha$=2.5 km , f: $\alpha$= 3 km

the filter parameter $\alpha$. Results are displayed Fig. 8. We clearly see that symmetric solutions are obtained for $\alpha \geq \Delta_{xy}$, while
further reducing the filter parameter leads to behavior similar to the no filter case. This is coherent with the LES that the filter should control what happens below the grid scale, which can only be done if $\alpha > \Delta_{xy}$.

### 4.6 Impacts of water flow consistency and filtering on the emergence of geomorphic structures

The three rivers synthetic case has highlighted the absolute necessity of considering a filtering strategy in LEMs using a
MFD/SFD water flow algorithm. We now switch on a second synthetic case study to observe the formation of geomorphic features, using either a homogeneous or a perturbed initial topography. The characteristics of this test case are relatively close to the models published by Perron et al. (2008); Armitage (2019). The numerical domain corresponds to a rectangular grid with the dimensions $Lx = 600$ km in the x axis and $Ly = 80$ km is the y axis containing a mesh of resolution $\Delta_{xy} = 0.25$ km. The basement is $b$ is constant equal to $0$ m, while the sediment thickness $h_s$ is initially given by a uniform in x smooth bump:





$$g(x,y) = \left| \begin{array}{ll} H \exp\left(\dfrac{-1}{1-r_y^2}\right) & \text{for } r_y = \dfrac{(y-y_c)}{\delta_y} \leq 1, \\ 0 & \text{otherwise}, \end{array} \right.$$

with $H = 20$m , $y_c = 40$ km and $\delta_y = 20$ km. This symmetry in the x direction of the initial topography is then perturbed by a $N_b$ small smooth bumps randomly positioned at points $(x_p, y_p)$:

$$g_{pert}(x,y) = \left| \begin{array}{ll} H_{pert} \exp\left(\dfrac{-1}{1-r^2}\right) & \text{for } r^2 = \dfrac{(x-x_p)^2}{\delta^2} + \dfrac{(y-y_p)^2}{\delta^2} \leq 1, \\ 0 & \text{otherwise}, \end{array} \right.$$

with $H_{pert} = 1$ m and $\delta = 2$ km. Rain-fall is constant in time and space (3000 mm/y) and is the unique water supply for this
case. The sediment source (here we simulate a sediment production) goes from $S_s = 0$ m.My$^{-1}$ at $y = 0$ and $y = Ly$ sides to $S_s = 100$ m.My$^{-1}$ at $y = Ly/2 = y_c$. The variation is continue over the whole domain following :

$$S_s(x,y) = \left| \begin{array}{ll} S_{max} \exp\left(\dfrac{-1}{1-r_y^2}\right) & \text{for } r_y = \dfrac{(y-y_c)}{\delta_y} \leq 1 \\ 0 & \text{otherwise} \end{array} \right.$$

with $\delta_y = 40$ km. Model boundary conditions are fixed elevation on the sides normal to the $x$ axis and zero gradient on the sides normal to the $y$ axis. Models parameters controlling the non-linearity in the water-sediment coupling are set as $r_s = 2$,
$p_s = 0$, $p_w = 0$ and $k_m = 1$ m.s$^{-1}$. Simulation takes place over the time period $T = 6$ My, using the numerical schemes detailed in section A.

The first simulation use constant diffusive coefficients $k_g = 50$ km$^2$.My$^{-1}$ and $k_w = 5$ km$^2$.My$^{-1}$, and the initial topography is built with $N_b = 30$. In order to analyze the results of this simulation and the following ones, it is also important to discuss the implications of the values of the diffusion coefficients. At this point, we want to stress the fact that those specific values
$k_g = 50$ km$^2$.My$^{-1}$ and $k_w = 5$ km$^2$.My$^{-1}$ have been chosen on purpose, such that our non-linear diffusive model should be able to diffuse quickly small initial perturbations such as the ones we introduce and thus loose memory of its initial state as would a classical linear diffusive model. Of course the key parameter is the dimensionless ratio $\tau = (k_w q_w^{r_s})/(k_g q_{ref}^{T_s})$ which plays an essential role here, small values implying a gravity dominated case while larger values correspond to a water dominated case. The ratio $\tau$ is clearly reminiscent of the Reynolds number for turbulent flows, with true turbulence appearing with large
Reynolds numbers. It is our belief, although we do not have any formal proof at this stage, that one can anticipate the "chaotic" or "non chaotic" behavior of the solution by considering the values of $\tau$. For this test case, in the part of the domain where the slope is significant, the ratio $\tau$ is below 10. We expect this case to be close to a quite gravity dominated one, but not too much such that the potential problems linked to numerical errors are not completely dissipated by the gravity diffusion. With that in mind, we could expect no specific amplification of the initial bumps in the topography. In order to emphasize the impact of the
consistency and the effect of filtering on the emergence of complex geomorphic features, we have first performed a simulation mimicking those of the literature using the non-consistent version of the water flux without filtering, i.e. we momentarily chose





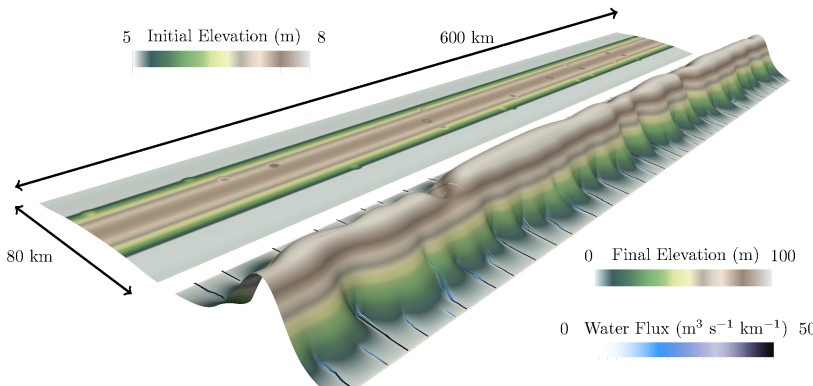

**Figure 9.** Initial and final topographies for model (2)-(17) with $\mathcal{Q}_w$ computed from Freeman (1989) and Pelletier (2013), with $r_s = 2, p_s = 0$. Dimensions of the rectangular domain are $Lx = 600$ km in the x axis and $Ly = 80$ km. Gradient color shows topographic variation, except the blue color blue color that corresponds to the water flowing over the topographic surface. The large spatial-scale of the initial topography is homogeneous in the x direction, but small spatial scales and dominated by 30 small bumps (radius $\delta = 2$ km and height $H_{pert} = 1$ m. Final elevation is not homogeneous in the x direction, suggesting a dominant effect of the bumps on the results.

for $\mathcal{Q}_w$ to use the non consistent $\widetilde{q}_K$ of the MFD literature chosen from Freeman Freeman (1989) and updated by Pelletier (2013)). However, final elevation is clearly not homogeneous in the x direction, suggesting a dominant effect of the bumps on the results. Figure 9 shows that in this non consistent, unfiltered setting the obtained elevation is quite complex and the final

heterogeneity do not seem necessary located at the same $x_p$ positions. In fact, all the domain seems to have been impacted by the small perturbations in the initial topography. Water flow distribution (represented by the dark blue color in figure 9) is organized, suggesting the emergence of characteristic length-scale that controls the valley spacing. The illustrates the issue describe din the introduction: without any reference it is very hard to decide wether this result is the correct physcial one or not.


As a first step towards a clear answer to this, we perform a set of four simulations using an initial homogeneous in the x-direction topography ($N_b = 0$), still with the same diffusive coefficients. We respectively display in Fig. 10 the initial topography without any perturbation and the final result obtain without any filter, for $\Delta_{xy}$=1 km and $\Delta_{xy}$=0.25 km, for the consistent and non consistent MFD. In all cases, the final topography remains uniform in the large direction as expected, and the result is clean

of any perturbations for both mesh sizes, for both choices of water flux. This series of run indicate here that under the right circumstances (probably linked to $\tau$) the corrections such as the one of Pelletier (2013) can lead to the false impression that they do correct $\widetilde{q}_K$ in the right way.

Simulations shown Fig. 11 are similar to those in Fig. 10 but the initial topography contains a single perturbation ($N_b = 1$). We display the initial topography with a single perturbation (one bump), as well as the final topography obtained using again the





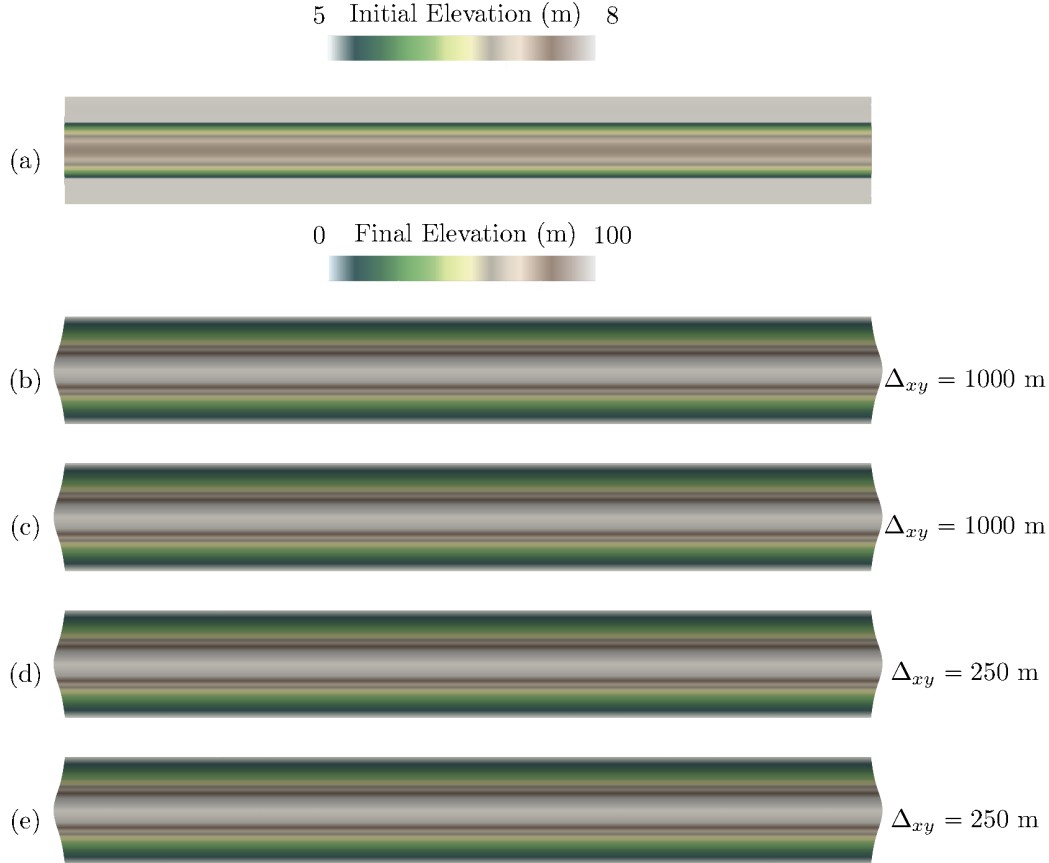

**Figure 10.** a: initial unperturbed topography, b: final state for consistent MFD and $\Delta_{xy} = 1$ km, c: final state for non consistent MFD and $\Delta_{xy} = 1$ km, d: final state for consistent MFD and $\Delta_{xy} = 0.25$ km, e: final state for non consistent MFD and $\Delta_{xy} = 0.25$ km

consistent and non-consistent MFD, both without filter and again for $\Delta_{xy}$=1 km and $\Delta_{xy}$ =0.25 km. Looking at the results for $\Delta_{xy} = 1$ km, we see that the consistent version leads to the same uniform final topography than in the unperturbed situation. On the contrary, the non-consistent MFD introduces a non negligible error in the final result: we see here how the non-consistent MFD of the literature can clearly introduce numerical artifacts. However when we look at the results for $\Delta_{xy}$= 0.25 km, both schemes produce finals topographies with large perturbations induced by the initial bump, but with much wider ones for the non-consistent case as should be expected. At this point it is hard to decide if the solution 11-D of the consistent MFD with $\Delta_{xy}$ =0.25 km is the correct approximation of the solution of (2)-(17)-(4), implying in this case that the previous mesh size $\Delta_{xy}$= 1 km was too coarse, or if what we see are again numerical artifacts. Comparing with the results obtained combining the consistent MFD with filters, which are presented for the case with a single perturbation in Fig. 13-b for $(\alpha, \Delta_{xy})$= (1.2 km, 1 km) and 13-c for $(\alpha, \Delta_{xy})$= (0.3 km, 0.25 km), we see that the consistent plus filter version always leads to the uniform final topography for both mesh sizes. Of course, only one of the two consistent results (with or without filters) can be the correct





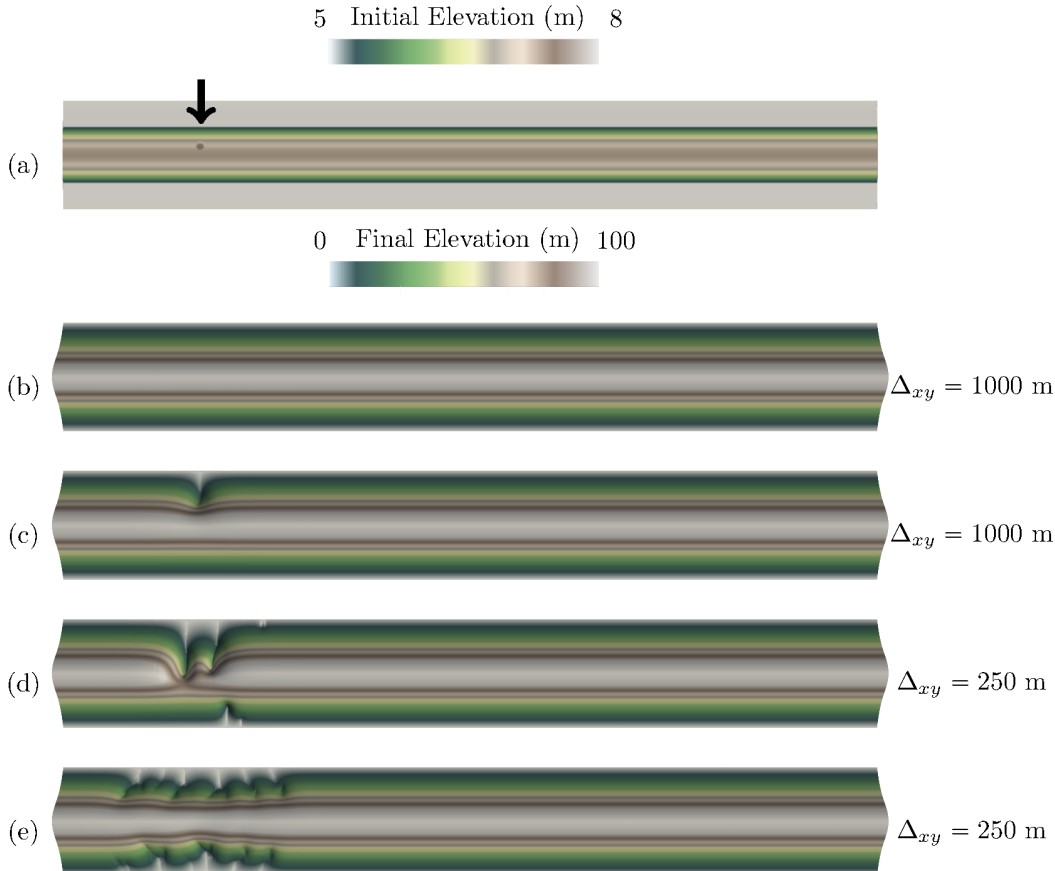

**Figure 11.** a: initial topography with a single perturbation, b: final state for consistent MFD and $\Delta_{xy}$ = 1 km, c: final state for non consistent MFD and $\Delta_{xy}$ = 1 km, d: final state for consistent MFD and $\Delta_{xy}$ = 0.25 km, e: final state for non consistent MFD and $\Delta_{xy}$ = 0.25 km

one. At the very least this first comparison emphasizes why the non-consistent MFD should no longer be used: indeed, it should now be obvious that the complexity observed in Fig. 9 was undoubtedly the mainly the product of amplified numerical errors. However as the values $\Delta_{xy}$=0.25 km and $\alpha$=0.3 km have been chosen small enough to resolve the small initial bumps or the "width" of the water flow appearing around the bumps, we are confident that the consistent plus filter uniform solution

(Fig 13-c) that used those parameters is the correct approximation of the solution of (2)-(17)-(4), implying that the consistent without filter solution (Fig. 11-d) was erroneous. Another strong argument in this sense is that noise appears in the unfiltered consistent case when refining the mesh, with even more noise for more refined meshes: our interpretation is that numerical diffusion, which is much smaller that the true physical diffusion in view of the values of $k_g$ adds nevertheless enough additional smoothing for $\Delta_{xy}$ = 1 km to dissipate large parts of the numerical errors while this is no longer the case for the finer mesh

$\Delta_{xy}$ = 0.3 km.



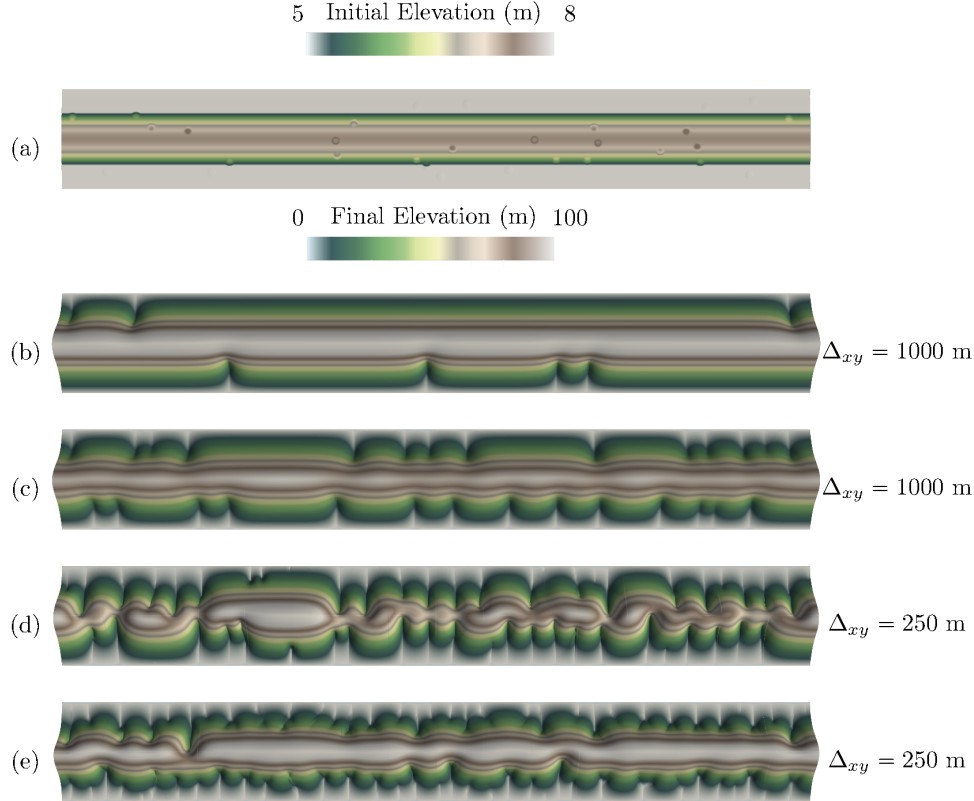

**Figure 12.** a: initial perturbed topography, b: final state for consistent MFD and $\Delta_{xy} = 1$ km, c: final state for non consistent MFD and $\Delta_{xy}$ = 1 km, d: final state for consistent MFD and $\Delta_{xy} = 0.25$ km, e: final state for non consistent MFD and $\Delta_{xy} = 0.25$ km

We finally perform the very same experiments with the fully perturbed initial topography (30 bumps). The results for the un-filtered non consistent and consistent MFD are presented in Fig. 12, while the consistent plus filter results are displayed in Fig. 13. It is obvious that the same conclusions apply to this more complex case: considering the results for $\Delta_{xy} = 1$ km, we recover the fact that using a consistent scheme improves the results, but this time not even enough to keep numerical errors

under control. However we see that in the consistent case, not all bumps correspond to a final deformation. This is another clear sign that something is wrong with this solution, even if the errors remain relatively small. For $\Delta_{xy} = 0.25$ km we obtain very complex topographies which may appear as realistic but are in fact clearly solutions blurred by numerical noise. This affirmation is enhanced when look at the clean uniform final state obtain with filters which is again the correct approximation of the solution of (2)-(17)-(4).

As we have already explained, the values of $k_g$ and $k_w$ were purposely chosen in the above experiments to lead to this treacher-ous situation where the correct solutions are clean and uniform despite of the initial perturbations. However, this does not mean that solutions obtained using filters will never develop complex topographies, but that they can only contain heterogeneity that



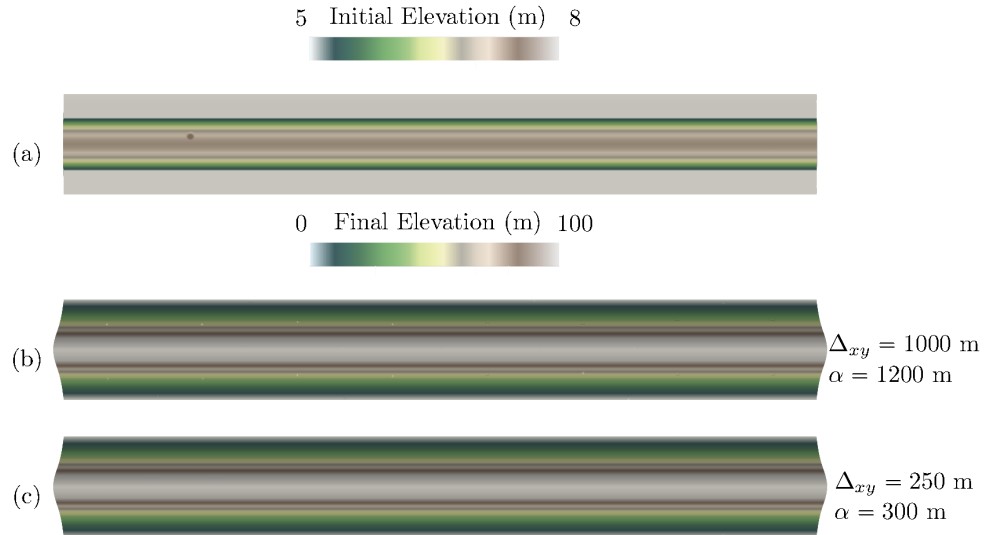

**Figure 13.** a: initial topography with a single perturbation, b: final state for with consistent MFD with filter, $\alpha$= 1200 m and $\Delta_{xy}$ = 1 km, c: final state with consistent MFD with filter, $\alpha$= 0.3 km and $\Delta_{xy}$ = 0.25 km

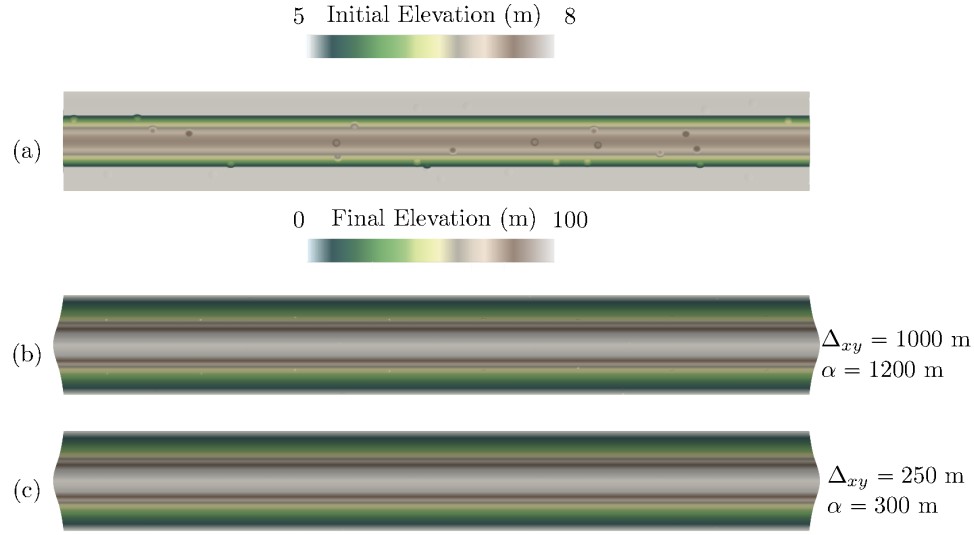

**Figure 14.** a: initial perturbed topography, b: final state with consistent MFD with filter, $\alpha$= 1200 m and $\Delta_{xy}$ = 1 km, f: final state with consistent MFD with filter, $\alpha$= 0.3 km and $\Delta_{xy}$ = 0.25 km

is not the product of amplified numerical errors. To illustrate this, we consider again the perturbed test case but this time using $k_g$= 5 km$^2$.My$^{-1}$, then $k_g$= 1 km$^2$.My$^{-1}$ (Fig. 15), for which we easily get $\tau >> 10$ in the key parts of the domain and we expect



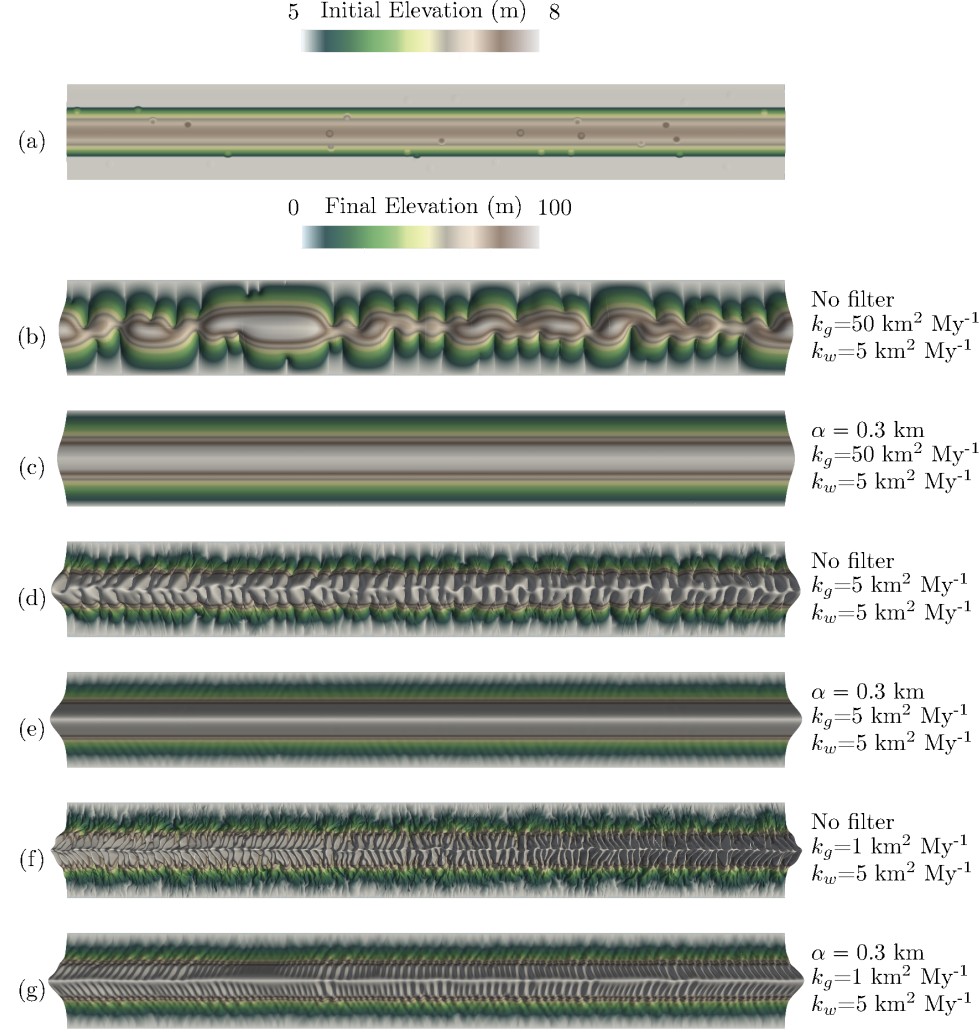

**Figure 15.** Final topographies obtained for three different set of diffusive coefficients, systematically tested without filter and with a filter using $\alpha = 0.3$ km. a: initial perturbed topography. b: solution without filter for $(k_g, k_w)$=(50,5) km$^2$.My$^{-1}$, c: solution with filter for $(k_g, k_w)$=(50,5) km$^2$.My$^{-1}$, d: solution without filter for $(k_g, k_w)$=(5,5) km$^2$.My$^{-1}$, e: solution with filter for $(k_g, k_w)$=(5,5) km$^2$.My$^{-1}$, f: solution without filter for $(k_g, k_w)$=(1,5) km$^2$.My$^{-1}$, g: solution with filter for $(k_g, k_w)$=(1,5) km$^2$.My$^{-1}$

the system to start producing more complex topographies. We only consider the situations with $\alpha$ =0.3 km and $\Delta_{xy}$=0.25 km, again to ensure that at least some of the small scales of the model are correctly resolved by both the filter and the mesh. Of course, for such large values of $\tau$ what we obtain is potentially an averaged version at the filter scale of the correct continuous solution of (2)-(17)-(4). We present in Fig. 15 results with and without filters for the consistent scheme, recovering the fact that the numerical solution without filters is blurred by noise. In the intermediate case $k_g$= 5 km$^2$.My$^{-1}$, we start to see some small





topographic perturbations, while we also see that for the case with $k_g$= 1 km$^2$.My$^{-1}$ and thus the largest value for $\tau$, complex

structures finally develop, assessing the fact that the model with filter is perfectly capable of producing complex topographies

when this corresponds to the correct averaged solution. Yet we insist on the fact that recovering all the details of the correct

continuous solution of (2)-(17)-(4) requires $\alpha$ to correctly resolve all the scales of the model, otherwise what we obtain is a

numerical approximation of the averaged solution at the filter scale. For instance if we use a filter with parameter $\alpha$=1.2 km

as previously done the final topographies for $k_g$= 5 km$^2$.My$^{-1}$ and $k_g$= 1 km$^2$.My$^{-1}$ are indeed much more smooth that those

presented in Fig. 15.

## 5   Discussion

This work belongs to the common effort of the scientific community to harmonize landscape evolution models. The imple-

mentations of the consistent water flux and the large structure simulation strategy should be accessible to every LEMs, and

in particular for the models of Perron et al. (2009); Hooshyar and Porporato (2021a); Porporato (2022) that takes the general

form:

$$
\left|
\begin{array}{ll}
\dfrac{\partial h_s}{\partial t} + div\,(\boldsymbol{J}_s) = S_s & \text{in } \Omega\times]t_0,T[, \\[2mm]
-\boldsymbol{J}_s \cdot \boldsymbol{n} = B_s & \text{on } \partial\Omega_{\mathcal{N}}\times]t_0,T[, \\[2mm]
h_s = 0 & \text{on } \partial\Omega_{\mathcal{D}}\times]t_0,T[, \\[2mm]
h_s(t=t_0) = h_{s,0} & \text{in } \Omega,
\end{array}
\right.
\tag{28}
$$

with a source given by

$$
S_s = U - \kappa_w s_{ref}^{-p_{s,2}}\left(\frac{q_w}{q_{ref}}\right)^{r_s}||\nabla(h_s+b)||^{p_{s,2}},
$$

with $U$ a sediment source term (or an uplift depending on the interpretation of $b$) and a sediment flux given by

$$
\boldsymbol{J}_s = -s_{ref}^{-p_s}k_g||\nabla(h_s+b)||^{p_s}\nabla(h_s+b) \quad \text{in } \Omega\times]t_0,T[.
$$

Those models are relatively close to model (2)-(17) that we have studied in detail here, with the main difference that the

non-linear term $q_w^{r_s}||\nabla(h_s+b)|^{p_s}$ appears as a reaction term rather than in a diffusive term. An immediate application of the

LSS in this context consists of course in replacing $q_w$ by its filtered version $\mathcal{F}^{\mathcal{N}}(q_w)$ in the second member of (28). We also

believe that the $\xi$-q model of Davy and Lague (2009) could benefit from a similar filtering strategy. Notice nevertheless that

the last test cases displayed in Fig. 15 emphasize the fact that correctly using filters requires some understanding of the scales

involved in the model. Although this is not a such easy task in general, we believe that it is very likely that to get an idea of

those scales one can use $\tau$ in the same way than the viscosity and more generally Reynolds number can be used to anticipate

the flow scales. Nevertheless, we can give some generic guidelines that should apply in any situation: first, at the very least the

constraint ont the mesh size $\varepsilon < \alpha$ must be fulfilled to allow the filter to correctly clean the sub-cell scale phenomenons. Next,





the chosen filtering parameter $\alpha$ should resolve the main sediment structures that one wants to correctly represent in the flow, ideally fulfilling an equivalent of Nyquist's rule. For instance if an essential valley is 1 km large, then $\alpha$ should be several times smaller (and ideally smaller than 100 m). A good practical test consists in comparing the filtered topography $\mathcal{F}(h_s + b)$ and the unfiltered one $h_s + b$. The structures of $h_s + b$ that one wants to simulate accuratelly should be preserved in $\mathcal{F}(h_s + b)$, of course

in a smoother way. For instance, for a given value of $\alpha$ if a small topographic depression in which water could in principle flow is observed on $h_s + b$ but is absent in $\mathcal{F}(h_s + b)$, then if one really wants to capture water flow inside this "channel" the value of $\alpha$ must be reduced and the mesh refined accordingly if needed.

## 5.1 Recovering realistic landscapes

Both the consistent MFD and the use of filters are introduced to get rid of any mesh dependency and influence of numerical
noise in the solution. An apparent drawback is that for unperturbed data, complex topographies are less likely to appear by themselves through the perturbations induced by either the numerical approximation or the numerical solvers. Moreover, natural landscapes do have some heterogeneity even for situations where $\tau$ is not that high and that we thus suspect to not being "chaotic". Consequently it would be highly interesting to have a simple tool that allows to recover realistic looking topographies. Fortunately, thanks to our interpretation of MFD as a discretization of (4), we see that the coefficient $k_m$ can
play this role as it will naturally induce heterogeneity in the flow. To illustrate this, we resort to an artificial yet efficient trick, namely the Perlin noise Perlin (1985) that is often used in animated movies or video games to produce realistic looking mountains or river networks.

We thus consider our "three rivers" test case using variable coefficients $k_m$ in space and time (Fig. 16). Figure 16b illustrates a typical distribution in space of the $k_m$ coefficients when using a Perlin noise. The water flow is still distributed between
neighboring cells according to the gradient of the slope, but it will also preferentially choose to enter the cell at the highest $k_m$, especially when the slopes become gentle and relatively close between neighbors. Consequently, heterogeneous $k_m$ coefficients help in keeping the water flow focused even on gentle slopes. Having a realistic range of $k_m$ values may seem uncertain. Indeed, range of $k_m$ values can be compared with the range of typical Gauckler-Manning-Strickler roughness coefficients measured in nature for different vegetation and lithology (e.g. Chow (1959)) but only if the mobility function is considered to be the
hydraulic radius with a unit value of one meter. However, as this assumption is not relevant for the spatial-scales currently considered for in LEMs, there is no direct link between $k_m$ and currently available roughness values. We thus simply fixed for this simulation minimum coefficients values at $k_m = 0.01$ m.s$^{-1}$ and maximum coefficients values at $k_m = 10$ m.s$^{-1}$ to get realistic ranges of the flow velocity in rivers and $k_m$ variation.

The same approach can be applied in the other synthetic test case used in section 3.3. The set of simulations shown in Fig. 17 are performed with spatially and temporally varying $k_m$ coefficients (the same range of $k_m$ values is also used here). The first observation is that more complex topographic structures are simulated (to be compared with Fig. 14). The two first simulations (Fig. 17-a-b) have the same $k_m$ coefficients distribution. We set $(\alpha, \Delta_{xy})$= (1.2 km, 1 km) for the first simulation



(a)  (b)

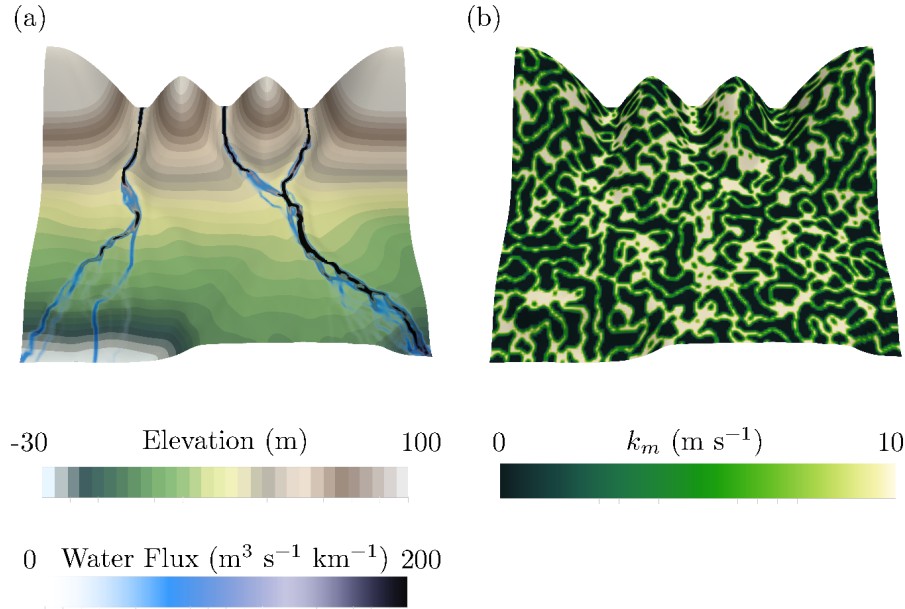

**Figure 16.** The "three rivers" test case with Perlin noise based coefficient $k_m$. a: Final (at T=6My) elevation and associated water flow with variable in space and time $k_m$ coefficients. b: $k_m$ coefficients at T=6My

(Fig. 17-a) while $(\alpha, \Delta_{xy})$= (1.2 km, 0.25 km) is used in the second simulation (Fig. 17-b). The final elevation between those

two simulations are very close, which is what is expected. Differences are still observable, but mainly at small spatial scales, where the effect of the grid refinement allows to capture more details. We then set $(\alpha, \Delta_{xy})$= (0.3 km, 0.25 km) for the third simulation (Fig. 17-c). It shows that using a a filter size close but slighty above to $\Delta_{xy}$ is also important to capture the whole small-scale features that can be simulated on this discretized domain. Finally, the last simulation (Fig. 17-d) has exactly the same parameters that the third one, but another heterogeneity is added on rain-fall, using here again a Perlin noise. Of course the

more complex topography is produced when the more heterogeneity in data is injected. Figure 18 shows a frontal representation of the same simulation with the associated water flow. Comparing with Fig. 9, we have jumped from a non-consistent, non-reproducible and numerical errors dominated model to consistent and reproducible model able to produce complex geomorphic features.

The two synthetic sedimentary systems we have presented so far can be considered as large-scale configurations. However,

some structures emerge clearly at lower spatial scales, as it can be seen in the Gabilan Mesa (California), a case already discussed in detail in Perron et al. (2008) and Perron et al. (2008) and Richardson et al. (2020). We perform a third synthetic case study that has two particularities compared to the two previous ones. First the length scale considered here is small: the domain simulation is a rectangular area of 5 km width over 10 km long and the mesh size mesh size is $\Delta_{xy} = 4$ meters. The filter size is $\alpha = 6$ meters, respecting $\alpha > \Delta_{xy}$. Second, there is no heterogeneity but the sediment production zone



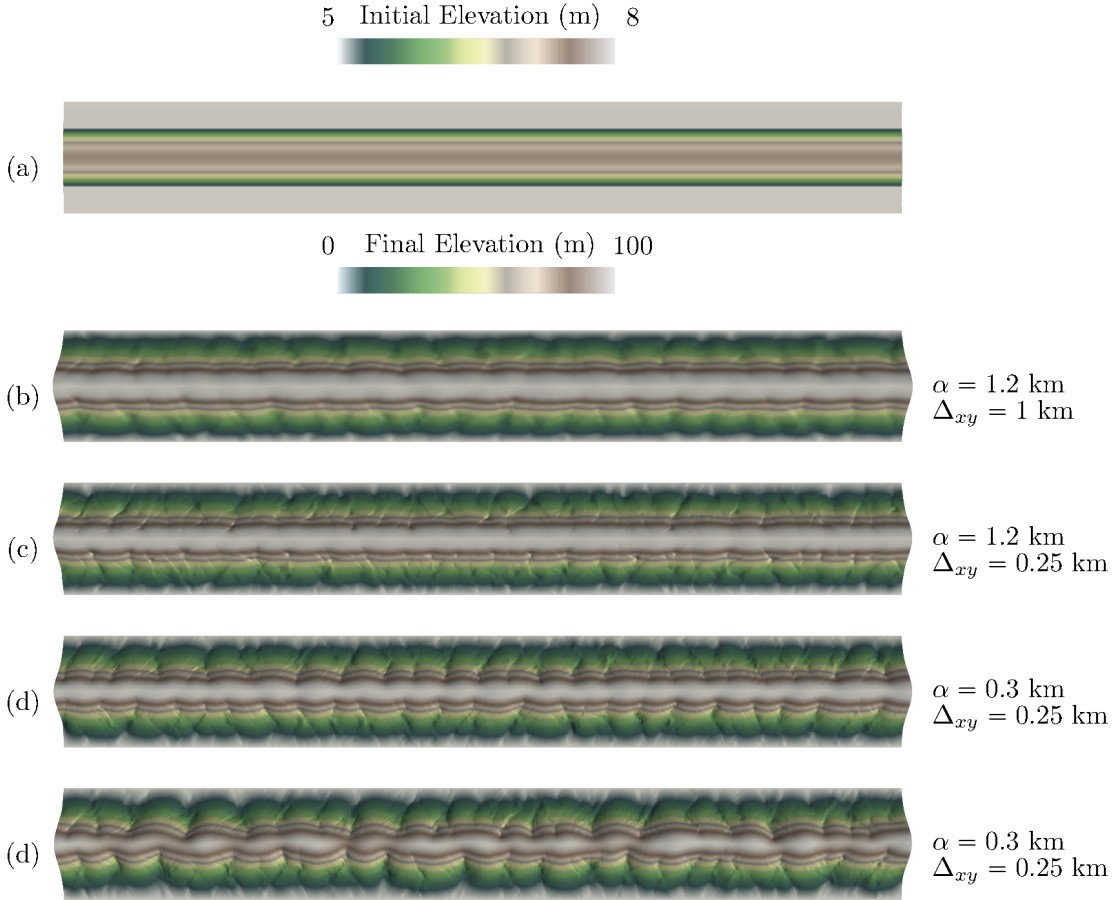

**Figure 17.** Results with filters and Perlin noise based $k_m$ coefficient. a : results for $(\alpha, \Delta_{xy})$= (1.2 km, 1 km), a : results for $(\alpha, \Delta_{xy})$= (1.2 km, 0.25 km), c: results for $(\alpha, \Delta_{xy})$= (0.3 km, 0.25 km), d: results for $(\alpha, \Delta_{xy})$= (0.3 km, 0.25 km) with additional Perlin noise based perturbation of rain fall

($S_s = 100$ m.My$^{-1}$) is a rectangular sub-area which creates topographic discontinuities at the sub-area boundaries. For the other parameters, we keep a similar configuration. Rain-fall is constant in time and space (3000 mm/y) and is the unique water supply. Diffusive coefficients are chosen constant with $k_g$=0.05 km$^2$.My$^{-1}$ and $k_w$= 50000 km$^2$.My$^{-1}$, and models parameters controlling the non-linearity in the water-sediment coupling are set as $r_s = 2$, $p_s = 0$. Simulation is performed over 3 millions and a steady state is achieved.

Results are shown in Fig. 19. The first observation done at the scale of the whole numerical domain shows without any ambiguity the capacity of our model in particularly well preserving symmetry in the sub-area. This is less true outside the sub-area and the reason is probably because the slopes are so gentle that some zones they are considered by the model as flat area, which is not in agreement with the drainage assumption. Second observation is done by comparing the spacing between





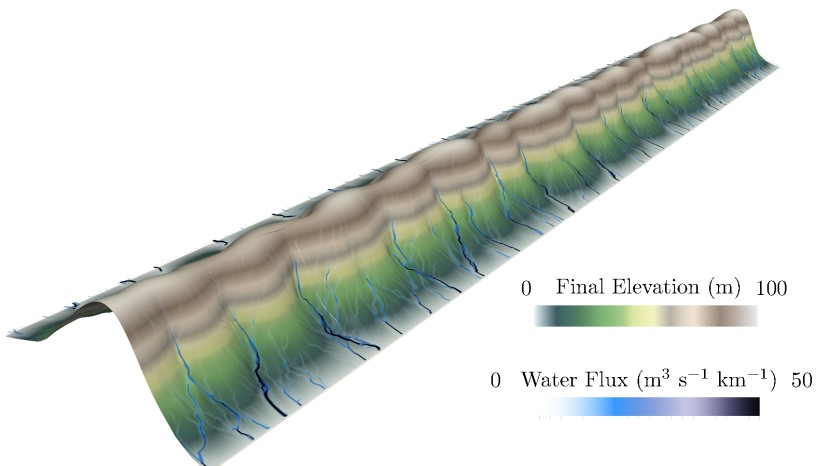

**Figure 18.** Front view of the result of figure 17-d

valleys in this simulation with the valleys spacing observed in the Gabilan Mesa. In our model the valley are spaced by a
lenghtscale of approximately 160 meters, which is at the first order relatively close to what is observed in the Galiban Mesa.
The third observation concerns the high values of $\tau$ here obtained through small values of $k_g$ necessary to obtain such narrow
valleys. With all other parameters left unchanged, we observed that higher values of $k_g$ lead to larger valleys. For such high
values of $\tau$, this test case highlights the absolute necessity of using filters to reproduce realistic structures at such spatial scales.

This result encourages us to say that our model has the ability to reproduce complex and realistic structures. However, further
investigations will have to be performed to confirm. In particular, an detailed analysis of the valleys geometry and spacing will
have to be undertaken to understand more precisely the dependency on $\tau = (k_w q_w^{r_s})/(k_g q_{ref}^{r_s})$.

### 5.2 Overcoming the accumulation and flat areas limitations of MFD approaches

In the general setting, there is no reason why the sediments should evolve in such a way that the "drainage" or "curvature"
assumption (5) is always fulfilled, which can lead to some non physical behavior of the pure MFD algorithms. Indeed, for cells
such that $s_K = 0$, the MFD algorithms stop water in cell $K$ and no flow can go to the neighbours of cell $K$. This can occur
in two obvious situations: when $K$ belongs to an accumulation area (a topographic depression) or a flat area (all neighbours
either higher or at the same level than $K$). In principle, water arriving into an accumulation area should create a "lake" whose
bathymetry will be determined by a water balance between incoming flow, infiltration and evaporation. If the surface reaches
the threshold of the lake, then some water leaves the lake and the water flow restarts from the lake threshold. In flat areas,
water will spread diminishing its height until the full area is covered. To reproduce those effects that are not originally taken





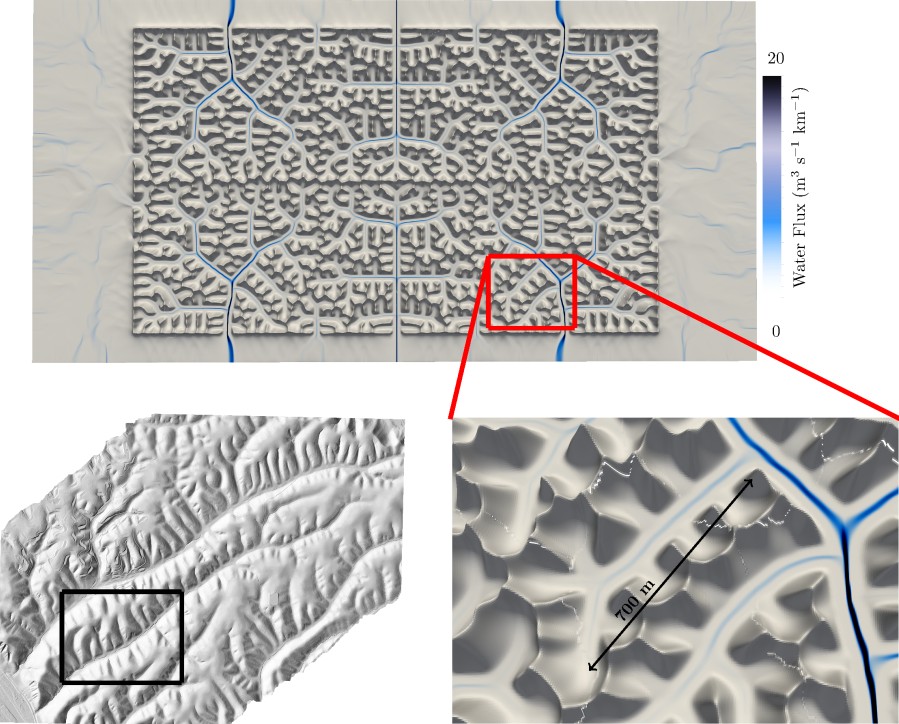

**Figure 19.** Application of our model on a small-scale synthetic sedimentary system (top view). The global domain is $10 \times 5$ km$^2$ with a grid resolution $\Delta_{xy} = 4$ meters. The filter size is $\alpha = 6$ meters. The red square corresponds to a 1 km$^2$ area that contains valleys. Wavelength spacing between two valleys is approximately 160 meters. The Lidar image corresponds to a shaded relief map of a portion of the Gabilan Mesa, California, at approximately 35.9°N, 120.8°W extracted from Dietrich. The black square is around 1 km$^2$ and contains similar valley structures than those obtained in the final state of the simulation

into account, the MFD algorithms and more generally the flow routing algorithms of the literature all incorporate practical workarounds that can take different forms: a uniform distribution of flow over neighbours for flat areas, a water balance under the hypothesis of a flat water surface for accumulation areas, etc. Thanks to our interpretation as the discretization of a continuous model, first we can observe that those limitations should be expected., as (4) is well-posed from the mathematical point of view if $-\Delta (h_s + b) > 0$ which corresponds to a no water accumulation areas or flat areas assumption. The second observation is that we can easily propose a generalization of (4) that overcomes those limitations, by noticing that model (4) is in fact a simplification of the shallow water equations with friction. Indeed, appropriately choosing the friction model and assuming that the mass conservation of water is at steady state a quite general model arising from applying the hydrostatic

805



approximation to the shallow water equations would be to consider (see appendix B):

$$\left|\begin{array}{ll} -div\left(k_m h_w \eta_w(h_w) s_{ref}^{-p_w} ||\nabla(h_w + h_s + b)||^{p_w} \nabla(h_w + h_s + b)\right) = S_w & \text{in } \Omega, \\ -k_m h_w \eta_w(h_w) s_{ref}^{-p_w} ||\nabla(h_w + h_s + b)||^{p_w} \nabla(h_w + h_s + b) \cdot n = B_w & \text{on } \partial\Omega_{\mathcal{N}}, \\ h_w = 0 & \text{on } \partial\Omega_{\mathcal{D}}, \end{array}\right. \tag{29}$$

with the associated water flux strength:

$$q_w = |k_m h_w \eta_w(h_w)| s_{ref}^{-p_w} ||\nabla(h_w + h_s + b)||^{p_w+1}. \tag{30}$$

This is almost (4) except that it uses the hydraulic gradient instead of the topographic one. The assumption $\nabla(h_s+b) \approx \nabla(h_w + h_s + b)$ while valid on pronounced slopes is obviously not valid anymore in accumulation areas (at equilibrium, the hydraulic gradient is almost zero while the topographic gradient is large) and flat areas (where the topographic gradient is zero and the hydraulic one is not), which is coherent with the restriction to areas such that $-\Delta(h_s+b) > 0$. The non-linear model (29) is thus a natural generalization of (4) with a built-in handling of accumulation and flat areas which no longer requires practical workarounds. However, model (29) does not come without any drawbacks. The first one is that we now have to choose the water mobility function $\eta_w$, as we are solving for the water height unknown. This will both influence the repartition of water and the strength of the water flow, while it was completely transparent for the MFD approach with model (4). In the same way, the absolute value of the coefficient $k_m$ will now impact the strength of the water flux through $h_w$, while only its contrasts were relevant for (4). Thus, some fine tuning is required for (29) to produce meaningful results. The last and probably more important drawback is that (29) being non-linear in its unknown $h_w$, its discretization will be more involved than for (4). We perform it using again finite volumes which will in practice require solving non-linear equations instead of solving the well-behaved MFD linear system (8). This is the reason why the MFD remains an attractive alternative when no flat areas appear in the topography, the water balance and flat water surface assumption giving in general good results for accumulation areas. Indeed, let us compare the results obtained with the original Gauckler-Manning-Strickler model (4) and with the more involved hydrologic model (29) on the "three rivers" test case, using filters in both cases. The water mobility function $\eta_w$ for (29) is simply chosen as equal to one if $h_w$ is positive and 0 otherwise.

As we can observe in Fig. 20, if the two models of course do not produce exactly the same results the general behavior is very similar. Even more close results could certainly be obtained by finely tuning the mobility function. We do not want to explore this any further in the present paper and simply want to illustrate that while suffering from some limitations, the consistent MFD (model (4)) is a very strong and attractive approximation on draining topographies. In particular, the MFD computations can easily be an order of magnitude faster that the full hydrologic computations which fully justifies using MFD for draining topographies provided the consistency correction depicted in the first part of the paper is used.

## 6 Conclusions

After recalling the interpretation of Coatléven (2020) of MFD algorithms as discretization of Gauckler-Manning-Strickler and the associated consistency correction, we have explained how to extend it to the most classical MFD algorithms of the





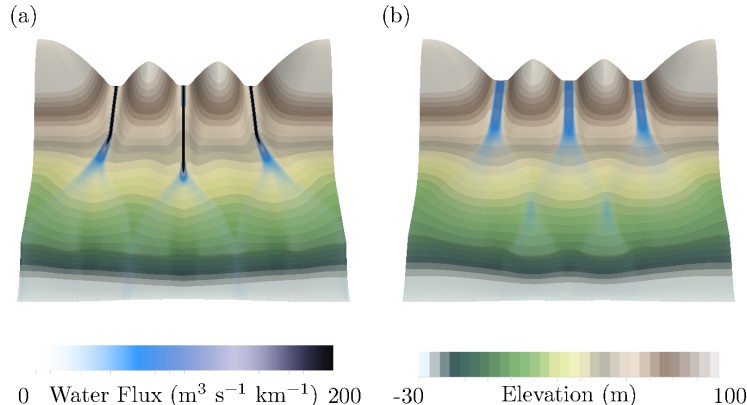

**Figure 20.** Comparison of models (4) and (29) on the "three rivers" test case for $\alpha$=2200 m and $\Delta_{xy}$=2000 m

literature in order to get rid of their well documented mesh dependency. This is a necessary yet not sufficient step to obtain reproducible landscape evolution simulations. Next, we have proposed to mimic the LES strategy for CFD computation in the context of landscape evolution models, relying on the well known Leray-$\alpha$ differential filter. Numerical experiments assess that the combination of consistent MFD and filtering produces results robust to numerical perturbations. It is our belief that this "large structures simulation" (LSS) approach goes far beyond the specific model considered here and that any LEMs

could benefit from it, especially as the cost of implementing and using filtering is not high. Indeed, experiments performed without any filtering strategy have shown that it is extremely difficult to distinguish between the imprint of numerical errors and physical processes. Provided fine enough filter parameter and mesh size are used, only the non physical heterogeneity will disappear. The apparently missing visual complexity that previously arose from numerical noise can be re-introduced when heterogeneous data are considered. Similarly to LES models, we believe that a mathematical analysis and numerical analysis

of the filtered model should be achievable. Both are the subject of active research and we hope to be able to publish such analysis in a future paper. In particular we anticipate that this may bring a more straightforward link between the value of the ratio $\tau$ and the transition from non chaotic to chaotic behaviors. Let us also mention that although we have only presented here a mono-lithologic version of our model, our implementation considers the mutli-lithologic case which opens perspectives for realistic case studies. Finally, pursuing the analogy with LES, an interesting perspective would be to analyze whether it is

feasible to develop sub-filter models to increase the filtered model accuracy when $\alpha$ is quite large, in order to reduce the need for fine $\alpha$ and thus fine meshes and consequently the overall cost of the approach.

*Code availability.* All the numerical schemes used in this paper are fully described in the appendix A. Implementation was performed in code ArcaDES, which is available through the commercial simulator DionisosFlow™.





## Appendix A: Finite volume discretization

In this section we describe the full finite volume discretization of system (2)-(25)-(26)-(27).We assume that the mesh is compatible with the boundary decomposition, i.e. there exists subsets $\mathcal{F}_{ext}^{\mathcal{N}}$ and $\mathcal{F}_{ext}^{\mathcal{D}}$ such that:

$$\overline{\partial\Omega_{\mathcal{N}}} = \bigcup_{\sigma\in\mathcal{F}_{ext}^{\mathcal{N}}} \overline{\sigma} \quad \text{and} \quad \overline{\partial\Omega_{\mathcal{D}}} = \bigcup_{\sigma\in\mathcal{F}_{ext}^{\mathcal{D}}} \overline{\sigma}.$$

Notice that all our simulations without filters employs the same numerical schemes but of course replacing the filtered values by the original ones.

*Leray-$\alpha$ filtering equation:*

Using the TPFA the approximate filter $\mathcal{F}_{\alpha,h}$ is defined for $u_{\mathcal{T}} = ((u_K)_{K\in\mathcal{T}},(u_\sigma)_{\sigma\in\mathcal{F}_{ext}})$ by

$$\mathcal{F}_{\alpha,h}(u_{\mathcal{T}}) = ((\mathcal{F}_{\alpha,K}(u_{\mathcal{T}}))_{K\in\mathcal{T}},(\mathcal{F}_{\alpha,\sigma}(u_{\mathcal{T}}))_{\sigma\in\mathcal{F}_{ext}}),$$

where:

$$\left|\begin{array}{l} \alpha^2 \displaystyle\sum_{\sigma\in\mathcal{F}_K\cap\mathcal{F}_{int}} \frac{|\sigma|}{\overline{d}_{KL}}(\mathcal{F}_{\alpha,K}(u_{\mathcal{T}}) - \mathcal{F}_{\alpha,L}(u_{\mathcal{T}})) + |K|\mathcal{F}_{\alpha,K}(u_{\mathcal{T}}) = |K|u_K \quad \text{for all } K\in\mathcal{T}, \\[2em] \mathcal{F}_{\alpha,\sigma}(u_{\mathcal{T}}) = \mathcal{F}_{\alpha,K}(u_{\mathcal{T}}) \quad \text{for all } K\in\mathcal{T} \text{ and all } \sigma\in\mathcal{F}_K\cap\mathcal{F}_{ext}^{\mathcal{N}}, \\[2em] \mathcal{F}_{\alpha,\sigma}(u_{\mathcal{T}}) = 0 \quad \text{for all } K\in\mathcal{T} \text{ and all } \sigma\in\mathcal{F}_K\cap\mathcal{F}_{ext}^{\mathcal{D}}. \end{array}\right. \tag{A1}$$

The discrete Neumann filter $\mathcal{F}_{\alpha,h}^{\mathcal{N}}$ of course satisfies (A1) but with Neumann boundary conditions on every $\sigma\in\mathcal{F}_{ext}$.

*Sediment mass conservation equations:*

We now assume that the time interval $]0,T[$ is subdivided into $N_T$ subintervals $]t_n,t_{n+1}[$, where $t_0 = 0$ and $t_{N_T+1} = T$. We denote $\Delta t^n = t_{n+1} - t_n$. The discrete quantities associated with time $t_n$ will be denoted as usual with a superscript $n$. The TPFA finite volume scheme for the mass conservation of sediments (2) for the flux (25) is given by:

$$\left|\begin{array}{l} \dfrac{|K|}{\Delta t^n}(h_{s,K}^{n+1} - h_{s,K}^n) + \displaystyle\sum_{\sigma\in\mathcal{F}_K\cap\mathcal{F}_{int}} \frac{|\sigma|}{\overline{d}_{KL}s_{ref}^{p_w}}\eta_{s,\sigma}^{n+1}\Delta\Psi_{KL}^{n,n+1} + \sum_{\sigma\in\mathcal{F}_K\cap\mathcal{F}_{ext}^{\mathcal{D}}} \frac{|\sigma|}{\overline{d}_{K\sigma}s_{ref}^{p_w}}\eta_{s,\sigma}^{n+1}\Delta\Psi_{K\sigma}^{n,n+1}, \\[2em] - \displaystyle\sum_{\sigma\in\mathcal{F}_K\cap\mathcal{F}_{ext}^{\mathcal{N}}} |\sigma|B_{s,\sigma}^{n+1} = |K|S_{s,K}^n \quad \text{for all } K\in\mathcal{T}, \\[2em] h_{s,\sigma}^{n+1} + b_\sigma^{n+1} = h_{s,K}^{n+1} + b_K^{n+1} + \boldsymbol{G}_{s,K}^{n+1}\cdot(\overline{\boldsymbol{x}}_\sigma - \overline{\boldsymbol{x}}_K) \quad \text{for all } K\in\mathcal{T} \text{ and all } \sigma\in\mathcal{F}_K\cap\mathcal{F}_{ext}^{\mathcal{N}}, \\[2em] h_{s,\sigma}^{n+1} = 0 \quad \text{for all } \sigma\in\mathcal{F}_{ext}^{\mathcal{D}}, \end{array}\right. \tag{A2}$$




where

$$\Delta\Psi_{KL}^{n,n+1} = (q_{w,\sigma}^{n+1})^{r_s}||\boldsymbol{G}_{s,\sigma}^{\dagger,n+1}||^{p_{s,1}}(\psi_w(h_{s,K}+b_K)-\psi_w(h_{s,L}+b_L))+||\boldsymbol{G}_{s,\sigma}^{\dagger,n+1}||^{p_{s,2}}(\psi_g(h_{s,K}+b_K)-\psi_g(h_{s,L}+b_L)), \quad \text{(A3)}$$

and

$$\Delta\Psi_{K\sigma}^{n,n+1} = (q_{w,\sigma}^{n+1})^{r_s}||\boldsymbol{G}_{s,\sigma}^{\dagger,n+1}||^{p_{s,1}}(\psi_w(h_{s,K}+b_K)-\psi_w(h_{s,\sigma}+b_\sigma))+||\boldsymbol{G}_{s,\sigma}^{\dagger,n+1}||^{p_{s,2}}(\psi_g(h_{s,K}+b_K)-\psi_g(h_{s,\sigma}+b_\sigma)), \quad \text{(A4)}$$

where the mobility $\eta_{s,\sigma}^{n+1}$ is upwinded using $\Delta\Psi_{KL}^{n,n+1}$ for $\sigma \in \mathcal{F}_{int}$:

$$\eta_{s,\sigma}^{n+1} = \left| \begin{array}{ll} \eta_s(h_{s,K}^{n+1}) & \text{if } \Delta\Psi_{KL}^{n,n+1} \geq 0, \\[2mm] \eta_s(h_{s,L}^{n+1}) & \text{if } \Delta\Psi_{KL}^{n,n+1} < 0, \end{array} \right. \quad \text{(A5)}$$

and using $\Delta\Psi_{K\sigma}^{n,n+1}$ for $\sigma \in \mathcal{F}_{ext}^{\mathcal{D}}$:

$$\eta_{s,\sigma}^{n+1} = \left| \begin{array}{ll} \eta_s(h_{s,K}^{n+1}) & \text{if } \Delta\Psi_{K\sigma}^{n,n+1} \geq 0, \\[2mm] \eta_s(h_{s,\sigma}^{n+1}) & \text{if } \Delta\Psi_{K\sigma}^{n,n+1} < 0, \end{array} \right. \quad \text{(A6)}$$

and where the filtered water flux magnitude is approximated by the harmonic mean whenever possible and the mean value
otherwise:

$$q_{w,\sigma}^{n+1} = \left| \begin{array}{ll} \mathcal{F}_{\alpha,K}^{\mathcal{N}}(q_{w,\mathcal{T}}^{n+1}) & \text{if } \sigma \in \mathcal{F}_{ext}^{\mathcal{D}} \\[3mm] \dfrac{\overline{d}_{KL}\mathcal{F}_{\alpha,K}^{\mathcal{N}}(q_{w,\mathcal{T}}^{n+1})\mathcal{F}_{\alpha,L}^{\mathcal{N}}(q_{w,\mathcal{T}}^{n+1})}{\mathcal{F}_{\alpha,K}^{\mathcal{N}}(q_{w,\mathcal{T}}^{n+1})\overline{d}_{L\sigma}+\mathcal{F}_{\alpha,L}^{\mathcal{N}}(q_{w,\mathcal{T}}^{n+1})\overline{d}_{K\sigma}} & \text{if } \sigma \in \mathcal{F}_{int} \text{ and } \mathcal{F}_{\alpha,K}^{\mathcal{N}}(q_{w,\mathcal{T}}^{n+1}) > 0 \text{ and } \mathcal{F}_{\alpha,L}^{\mathcal{N}}(q_{w,\mathcal{T}}^{n+1}) > 0, \\[3mm] \dfrac{1}{2}(\mathcal{F}_{\alpha,K}^{\mathcal{N}}(q_{w,\mathcal{T}}^{n+1})+\mathcal{F}_{\alpha,L}^{\mathcal{N}}(q_{w,\mathcal{T}}^{n+1})) & \text{if } \sigma \in \mathcal{F}_{int} \text{ and } \mathcal{F}_{\alpha,K}^{\mathcal{N}}(q_{w,\mathcal{T}}^{n+1}) = 0 \text{ or } \mathcal{F}_{\alpha,L}^{\mathcal{N}}(q_{w,\mathcal{T}}^{n+1}) = 0. \end{array} \right. \quad \text{(A7)}$$

We recall that the discrete full topographic gradient is given for any cell $K \in \mathcal{T}$ by:

$$\boldsymbol{G}_{s,K}^n = \frac{1}{|K|} \sum_{\sigma \in \mathcal{F}_K \cap \mathcal{F}_{int}} \frac{|\sigma|}{\overline{d}_{KL}}(h_{s,L}^n+b_L^n-h_{s,K}^n-b_K^n)(\boldsymbol{x}_\sigma-\boldsymbol{x}_K)$$

$$+\frac{1}{|K|} \sum_{\sigma \in \mathcal{F}_K \cap \mathcal{F}_{ext}} \frac{|\sigma|}{\overline{d}_{K\sigma}}(h_{s,\sigma}^n+b_\sigma^n-h_{s,K}^n-b_K^n)(\boldsymbol{x}_\sigma-\boldsymbol{x}_K),$$

while its stabilized version $\boldsymbol{G}_{s,\sigma}^{\dagger,n}$ is given by $\boldsymbol{G}_{s,\sigma}^{\dagger,n} = \boldsymbol{G}_{s,\sigma}^n + \boldsymbol{R}_{s,\sigma}^n$ with:

$$\boldsymbol{G}_{s,\sigma}^n = \left| \begin{array}{ll} \dfrac{1}{2}(\boldsymbol{G}_{s,K}^n+\boldsymbol{G}_{s,L}^n) & \text{if } \mathcal{T}_\sigma = \{K,L\}, \\[3mm] \boldsymbol{G}_{s,K}^n & \text{if } \mathcal{T}_\sigma = \{K\}, \end{array} \right. \quad \text{(A8)}$$

as well as:

$$\boldsymbol{R}_{s,\sigma}^n = \left| \begin{array}{ll} \dfrac{1}{\overline{d}_{KL}^2}\left(h_{s,L}^n+b_L^n-h_{s,K}^n-b_K^n-\boldsymbol{G}_{s,\sigma}^n\cdot(\overline{\boldsymbol{x}}_L-\overline{\boldsymbol{x}}_K)\right)(\overline{\boldsymbol{x}}_L-\overline{\boldsymbol{x}}_K) & \text{if } \mathcal{T}_\sigma = \{K,L\}, \\[3mm] \dfrac{1}{\overline{d}_{K\sigma}^2}\left(h_{s,\sigma}^n+b_\sigma^n-h_{s,K}^n-b_K^n-\boldsymbol{G}_{s,\sigma}^n\cdot(\overline{\boldsymbol{x}}_\sigma-\overline{\boldsymbol{x}}_K)\right)(\overline{\boldsymbol{x}}_\sigma-\overline{\boldsymbol{x}}_K) & \text{if } \mathcal{T}_\sigma = \{K\}. \end{array} \right. \quad \text{(A9)}$$



*Water equations:*

The finite volume scheme for the water equations (26)-(27) is simply obtained by replacing $(h_{s,K} + b_K)_{K \in \mathcal{T}}$ by $(\mathcal{F}_{\alpha,K}(h_{s,\mathcal{T}}^n + b_{\mathcal{T}}^n))_{K \in \mathcal{T}}$ in (8)-(9). In other words we apply a consistent MFD algorithm on the filtered topography and reconstruct a consistent water flux by setting $q_K^{n+1} = \|\boldsymbol{Q}_K^{n+1}\|$ with:

$$\boldsymbol{Q}_K^{n+1} = \sum_{\sigma \in \mathcal{F}_K \cap \mathcal{F}_{int}, \mathcal{F}_{\alpha,K}(h_{s,\mathcal{T}}^n + b_{\mathcal{T}}^n) > \mathcal{F}_{\alpha,L}(h_{s,\mathcal{T}}^n + b_{\mathcal{T}}^n)} \frac{\tau_{KL}^{n,n+1} \widetilde{q}_K^{n+1}}{|K| s_K^{n,n+1}} (\mathcal{F}_{\alpha,K}(h_{s,\mathcal{T}}^n + b_{\mathcal{T}}^n) - \mathcal{F}_{\alpha,L}(h_{s,\mathcal{T}}^n + b_{\mathcal{T}}^n))(\boldsymbol{x}_\sigma - \boldsymbol{x}_K) -$$

$$\sum_{\sigma \in \mathcal{F}_K \cap \mathcal{F}_{int}, \mathcal{F}_{\alpha,K}(h_{s,\mathcal{T}}^n + b_{\mathcal{T}}^n) < \mathcal{F}_{\alpha,L}(h_{s,\mathcal{T}}^n + b_{\mathcal{T}}^n)} \frac{\tau_{KL}^{n,n+1} \widetilde{q}_L^{n+1}}{|K| s_L^{n,n+1}} (\mathcal{F}_{\alpha,L}(h_{s,\mathcal{T}}^n + b_{\mathcal{T}}^n) - \mathcal{F}_{\alpha,K}(h_{s,\mathcal{T}}^n + b_{\mathcal{T}}^n))(\boldsymbol{x}_\sigma - \boldsymbol{x}_K)$$

$$- \sum_{\sigma \in \mathcal{F}_K \cap \mathcal{F}_{ext}} |\sigma| B_{w,\sigma}^{n+1}, \tag{A10}$$

and

$$\left| \begin{aligned} &\widetilde{q}_K^{n+1} - \sum_{\sigma \in \mathcal{F}_K \cap \mathcal{F}_{int}, \mathcal{F}_{\alpha,K}(h_{s,\mathcal{T}}^n + b_{\mathcal{T}}^n) < \mathcal{F}_{\alpha,L}(h_{s,\mathcal{T}}^n + b_{\mathcal{T}}^n)} \tau_{KL}^{n,n+1} \frac{\widetilde{q}_L^{n+1}}{s_L^{n,n+1}} \left( \mathcal{F}_{\alpha,L}(h_{s,\mathcal{T}}^n + b_{\mathcal{T}}^n) - \mathcal{F}_{\alpha,K}(h_{s,\mathcal{T}}^n + b_{\mathcal{T}}^n) \right) \\ &- \sum_{\sigma \in \mathcal{F}_K \cap \mathcal{F}_{ext}} |\sigma| B_{w,\sigma}^{n+1} = |K| S_{w,K}^n \quad \text{for all } K \in \mathcal{T}, \\ &s_K^{n,n+1} = \sum_{\sigma \in \mathcal{F}_K \cap \mathcal{F}_{int}, \mathcal{F}_{\alpha,K}(h_{s,\mathcal{T}}^n + b_{\mathcal{T}}^n) \geq \mathcal{F}_{\alpha,L}(h_{s,\mathcal{T}}^n + b_{\mathcal{T}}^n)} \tau_{KL}^{n,n+1} \left( \mathcal{F}_{\alpha,K}(h_{s,\mathcal{T}}^n + b_{\mathcal{T}}^n) - \mathcal{F}_{\alpha,L}(h_{s,\mathcal{T}}^n + b_{\mathcal{T}}^n) \right) \\ &\tau_{KL}^{n,n+1} = \frac{|\sigma| k_{m,\sigma}^{n+1}}{\overline{d}_{KL} s_{ref}^{p_w}} \| \boldsymbol{G}_{\mathcal{F},s,\sigma}^n \|^{p_w}, \end{aligned} \right. \tag{A11}$$

where

$$\boldsymbol{G}_{\mathcal{F},s,\sigma}^n = \left| \begin{aligned} &\frac{1}{2}(\boldsymbol{G}_{\mathcal{F},s,K}^n + \boldsymbol{G}_{\mathcal{F},s,L}^n) && \text{if } \mathcal{T}_\sigma = \{K, L\}, \\ &\boldsymbol{G}_{\mathcal{F},s,K}^n && \text{if } \mathcal{T}_\sigma = \{K\}, \end{aligned} \right. \tag{A12}$$

and the gradient of the filtered topography is of course given by:

$$\boldsymbol{G}_{\mathcal{F},s,K}^n = \frac{1}{|K|} \sum_{\sigma \in \mathcal{F}_K \cap \mathcal{F}_{int}} \frac{|\sigma|}{\overline{d}_{KL}} (\mathcal{F}_{\alpha,L}(h_{s,\mathcal{T}}^n + b_{\mathcal{T}}^n) - \mathcal{F}_{\alpha,K}(h_{s,\mathcal{T}}^n + b_{\mathcal{T}}^n))(\boldsymbol{x}_\sigma - \boldsymbol{x}_K)$$

$$+ \frac{1}{|K|} \sum_{\sigma \in \mathcal{F}_K \cap \mathcal{F}_{ext}} \frac{|\sigma|}{\overline{d}_{K\sigma}} (\mathcal{F}_{\alpha,\sigma}(h_{s,\mathcal{T}}^n + b_{\mathcal{T}}^n) - \mathcal{F}_{\alpha,K}(h_{s,\mathcal{T}}^n + b_{\mathcal{T}}^n))(\boldsymbol{x}_\sigma - \boldsymbol{x}_K).$$





**Appendix B: From shallow water model to the steady-state hydrologic model** (29)

Recall that the shallow water systems is given by (see Birnir et al. (2001); Peton et al. (2020)):

$$
\left|
\begin{array}{l}
\dfrac{\partial h_w}{\partial t} + div(h_w \boldsymbol{u}_w) = 0, \\[2mm]
\dfrac{\partial}{\partial t}(h_w \boldsymbol{u}_w) + div(h_w \boldsymbol{u}_w \otimes \boldsymbol{u}_w) + g h_w \nabla(h_s + b + h_w) = -\kappa_w \left(h_w, ||\nabla(h_w + h_s + b)||\right) |\boldsymbol{u}_w|^{r_w} \boldsymbol{u}_w,
\end{array}
\right.
\tag{B1}
$$

where $\boldsymbol{u}_w$ denotes the water speed, $g$ the acceleration due to gravity, and $\kappa_w$ is the friction coefficient. Then, following Peton et al. (2020) and defining $H_{s,c}$ to be the characteristic sediment height, $H_{w,c}$ the characteristic water height, $L_c$ the characteristic domain length, $T_c$ the characteristic time and defining the nondimensional variables:

$$
\hat{h}_s = \frac{h_s}{H_{s,c}}, \quad \hat{b}_s = \frac{b}{H_{s,c}}, \quad \hat{h}_w = \frac{h_w}{H_{w,c}}, \quad \hat{\boldsymbol{u}}_w = \frac{T_c \boldsymbol{u}_w}{L_c}, \quad \hat{x} = \frac{x}{L_c}, \quad \hat{y} = \frac{y}{L_c}, \quad \hat{t} = \frac{t}{T_c},
$$

we see that (B1) is equivalent to:

$$
\left|
\begin{array}{l}
\dfrac{\partial \hat{h}_w}{\partial \hat{t}} + \hat{div}(\hat{h}_w \hat{\boldsymbol{u}}_w) = 0, \\[3mm]
\dfrac{\partial}{\partial \hat{t}}(\hat{h}_w \hat{\boldsymbol{u}}_w) + \hat{div}(\hat{h}_w \hat{\boldsymbol{u}}_w \otimes \hat{\boldsymbol{u}}_w) + g\dfrac{H_{s,c} T_c^2}{L_c^2} \hat{h}_w \hat{\nabla}(\hat{h}_s + \hat{b}) + g\dfrac{H_{w,c} T_c^2}{L_c^2} \hat{h}_w \hat{\nabla}(\hat{h}_w), \\[3mm]
= -\kappa_w \left(h_w, ||\nabla(h_w + h_s + b)||\right) \dfrac{L_c}{H_{w,c}} \left(\dfrac{L_c}{T_c}\right)^{r_w - 1} |\hat{\boldsymbol{u}}_w|^{r_w} \hat{\boldsymbol{u}}_w.
\end{array}
\right.
$$


The "shallow" hypothesis corresponds to assuming that $L_c/H_{w,c} >> 1$, while the two numbers

$$
F_{r,w} = \frac{L_c}{\sqrt{gH_{w,c}} T_c} \quad \text{and} \quad F_{r,s} = \frac{L_c}{\sqrt{gH_{s,c}} T_c},
$$

are equivalent to Froude numbers for the water and sediment flows. For long term sediment evolution, it is reasonable to assume that $F_{r,w} << 1$ and $F_{r,s} << 1$, i.e. that gravity is the dominant phenomenon. Combined with the shallow water assumption
this suggests to neglect the inertia terms in the nondimensional momentum balance, leading to the hydrostatic assumption:

$$
g h_w \nabla(h_s + b + h_w) = -\kappa_w \left(h_w, ||\nabla(h_w + h_s + b)||\right) |\boldsymbol{u}_w|^{r_w} \boldsymbol{u}_w,
\tag{B2}
$$

Inverting formula (B2) we obtain the following expression for the water speed:

$$
\boldsymbol{u}_w = -\mu_w \left(h_w, ||\nabla(h_w + h_s + b)||\right) \nabla(h_s + b + h_w),
\tag{B3}
$$

where

$$
\mu_w \left(h_w, ||\nabla(h_w + h_s + b)||\right) = \frac{g^{\frac{1}{r_w+1}} h_w^{\frac{1}{r_w+1}}}{\kappa_w \left(h_w, ||\nabla(h_w + h_s + b)||\right)^{\frac{1}{r_w+1}}} ||\nabla(h_s + b + h_w)||^{-\frac{r_w}{r_w+1}}.
\tag{B4}
$$

Thus, appropriately choosing the friction model, for instance by setting $r_w = 0$ and

$$
\kappa_w \left(h_w, ||\nabla(h_w + h_s + b)||\right) = \frac{g h_w}{k_m \eta_w(h_w) s_{ref}^{-p_w} ||\nabla(h_w + h_s + b)||^{p_w}},
\tag{B5}
$$



and assuming that the mass conservation of water is at steady state we obtain the following quite general hydrostatic approximation to the shallow water equations:

$$
\left|
\begin{array}{ll}
-div\left(k_m h_w \eta_w\left(h_w\right) s_{ref}^{-p_w}||\nabla(h_w + h_s + b)||^{p_w}\nabla(h_w + h_s + b)\right) = S_w & \text{in } \Omega, \\[2mm]
-k_m h_w \eta_w\left(h_w\right) s_{ref}^{-p_w}||\nabla(h_w + h_s + b)||^{p_w}\nabla(h_w + h_s + b)\cdot n = B_w & \text{on } \partial\Omega_{\mathcal{N}}, \\[2mm]
h_w = 0 & \text{on } \partial\Omega_{\mathcal{D}},
\end{array}
\right.
$$

with the associated water flux strength:

$$
q_w = |k_m h_w \eta_w\left(h_w\right)|\, s_{ref}^{-p_w}||\nabla(h_w + h_s + b)||^{p_w+1}.
$$

**Remark B.1.** The friction model (B5) becomes singular when $||\nabla(h_w + h_s + b)|| = 0$. Thus, an alternate choice would be to use:

$$
\kappa_w\left(h_w, ||\nabla(h_w + h_s + b)||\right) = \frac{g h_w}{k_m \eta_w(h_w)(\beta + s_{ref}^{-p_w}||\nabla(h_w + h_s + b)||^{p_w})},
$$

for some $\beta > 0$ (the same holds for function $\eta_w$ such that $\eta(0) = 0$). This alternate choice is probably more physical, as the term in $s_{ref}^{-p_w}||\nabla(h_w + h_s + b)||^{p_w}$ can be interpreted as modeling some deceleration in accumulation areas. We have chosen to use (B5) to be as close as possible to the MFD algorithms of the literature.

*Author contributions.* Julien Coatléven: conceptualization, writing, software, simulations. Benoit Chauveau: writing, software, simulations

*Competing interests.* Both the authors are core developers of the ArcaDES simulator supporting DionisosFlow™, a commercial stratigraphic simulator.

*Acknowledgements.* The authors would like to thank John J. Armitage and Didier Granjeon for their careful reading of the present paper.



## Appendix: References

Armitage, J. J.: Short communication: flow as distributed lines within the landscape, Earth Surface Dynamics, 7, 67–75,
https://doi.org/10.5194/esurf-7-67-2019, 2019.

Audusse, E., Bouchut, F., Bristeau, M.-O., Klein, R., and Perthame, B.: A Fast and Stable Well-Balanced Scheme with Hydrostatic Recon-
struction for Shallow Water Flows, SIAM Journal on Scientific Computing, 25, 2050–2065, https://doi.org/10.1137/S1064827503431090,
2004.

Balay, S., Gropp, W., McInnes, L. C., and Smith, B. F.: PETSc, the portable, extensible toolkit for scientific computation, Argonne National
Laboratory, 2, 1998.

Banninger, D.: Technical Note: Water flow routing on irregular meshes, Hydrol. Earth Syst. Sci., Vol. 11, pp. 1243-1247, 2007.

Bardos, C.: Problèmes aux limites pour les équations aux dérivées partielles du premier ordre à coefficients réels ; Théorèmes
d'approximation ; Application à l'équation de transport, Ann. Sci. Ec. Norm. Sup. Ser. 4, Vol. 3, pp. 185-233, 1970.

Bates, P. D., Horritt, M. S., and Fewtrell, T. J.: A simple inertial formulation of the shallow water equations for efficient two-dimensional
flood inundation modelling, Journal of Hydrology, 387, 33–45, 2010.

Beasley, D. . B., Huggins, L. F., and Monke, E. J.: ANSWERS: A Model for Watershed Planning, Transactions of the ASAE, Vol. 23(4), pp.
938-944, 1980.

Berselli, L. C., Iliescu, T., and Layton, W. J.: Mathematics of Large Eddy Simulation of Turbulent Flows, Springer Berlin, Heidelberg, 2005.

Birnir, B., Smith, T. R., and Merchant, G. E.: The scaling of fluvial landscapes, Comput. Geosci., 27(10), 1189–1216,
https://doi.org/https://doi.org/10.1016/S0098-3004(01)00022-X, 2001.

Bonetti, S., Bragg, A. D., and Porporato, A.: On the theory of drainage area for regular and non-regular points, Proc. R. Soc. A 474: 20170693,
2018.

Bonetti, S., Hooshyar, M., Camporeale, C., and Porporato, A.: Channelization cascade in landscape evolution, PNAS, Vol. 117(3), pp. 1375-
1382, 2020.

Braun, J. and Willett, S. D.: A very efficient O(n), implicit and parallel method to solve the stream power equation governing fluvial incision
and landscape evolution, Geomorphology, 180-181, 170–179, https://doi.org/https://doi.org/10.1016/j.geomorph.2012.10.008, 2013.

Braun, J., Robert, X., and Simon-Labric, T.: Eroding dynamic topography, Geophysical Research Letters, Vol 40(8), pp 1494-1499, 2013.

Braun, J., Voisin, C., Gourlan, A. T., and Chauvel, C.: Erosional response of an actively uplifting mountain belt to cyclic rainfall variations,
Earth Surface Dynamics, 3, 1–14, https://doi.org/10.5194/esurf-3-1-2015, 2015.

Carretier, S., Guerit, L., Harries, R., Regard, V., Maffre, P., and Bonnet, S.: The distribution of sediment residence times at the foot
of mountains and its implications for proxies recorded in sedimentary basins, Earth and Planetary Science Letters, 546, 116 448,
https://doi.org/https://doi.org/10.1016/j.epsl.2020.116448, 2020.

Cheskidov, A., Olson, E., Holm, D., and Titi, E.: On a Leray-$\alpha$ model of turbulence, Proc. R. Soc. Lond. Ser. A Math. Phys. Eng. Sci., Vol.
146, pp. 1-21, 2005.

Chow, V.: Open-channel hydraulics, McGraw-Hill, New York, 1959.

Coatléven, J.: Some multiple flow direction algorithms for overland flow on general meshes, ESAIM: Mathematical Modelling and Numerical
Analysis, Vol. 54 (6), pp. 1917-1949, 2020.

Costa-Cabral, M. C. and Burge, S. J.: Digital elevation model networks (DEMON): A model of flow over hillslopes for computation of
contributing and dispersal areas, Water resources research, Vol. 30(6), pp. 1681-1692, 1994.



Coulthard, T. J., Kirkby, M. J., and Macklin, M. G.: Modelling geomorphic response to environmental change in an upland catchment, Vol 14(11-12), pp 2031-2045, Hydrological Processes, 2000.

Coulthard, T. J., Neal, J. C., Bates, P. D., Ramirez, J., de Almeida, G. A. M., and Hancock, G. R.: Integrating the LISFLOOD-FP 2D hydrodynamic model with the CAESAR model: implications for modelling landscape evolution, Earth Surface Processes and Landforms, 38, 1897–1906, 2013.

Davy, P. and Lague, D.: Fluvial erosion/transport equation of landscape evolution models revisited, Journal of Geophysical Research: Earth Surface, 114, https://doi.org/https://doi.org/10.1029/2008JF001146, 2009.

Davy, P., Croissant, T., and Lague, D.: A precipiton method to calculate river hydrodynamics, with applications to flood prediction, landscape evolution models, and braiding instabilities, Journal of Geophysical Research: Earth Surface, 122, 1491–1512, https://doi.org/https://doi.org/10.1002/2016JF004156, 2017.

Desmet, P. J. J. and Govers, G.: Comparison of routing algorithms for digital elevation models and their implication for predicting ephemeral gullies, Int. J. Geo. Inf. Syst., Vol. 10(3), pp. 311-331, 1996.

Dietrich, W.: High Resolution Topography over Gabilan Mesa, CA 2003. National Center for Airborne Laser Mapping (NCALM)., OpenTopography. https://doi.org/10.5069/G947481V.. Accessed: 2023-03-29.

DiPerna, R. and Lions, P.: Ordinary differential equations, transport theory and Sobolev spaces, Invent. Math. 98, 511-547, 1989.

Egholm, D., Pedersen, V., Knudsen, M., and Larsen, N.: Coupling the flow of ice, water, and sediment in a glacial landscape evolution model, Geomorphology, 141-142, 47–66, https://doi.org/https://doi.org/10.1016/j.geomorph.2011.12.019, 2012.

Egholm, D. L., Knudsen, M. F., Clark, C. D., and Lesemann, J. E.: Modeling the flow of glaciers in steep terrains: The integrated second-order shallow ice approximation (iSOSIA), Journal of Geophysical Research: Earth Surface, 116, https://doi.org/https://doi.org/10.1029/2010JF001900, 2011.

Erskine, R. H., Green, T. R., Ramirez, J. A., and MacDonald, L. H.: Comparison of grid-based algorithms for computing upslope contributing area, Water Resour. Res., Vol. 42, W09416, 2006.

Eymard, R., Gallouët, T., Gervais, V., and Masson, R.: Existence and uniqueness of a weak solution to a stratigraphic model, pp. 278–287, M. Feistauer, V. Dolejšì, P. Knobloch, K. Najzar (eds), Springer, Berlin, 2004.

Eymard, R., Gallouët, T., Gervais, V., and Masson, R.: Convergence of a numerical scheme for stratigraphic modeling, SIAM J. Numer.
Anal., Vol. 43(2), pp. 474-501, 2005.

Fairfield, J. and Leymarie, P.: Drainage Networks From Grid Digital Elevation Model, Water resources research, Vol. 27(5), pp. 709-717, 1991.

Fernández-Cara, E., Guillén, F., and Ortega, R.: Mathematical modeling and analysis of visco-elastic fluids of the Oldroyd kind, P.G. Ciarlet, J.L. Lions (Eds.), Numerical Methods for Fluids, Part 2, in: Handbook of Numerical Analysis, vol. VIII, North-Holland, Amsterdam, pp.
543–661, 2002.

Freeman, T. G.: Drainage with divergent flow over a regular grid, Proc. 8th Biennial Conf. Simulation Society of Australia, Canberra, pp. 160-165, 1989.

Freeman, T. G.: Calculating catchment area with divergent flow based on a regular grid, Computers & Geosciences Vol. 17(3), pp. 413-422, 1991.

Gallant, J. C. and Hutchinson, M. F.: A differential equation for specific catchment area, Water resources research, Vol. 47, W05535, 2011.

Gilbert, G.: Geology of the Henry Mountains, US Geographical and Geological Survey, Washington, D.C, 1880.



Girault, V. and Tartar, L.: $L^p$ and $W^{1,p}$ regularity of the solution of a steady transport equation, C. R. Acad. Sci. Paris, Ser. I, Vol. 348, pp. 885-890, 2010.

Graf, W. H. and Altinakar, M. S.: Hydraulique fluviale: Ecoulement et phénomènes de transport dans les canaux à géométrie simple, Traité de génie civil, vol. 16, Presses polytechniques et universitaires romandes, 2000.

Granjeon, D.: Modélisation stratigraphique déterministe: Conception et applications d'un modèle diffusif 3-d multilithologique, Ph.D. thesis, Universit é de Rennes I, 1996.

Granjeon, D.: 3D forward modelling of the impact of sediment transport and base level cycles on continental margins and incised valleys, chap. 16, pp. 453–472, 2014.

Grospellier, G. and Lelandais, B.: The Arcane development framework, POOSC 09: Proceedings of the 8th workshop on Parallel/High-Performance Object-Oriented Scientific Computing, pp. 1–11, https://doi.org/https://doi.org/10.1145/1595655.1595659, 2009.

Guermond, J.-L., Oden, J., and Prudhomme, S.: An interpretation of the Navier-Stokes-alpha model as a frame-indifferent Leray regularization, Phys. D, Vol. 177(1-4), pp. 23-30, 2003.

Hergarten, S.: Modeling glacial and fluvial landform evolution at large scales using a stream-power approach, Earth Surface Dynamics, 9, 937–952, https://doi.org/10.5194/esurf-9-937-2021, 2021.

Holmgren, P.: Multiple flow direction algorithms for runoff modelling in grid based elevation models: an empirical evaluation, Hydrological processes, Vol. 8, pp. 327-334, 1994.

Hooshyar, M. and Porporato, A.: Mean dynamics and elevation-contributing area covariance in landscape evolution models, Water Resources Research, Vol. 57, e2021WR029727, 2021a.

Hooshyar, M. and Porporato, A.: Spectral signature of landscape channelization, Geophysical Research Letters, Vol. 48, e2020GL091015, 2021b.

Hooshyar, M., S.Bonetti, Singh, A., Foufoula-Georgioui, E., and Porporato, A.: From turbulence to landscapes: Logarithmic mean profiles in bounded complex systems, Phys. Rev. E, 102, 033 107, 2020.

Lea, N. L.: An aspect driven kinematic routing algorithm, in: Overland Flow: Hydraulics and Erosion Mechanics, edited by Parsons, A. J. and Abrahams, A., chap. 16, Chapman and Hall, New York, 1992.

Leopold, L. B., Wolman, M. G., and Miller, J. P.: Fluvial Processes in Geomorphology, W. H. Freeman, San Francisco, California, 1964.

Leray, J.: Sur le mouvement d'un fluide visqueux emplissant l'espace, Acta Math., Vol. 63 pp. 193-248, 1934.

Maxwell, J. C.: On hills and dales, Philos. Mag. J. Sci., Vol. 4/40(269), pp. 421-427, 1870.

Nones, M.: On the main components of landscape evolution modelling of river systems, Acta Geophys. Vol 68, pp 459–475, 2020.

O'Callaghan, J. F. and Mark, D. M.: The Extraction of Drainage Networks from Digital Elevation Data, Computer vision, graphics and image processing, Vol. 28, pp. 323-344, 1984.

Orlandini, S. and Moretti, G.: Determination of surface flow paths from gridded elevation data, Water Resour. Res., Vol. 45, W03417, 2009.

Pelletier, J.: 2.3 Fundamental Principles and Techniques of Landscape Evolution Modeling, in: Treatise on Geomorphology, edited by Shroder, J. F., pp. 29–43, Academic Press, San Diego, 2013.

Pelletier, J. D.: Minimizing the grid-resolution dependence of flow-routing algorithms for geomorphic applications, Geomorphology, Vol. 122 , pp. 91-98, 2010.

Perlin, K.: An image synthesizer, ACM SIGGRAPH Computer Graphics, 19, 287–296, https://doi.org/https://doi.org/10.1145/325165.325247, 1985.

Perron, J. T., Dietrich, W. E., and Kirchner, J. W.: Controls on the spacing of first-order valleys, J. Geophys. Res., Vol. 113, F04016, 2008.





Perron, J. T., Kirchner, J. W., and Dietrich, W. E.: Formation of evenly spaced ridges and valleys, Nature, Vol. 460, pp. 502-505, 2009.

Peton, N., Cancès, C., Granjeon, D., Tran, Q.-H., and Wolf, S.: Numerical scheme for a water flow-driven forward stratigraphic model, Computational Geosciences, vol. 24 pp. 37-60, 2020.

Porporato, A.: Hydrology without dimensions, Hydrol. Earth Syst. Sci., Vol. 26, pp. 355-374, 2022.

Qin, C., Zhu, A.-X., Pei, T., Li, B., Zhou, C., and Yang, L.: An adaptive approach to selecting a flow-partition exponent for a multiple-flow-

direction algorithm, International Journal of Geographical Information Science Vol. 21(4), pp. 443-458, 2007.

Quinn, P., Beven, K., Chevallier, P., and Planchon, O.: The prediction of hillslope flow paths for distributed hydrological modelling using digital terrain models, Hydrological processes, Vol. 5, pp. 59-79, 1991.

Quinn, P., Beven, K., and Lamb, R.: The $\ln(a\tan\beta)$ index : how to calculate it and how to use it within the TOPMODEL framework, Hydrological processes, Vol. 9, pp. 161-182, 1995.

Richardson, A., Hill, C. N., and Perron, J. T.: IDA: An implicit, parallelizable method for calculating drainage area, Water Resour. Res. Vol. 50, pp. 4110-4130, 2014.

Richardson, P. W., Perron, J. T., Miller, S. R., and Kirchner, J. W.: Unraveling the Mysteries of Asymmetric Topography at Gabilan Mesa, California, Journal of Geophysical Research: Earth Surface, 125, e2019JF005 378, https://doi.org/https://doi.org/10.1029/2019JF005378, e2019JF005378 2019JF005378, 2020.

Salles, T.: eSCAPE: parallel global-scale landscape evolution model, Journal of Open Source Software, 3, 964, https://doi.org/10.21105/joss.00964, 2018.

Salles, T., Flament, N., and Müller, D.: Influence of mantle flow on the drainage of eastern Australia since the Jurassic Period, Geochemistry, Geophysics, Geosystems, 18, 280–305, https://doi.org/https://doi.org/10.1002/2016GC006617, 2017.

Scheingross, J. S., Limaye, A. B., McCoy, S. W., and Whittaker, A. C.: The shaping of erosional landscapes by internal dynamics, Nat Rev

Earth Environ Vol. 1, pp. 661-676, 2020.

Schoorl, J. M., Sonneveld, M. P. W., and Veldkamp, A.: Three-dimensional landscape process modelling: the effect of DEM resolution, Earth Surface Processes and Landforms, Vol 25(9), pp 1025-1034, 2000.

Seibert, J. and McGlynn, B. L.: A new triangular multiple flow direction algorithm for computing upslope areas from gridded digital elevation models, Water Resour. Res., Vol. 43, W04501, 2007.

Smith, T. R. and Bretherton, F. P.: Stability and the Conservation of Mass in Drainage Basin Evolution, Water Resour. Res., Vol. 8(6), W03417, 1972.

Srivastava, A., Yetemen, O., Saco, P. M., Rodriguez, J. F., Kumari, N., and Chun, K. P.: Influence of orographic precipitation on coevolving landforms and vegetation in semi-arid ecosystems, Earth Surface Processes and Landforms, 47, 2846–2862, https://doi.org/https://doi.org/10.1002/esp.5427, 2022.

Tarboton, D. G.: A new method for the determination of flow directions and upslope areas in grid digital elevation models, Water resources research, Vol. 33(2), pp. 309-319, 1997.

Tucker, G. and Slingerland, R.: Drainage basin responses to climate change, Water Resources Research, Vol 3(8),pp. 2031–2047, 1997.

Tucker, G. E. and Hancock, G. R.: Modelling landscape evolution, Earth Surface Processes and Landforms, 35, 28–50, https://doi.org/https://doi.org/10.1002/esp.1952, 2010.

Valters, D.: Modelling Geomorphic Systems: Landscape Evolution, p. 6.5.12, https://doi.org/10.13140/RG.2.1.1970.9047, 2016.

Van der Beek, P. A.: Environmental Modelling: Finding Simplicity in Complexity,, pp. 309–332, 2nd Edition (eds. J. Wainwright and M. Mulligan), Wiley, Chichester, 2013.



Veiga, H. B. D.: Existence results in Sobolev spaces for a stationary transport equation, Ricerche Mat. Suppl. XXXVI pp. 173-184, 1987.

Willgoose, G.: User manual for SIBERIA (Version 8.30), Telluric Research, 2005.

Willgoose, G., Bras, R., and Rodriguez-Iturbe, I.: A physically based channel networkand catchment evolution model, Ph.D. thesis, TR 322, Ralph M. Parsons Laboratory, Dept. ofCivil Engineering, MIT, Boston, M, 1989.

Willgoose, G., Bras, R. L., and Rodriguez-Iturbe, I.: A coupled channel network growth and hillslope evolution model: 1. Theory, Water Resources Research, 27, 1671–1684, https://doi.org/https://doi.org/10.1029/91WR00935, 1991.

Wilson, J. P., Aggett, G., Deng, Y., and Lam, C. S.: Water in the Landscape: A Review of Contemporary Flow Routing Algorithms, pp.

213–236, Qiming Zhou and Brian Lees and Guo-an Tang. Springer Berlin Heidelberg, Springer Berlin, Heidelberg, 2008.

Zhiyin, Y.: Large-eddy simulation: Past, present and the future, Chinese Journal of Aeronautics, 28, 11–24, 2015.

Zhou, Q., Pilesjö, P., and Chen, Y.: Estimating surface flow paths on a digital elevation model using a triangular facet network, Water Resour. Res., Vol. 47, W07522, 2011.