# Peer review of "Large structures simulation for landscape evolution models"

_EGUsphere, 2023_

## Referee Comment (RC2)

**REVIEW OF "LARGE STRUCTURES SIMULATION FOR LANDSCAPE EVOLUTION MODELS"**

I have detailed first, what I believe, are major points of concern in the manuscript and then I have listed some specific minor comments. I have tried to express my reservations clearly. Some of my concerns may seem direct, but I offer them constructively. Overall, I think the submitted work would need at least (maybe more than) a major revision.

**1 Major Comments**

**1.1 Presentation**

The quality of the writing needs improvement. With the current way the paper is written, it will be unreadable to a vast majority of readers who know a bit about landscape evolution modeling (I am not even counting users or experimentalists here). I had to do multiple rounds of reading to understand the work; which led to a delay in submitting my review here. I am giving this opinion as a person whose research has focused on landscape evolution modeling. I understand that the authors have tried to bring mathematical correctness to their explanation, but I don't think it will be conveyed to the earth science community in this format. I have written papers that contained very few fractions of mathematical jargon that is here and the papers are not well received. So, just in terms of presentation, a major rewrite is required. The current draft feels to me like a paper from the SIAM journal on applied mathematics.

**1.2 Motivation/Premise**

I feel after reading the abstract and introduction multiple times that jargon and references to previous works are used without careful consideration. If anyone reads the abstract of this draft, it will seem that due to nonlinear (the authors say "chaotic" - I don't understand how that feedback is chaotic, which means a completely different phenomenon of the equations), the previous LEMs are likely full of (or blurred with) numerical errors. This is a very big claim in my understanding, without proper justification. Again, if authors have some interpretation of "chaotic" behavior, that should be written clearly and in the particular context they are analyzing.

Result from Line 505: The issue of a "chaotic" drainage network is only for $rs > 1$ in the authors' model. If that is correct, it should be made clearer way earlier in the manuscript rather than here. And for $rs < 1$, there is no numerical issue without filtering, Is that correct? Again this information is crucial that is lost in the middle of the manuscript. How is this observation related to the work of Shelef et al. and Tucker et al. based on the concavity index? Is $rs$ for your model working the same as the concavity index used in other landscape evolution models that are referenced in the beginning?

In the abstract, the authors discuss and compare "numerical instability" to "turbulence instability". I am familiar with the works of Porporato and co-authors cited in the introduction to which current authors attribute this comparison. Those authors introduced "channel instability" in reference to the properties of the landscape evolution equations and the physical feedback of eroding more where the accumulation of more water and then again accumulation, which results in a branching cascade, similar to the energy cascade in a turbulent flow. This is very different than numerical instability propagation, as claimed by the authors.

The authors write on page 2: "However, in the absence of reference analytic solutions, it is hard to decipher if the obtained landform results from physical processes or from the self-amplification of initially small numerical errors." For the same group of studies by Porporato and co-authors, analytical results have been presented and numerical solutions have been tested to agree well with the solutions. In Bonetti et al. (2020, PNAS) and Anand et al.(2022, JGR), linear stability analysis of the smooth solutions is matched with numerical solutions under different exponent values. I don't understand how these results don't represent the physical process with certain accuracy, as the current authors claim. I completely agree that numerical errors/grid orientation errors may not be completely absent, but they are not detrimental as appropriated by the authors in motivating their work.

In my understanding, the presented work attempts to improve the impact of grid orientation on the numerical approximation by flow-direction algorithms for the GMS mode. After reading Section 3 carefully, this work focuses on extending the work of Coatléven (2020), which I could not guess from the introduction. If I am not completely wrong here; I am getting more convinced that the title would need changes to convey that message clearly.

If I have understood this work correctly; the authors' new results are shown for the sediment model of Granjeon (1996); Eymard et al. (2004, 2005); Peton et al. (2020). These papers are not discussed/cited in the manuscript until the main result section 4. Also there, these papers are just cited quickly. I feel authors should focus explicitly more on these models and their specific results in a clean way. The approach of using different models mentioned before for motivation and using different models for explaining results is very confusing. I can see the urge to show the relevance and generalization of these results to other models, but the current presentation is not the way to do it.

**1.3 Technical Points**

In para 50-60, the authors say that flow-direction algorithms can not approximate well their GMS model (which is a shallow-water approximation). This does not look unexpected to me because the flow-direction algorithm are describing the kinetic flow of water under surface gradient (gravity balancing friction), which is different than the model using shallow water approximation. The physics of the two cases is different, I don't think that can be just attributed to big numerical error. Again, I am not saying grid resolution and numerical errors due to imposed orientation are completely absent, but this attribution does not seem accurate. Please correct me if I am wrong here.

For the LES part of the motivation, the authors claimed and described in lines 80-85, Hooshyar and Porporato (2021a); Porporato (2022) implemented LES. They described the potential use of LES based on the dimensional analysis results for different detachment-limited erosion exponents and they verified those claims using a DNS-type numerical approach (Anand et al. 2020), not an LES approach.

In line 150 the authors write, "Setting Qw = a, this allows to reduce the mesh dependency to the usual consistency errors of numerical schemes." This is already shown in the works of Porporato and co-authors, which is not attributed clearly here. They used $a$ specific drainage area (defined at a point) instead of total catchment area $A$ which by definition is grid-size dependent. Bonetti et al (2020) and related modeling papers showed that using the specific catchment area $a$ is the correct use of variable for water flux at a point rather than the total catchment area, $A$, and the resulting numerical results using $a$ are not grid dependent.

Line 515: If I have understood correctly, the claim of not having any numerical error in the presented results relies on the observation of symmetry for "three rivers" cases with LSS compared to solutions without the filter. I am not totally convinced by this argument. In my understanding, the authors have taken three water influx boundaries, with one in the middle and hence they say that the solution should be symmetric. However, this would be the case only when solution geometry matches the domain geometry, and if that is not the case, the obtained solution would try to adjust on the discrete grid and it will have some defects (symmetry breaking). This is also observed in other pattern-forming systems (see Debenedetti, Nature (2001)). How I see this is that the numerical solutions without filter bear some memory of initial conditions, which get removed by using the filter. I am curious to see if the average first-order valley spacing changes a lot for with/without filter cases for the long rectangular domain case. Or, is there a big change in the statistics of the landscapes (hypsometry, power spectra, drainage network statistics) for with/without filter cases? Based on the color labels of the solution in figures with and without filters, the range of water flux and elevation height seems to be in the same range for both scenarios. For this reason, does the author know what happens when you add additional noise to their governing equations, even when there is no filtering? Do we see a symmetrical case appearing in those simulations as well?

**Minor Comments:**

Figure 1 can be made much more intuitive than its current form.

Figures 3 and 4 could be combined.

The label in Figure 6 is missing.

Line 155: Spelling mistakes "though of".

Lines 160: Awkward sentence: "this choice of source, $k_m$ has the unit m.s-1 of a speed."

Figure 19: Missing the literature source "approximately 35.9°N, 120.8°W extracted from Dietrich."

---

## Author Response (AR1)

**Answers to the review of the article: Large structures simulation for landscape evolution models**

Julien Coatléven [*][†], Benoit Chauveau [*]

**To all reviewers**

We first thank the reviewers for their careful reading of the manuscript, their relevant comments and valuable suggestions. We might have misunderstood the way discussion on esurf works, and probably have answer too early to the first reviewer. Should this be the case, we apologize about that.

We understand from both reviews that publication under the present form is very unlikely, as well as that there are doubts on the fact that we can produce a good enough revised version. We have nevertheless tried to produce a revised version since it is our belief that the first version was quite misunderstood. In the revised version we have prepared, most parts have been completely rewritten, taking into account the remarks of the reviewers (notice that this made the latexdiff produced file requested by the journal quite useless). We felt that there were sufficient similarities between the reviewers remarks, therefore before giving separate answers to each of them we start by giving some precision on the two main topics of the first version of the paper, in the hope that it will clarify things. We have rewritten the major part of the article including the introduction, basically keeping only the model description and the numerical results. The first major modification of the new version consists in the fact that we have decided to postpone the publication of the mathematical details on the consistency correction for MFD and its extension to node-to-node methods to a future paper in a more suitable journal. Thus the large majority of the mathematical technicalities have been suppressed, allowing to focus on the treatment of numerical instabilities. Since it is clear that our example involving symmetry was not convincing enough, the second major modification is that we have added both a short theoretical analysis of the stability issues as well as numerical illustration on analytic test cases on which it should be obvious that the simulation does not produce the correct solution as well as the fact that the erroneous solution without considering a filtering strategy is more appealing to the eye than the correct one, leading to the treacherous situation we were trying to describe in the first version of the paper.

**The MFD algorithms and the GMS model**

We believe that because of the mathematical formalism, there has been a fundamental misunderstanding concerning the MFD algorithms. In [2], it was rigorously established that at the very least on cartesian meshes the cell-to-cell MFD is exactly a mesh-dependent mean of the water flux associated with the discrete Gauckler Manning-Strickler (GMS) model obtained through the two-point flux finite volume approximation. We perfectly understand that the MFD algorithms were not designed with the GMS model in mind, and that they are believed to be a very different model. They nevertheless unexpectedly truly happen to be a discretization of the GMS model, even if they were not intended to. Using this identification, in [2] it was proposed to replace the mesh-dependent mean of the water flux with a correct estimate of the discrete water flux, leading to a consistent, convergent and as mesh-independent as possible approximation of the continuous water flux. The GMS-MFD approach can then be extended to general polyhedral meshes, as detailed in [2]. A numerical convergence study
* * *
[*]IFP Énergies nouvelles, 1 et 4 avenue de Bois-Préau, 92852 Rueil-Malmaison, France
[†]julien.coatleven@ifpen.fr

was conducted there, in particular it illustrates the problematic behavior of the outcome of MFD algorithms, with a very striking bad behavior on Voronoï meshes despite the fact that they fulfill the requirements of the two-point flux finite volume approximation. We originally wanted to recall this result and explain how this allows to correct along the same path most of the MFD algorithms of the literature, at least those for which no additional artificial mesh dependency is introduced, still using the GMS discrete model as a reference. This excludes for instance single flow direction methods, as within the context of the GMS we can see that they introduce a very strongly mesh dependent coefficient. This reinterpretation also allows to directly link the MFD algorithm with the specific catchment area $a$ of [4, 1], since the GMS model is a generalization of the model defining $a$. The work of [2] was mostly dedicated to correcting the non consistency of MFD algorithms, rather than proposing a new and correct definition of the unit catchment area as in the works of Porporato and al, nevertheless leading to an abstract unification of both approaches through the general GMS model.

We were hoping that this would come as good news, as it provides a possible way to get rid of the mesh dependency of most classical MFD methods. The catchment area produced by the MFD algorithms, corrected using the formula of [3] to get a unit catchment area was historically used in our stratigraphic model. We observed the usual mesh dependency, which created much difficulties for obtaining reproducible results in particular in a parallel environment and completely prevented convergence with the mesh size. We have replaced it a long time ago by the discrete water flux coming from the GMS model, once we had understood the link between MFD and GMS.

This being said, the main objective of our paper was to discuss the problem of numerical instabilities arising from the self-amplification mechanisms of the coupling between water and sediments and how to control them using the LSS approach. Since it is necessary to first get rid of any $O(1)$ error, in the first version of the paper we felt that it was important to first discuss the consistency correction for the MFD as using a non consistent discretization for the water flow model would prevent from obtaining any meaningful result for the coupled water and sediment model. However it has apparently created much confusion and has diluted the main message that was supposed to be on the coupled system. This is the reason why we have decided in the proposed revision to simply replace this section on MFD by a short subsection explaining why we have chosen to use the GMS model to compute our water flow, simply emphasizing the link with MFD algorithms in what we hope will now be an understandable way.

**NUMERICAL INSTABILITIES**

We mentioned in the introduction of our first version that in the absence of analytic solution, it is hard to decipher if the obtained results are not the consequence of numerical noise. Of course we did not wanted to say that it is impossible to construct analytic solutions (one can always do that by prescribing a function that satisfies the boundary conditions, inject it in the system and compute the corresponding source terms) but rather than when performing real life simulations on complex topographies, we precisely do not know what is the correct solution and thus if we cannot certify that the numerical scheme is reliable, it is hard to be sure that the obtained result is trustworthy. This is the reason why we had chosen in our first version to illustrate the numerical stability issues on our "three rivers" tests case, as despite the fact we do not know the exact solution thanks to the symmetry preservation properties of the equations the continuous exact solution will be symmetric. Thus, a good numerical approximation must be symmetric up to the expected discretization error, which should tend to zero with the mesh size and should be relatively small (say $O(\Delta_{xy})$) for a fine enough mesh. This is not what we have observed, the error is of the same order of magnitude than the solution, as the main water flow is not located where it should. Obviously, this was not sufficiently convincing but we still believe that this example is important as it is an easy to perform test. This the reason why in the proposed revision, we have derived analytic solutions on which we study the numerical behavior of the system. We have added a short theoretical analysis of the evolution of perturbation's energy, from which the ratio $\tau$ between the product of the diffusive water-driven coefficients and the water flow and the diffusive gravity coefficients naturally appears. The ratio $\tau$

seems to be a key control parameter of this equation system. As expected, when the parameter $\tau$ is relatively small, we obtain a good approximation and the overall method is converging with the mesh size (showing by the way that our numerical scheme is convergent and that our implementation is correct). However when $\tau$ is large, not only do we loose convergence but the numerical solution is visually very different from the analytic one. We have chosen a quite artificial looking solution, a stationary monodimensional solution to which we add analytic stationary smooth bumps. For large $\tau$, we obtain a non stationary numerical solution with complex topographic structures that could be reminiscent of valleys or rivers and have a "landscape" look. However this is totally artificial and the result of self-amplified numerical noise. Using filters, those perturbations disappear and we recover the correct solution even for large values of $\tau$. In addition to the three rivers test case, we hope that those new examples will be convincing enough.

**Reviewer 1**

We thank the reviewer for his time and valuable comments.

. In the revised version, we have chosen to completely remove the technical discussion on the MFD algorithms and to focus on illustrating why numerical instabilities are an issue and how to control them. We are aware that even the revised version might in some respects be better suited to a community of pure modelers. However, the modeling approach based on MDF algorithms being one of the corner stones of the community, it is our belief that this type of publication, which provides these kinds of recommendations on the deployment of LEMs, should be visible by this community.

. As mentioned above, in the revised version of the manuscript we have added a comparison between numerical and analytic solutions, that should emphasize more clearly that the self amplification of numerical errors is a true issue.

. It is well documented in the mathematical literature that the stationary transport equations like the continuous Gauckler-Manning-Strickler (GMS) model considered here are well posed if a sufficient condition on the topography is satisfied, the two we proposed being probably the simplest ones. Otherwise the problem can have several solutions, infinite solutions or no solution at all. As the MFD algorithms are truly a discretization of this equation, even if they were not originally designed to be one, they suffer from the same deficiencies. The additional "and > 0" was a typo, our apologizes for that. By "drainage basin" we wanted to designate basins with no accumulation or flat areas which would then satisfy one of the sufficient conditions (the conditions on the Laplacian being the simplest one), and we are probably wrong on the use of the terminology "drainage" so we have simply chosen to avoid the use of "drainage" in the revised version. Valleys are not necessarily a problem provided they possess a downstream spill point. Since from a practical point of view on cartesian grid the MFD algorithm computes a value for the water flow even in the problematic areas, this creates a model error that if overlooked can also lead to non-physical flow patterns because of the self-amplification mechanisms of the coupled sediment/water model. This is why in the numerical experiments of the paper we have been careful to always use topographies that satisfy one of the sufficient conditions. In particular, this is the reason why we have avoided using random noise in our initial conditions. We hope that the much lighter discussion of the GMS model will make things clearer.

**Reviewer 2**

We thank the reviewer for his time and valuable comments.

. We have used the word chaos in the usual mathematical sense (see for instance Differential Equation Dynamical Systems & an introduction to Chaos, M. Hirsch, S. Smale and R. Devaney) which may be

quite different from the notion you have in mind. To avoid any misunderstanding, we completely removed any use of the word chaos in the revised version. Roughly speaking, what we wanted to say is that even very small perturbations in the data can result in large differences in the solution, including perturbation arising from numerical errors. Anyway, we are in fact quite worried by our own findings: we do believe that there is a risk that some numerical results are indeed blurred by numerical errors. Fortunately it depends on many factors like the use of high order methods, explicit schemes, etc... Of course our intention was not to criticize former works but rather to raise an alarm on the risks inherent to the coupling between water flow and sediment erosion and transport. We have deeply modified our introduction and abstract in this sense.

. The coefficient $r_s$ in our model represents the exponent of the non linearity linking the water flow and sediment model. We do not see any direct link with the concavity index.

. One of the consequences of turbulence is that small physical perturbations will be amplified and have a major impact on the final solution. The channel instability studied by Porporato and al indeed corresponds to a positive feedback loop between water and sediment. This is however not unrelated with numerical instabilities. If perturbations of the continuous solution are unstable and amplify with time, the same will happen for perturbations arising from numerical approximations. Continuous level "turbulent" behavior is always linked to unstable numerical behavior. For the coupling between sediment and flow the same phenomenon will occur: a small error on the sediment distribution, for instance a small lack of sediment somewhere will result in a modification of the water flow, slightly focusing the flow where the numerical error had creating a small noise. This will result in more erosion at this point, increasing the originally small perturbation in the sediment. The physical instability has necessarily a numerical counterpart. This is the reason why in computational fluid dynamics at high Reynolds number high order methods are used to reduce as much as possible the numerical errors and some numerical dissipation mechanism is always added to simulations (for instance sub grid scale diffusion). We wanted to emphasize that the similarities between CFD and LEMs will necessarily result in the need of similar numerical treatments.

. We might have missed some part of the literature. In the works of Porporato and al. as that we were citing, as well as the older works of Smith and al on the model we use, the stability of solutions of the continuous model is evaluated through a linear stability analysis under periodic perturbations. This analysis leads to monodimensional EDO that they solve to obtain the value of the parameter controlling the time evolution of the perturbation, hence classifying stable and unstable regime. We were absolutely not saying that those results do not represent the physical process with accuracy. What we wanted to express with this sentence is, as explained above, when performing a numerical situation in a realistic, complex setting, we do not have any tool to discriminate between correct and incorrect solutions, precisely because we do not know the exact solution. Of course, this is inherent in any interesting numerical simulation. In a "turbulent" context, if no special treatment is applied it is very likely that numerical error amplification can occur. This is not always the case, but if it is not taken into account by design in the numerical method, the risk exists. In the revised version of the paper, as explained above we compare our results to carefully designed analytic solutions, which consists in a stationary monodimensional solution similar to the ones usually used in the literature, but perturbed by analytic smooth bumps. We show that for the "non turbulent" regime everything is ok and the simulation converges to the exact solution. However, in the "turbulent" context, numerical instabilities are amplified up to the point that the numerical solution is very different from the correct one, and any case up to a point where it cannot be considered a reasonable approximation. However, the resulting error has a "landscape" look that is treacherous. In this well controlled setting, it is easy to see that this good looking solution is wrong. The intended meaning the sentence you cited was precisely to warn that it becomes very difficult on a real test case to identify this kind of errors.

. We have decided to remove almost all the section on the MFD for the reasons explained above, and we

simply motivate our use of the GSM model.

. We have added appropriate citations in the introduction. We also postpone to the discussion section the question of the generalization to other LEMs. We have tried to better motivate why we believe that our findings will carry to those models.

. A complete answer requires to enter into mathematical details, and we hope that the above section on the MFD will have clarified things. We nevertheless repeat that although they were not designed for this purpose, most of the MFD algorithms in fact exactly coincide with well chosen discretization of the GMS model, and can thus be analyzed within the GSM framework. This allows to understand easily why they have such a strong mesh dependency and how to correct it. Notice that the GSM model we speak of directs flow with the topographic and not the hydraulic gradient. We discuss its generalization at the end of the paper, as a potential way to overcome its limitations.

. Unless we have misunderstood their paper, which is of course possible, we understood that they derived and studied the evolution of the mean value in one direction of the model. This is a limit case of filtering: where the mean is taken over a large domain rather than a small one. This might be a strong simplification, but this can be seen as a LES model. Our intention was to avoid pretending that we were the first one to use a LES approach, as their work approached very closely the idea.

. We have rewritten this section entirely, and we hope that you will find that it does now justice to those papers. Again, our intention was to justify the use of the GSM model and this associated water flux $q_w$, which because of its many parameters is a generalization of the model underlying $a$.

. Provided the source terms, initial value and boundary conditions as well as the domain satisfy some symmetry, then this symmetry should be kept at all time. For instance, if the solution is symmetric with respect to the axis $x = 0$, then it is not difficult to see that $(h_s(-x,y), h_w(-x,y))$ solves the very same equation than $(h_s(x,y), h_w(x,y))$ with the same data, and thus that the two are equal (assuming uniqueness of solutions of course). If the mesh is symmetric, this property should carry easily to the numerical solution. However, even if the mesh is not symmetric, if the numerical solution is a reasonable approximation of the continuous one then symmetry should be obtained up to the numerical error, which should behave like $O(\Delta_{xy})$. This is incompatible with a water flow being at the wrong position: this is an $O(1)$ error. Furthermore, on a fixed grid, the position of the flow should never depend on the linear solver chosen to compute it. We hope that the added comparison with analytic solutions in the revised version will be more convincing, and that this will give more strength to this symmetry test, that we believe to be important. Indeed, symmetry testing can be done in situations where we do not know the exact solution. Concerning valley spacing, notice that we do not say that the system is incapable of producing patterns: we just say that if there is no significant instability seed in the physical data, no patterns should spontaneously emerge. We do not understand the end of your remark: if we add significant noise, there is no reason for the solution to remain symmetric as the data will no longer be symmetric.

. We have taken care of the minor comments.

**References**

[1] S. Bonetti, A. D. Bragg, and A. Porporato. On the theory of drainage area for regular and non-regular points. *Proc. R. Soc. A 474: 20170693*, 2018.

[2] J. Coatléven. Some multiple flow direction algorithms for overland flow on general meshes. *ESAIM: Mathematical Modelling and Numerical Analysis, Vol. 54 (6), pp. 1917-1949*, 2020.

[3] P. J. J. Desmet and G. Govers. Comparison of routing algorithms for digital elevation models and their implication for predicting ephemeral gullies. *Int. J. Geo. Inf. Syst., Vol. 10(3), pp. 311-331*, 1996.

[4] J. C. Gallant and M. F. Hutchinson. A differential equation for specific catchment area. *Water resources research, Vol. 47, W05535*, 2011.

---

## Referee Report (RR1)

**Review of "Large structures simulation for landscape evolution models" submitted to ESurf**

Summary

The authors demonstrate and provide a means to address the amplification of numerical errors in a basic landscape evolution model. They do so by drawing comparison between this model and the Navier-Stokes model, both of which have mathematical characteristics that support the amplification of small perturbations. Their solution involves filtering the water and sediment fluxes, effectively smoothing them to a scale greater than the grid cell width. They show that real perturbations above the scale of the filter can and do amplify, while results suggest those below the scale do not. They demonstrate that re-introducing heterogeneity in physical parameters can create the satisfyingly complex simulation results we expect from landscape evolution models, but acknowledge that in practice it can be difficult to discern the complexity that arises from numerical artefacts versus introduced heterogeneity.

Overall, I found this to be a compelling work. By the end of the paper, I was convinced that filtering such as they have done could help achieve reproducibility in landscape evolution models. As previous reviewers and the editor have noted though, the paper is challenging to read. While some of this is due to the use of mathematical formalism that is unfamiliar to the geomorphology community, I think the authors could still improve clarity of the formalism by working on the text in Section 2. Notation that is not defined should be defined, and a greater attempt should be made to distinguish between similar quantities, or at least remind the reader of their definitions along the way. Some notes on this are listed below.

A previous reviewer argued that there was limited value to the work because it simply introduced another length scale. I think they have clearly argued that some length scale is introduced whenever a numerical solution is implemented, and it would be best the solution were dependent on this scale rather than the grid cell size.

A note on the response to reviewers – the section numbers you give seem to be off by one, so section 1.1 is in fact 2.1, and 3.3 is 4.3.

Comments by line number

19. Define MFD
35. "from of a"
43. gravity-driven transport
44-40. This sentence needs restructuring
53. I do not know what the parenthetical means, and I suspect many other readers will not as well.
58. target length scale $\alpha$
96. "the principle of the conservation of mass" -> the principle of mass conservation
105. "b is a data" Rephrase.

108-109. Can you be clearer that Ss is in-situ production and Bs is boundary input?

111. "Let us precise that in the following the xy" Rephrase.

119-120. This sentence is unclear to me. Can you rephrase or describe in more detail?

137-138. This presumes we know what "most classical cell-to-cell MFD algorithms" you are talking about, and I am unsure.

144. thought

145 truly

146. For which choice of source? Are we talking about the case in the previous sentence, which you are not considering, or something else?

147. Definition of a. Is the $h_w$ in parenthesis indicating $\eta_w$ is a function of $h_w$?

173. You have just said that the specific catchment area (SCA) is an approximation of qw, but now you say that the specific catchment area a is in fact equal to qw? I am confused. Then later you are treating a as interchangeable with $h_w$? And qw with Qw?

209-225. This whole discussion of flat areas and the bowl is a little obtuse, and seems to me outside the main point of this paper. Can you just say that you will focus on well-posed problems, which are not completely flat or closed depressions, and move much of this to the supplement? (Edit after reading the response to previous reviewer: I am sorry to be asking you to downplay something added to address a past reviewer comment. Do as you please, but I do think that this section is a barrier for readers)

245. Where u is a stand-in variable?

247-255. This is really hard to follow with the stand-in variables, and the coefficients with plusses and minuses, much of which goes unexplained. I am not even sure why we need to see equations 12 and 13.

280-281. Rephrase for clarity.

324-328. This is a helpful description of the implications of this analysis. However, I do find myself wondering, if the continuous scheme has these self-amplifications, and the numerical scheme results in perturbations that are similar enough to those found in nature (which is full of heterogeneity and general messiness) then should we be so worried about the numerical errors that amplify? Or is there some difference between the perturbations derived from the numerical scheme? Perhaps I am missing something here, or this is more of a philosophical point, but you might add something about this.

357. Should this include Fig. 8 too?

Figures 4-8. It would be helpful to title the subplots with the parameter combination, rather than just listing them in the caption.

Figures 9. Again, it would be helpful to title the subplots with the parameter combination. In the caption, you can then reiterate which previous figure (4-8) shows the corresponding analytical solution.

371. "Which is no more high enough" It might be better to say "Problems are potentially more severe in finer meshes because numerical diffusion that can dissipate residual numerical errors declines with grid spacing" or something like this.

387-388. This also makes me think of how steady-state solutions to the imperfect numerical model do generate steady-state topography that appear to satisfy the governing equation. For example, the relationship between incision height and curvature (Figure 7) in (Theodoratos et al., 2018), or slope and area in many other studies. There is sometimes some scatter around the expected relationship though. So does your work suggest this scatter is the result of the error you describe, or are you saying we are even using the wrong measuring stick of success?

397. What would it mean for the cartesian mesh to not be symmetric?

419-421. I think I understand this from the perspective of reproduceable LEM simulations, but as you have shown, aren't these self-amplifications likely reflecting a natural phenomenon? I am not familiar with the CFD world to know how they accept or handle such features, but I suspect geomorphologists using these LEMs will wonder about this.

429. "refer the reader to a the quite recent review Zhiyin (2015)" Remove "the quite". Here and elsewhere your citation style should be checked too. I would expect (Zhiyin, 2015).

433. Remind us, or define \delta and d

457-459. Rephrase for clarity.

463. Are there existing formulations for steady state shallow water equations that have already used such filtering, or is this also new?

Figure 14. Again, it would be helpful to have title labels with the parameter information.

Figure 15. Is the x-axis here correct? Everywhere else when you have presented a "convergence curve" the x axis has been grid spacing.

495. This makes sense, as \tau is similar to *Pe* in Perron et al. (2008). The length scale derived from Pe=1 describes the scale at which advection (destabilizing) begins to exceed diffusion (stabilizing). Of course, this is just an analogy, the underlying model is different, as you discuss.

522. "implicitly"

539. "of the same magnitude *as* the analytic ones"

544-547. This seems like an essential point, and possibly an answer to my above comments on the realism of self-amplification. It might be worth highlighting that connection.

548. "loose" -> "lose"

575-576. "Undoubtedly the correct solution" Rather than saying this, can you support with other evidence? For instance, what does Perron et al. (2008) suggest should be the length scale spacing between ridges and valleys, and how does that relate to the filter parameter?

604. "similar numerical stability issues that" -> "similar numerical stability issues to"

659. "physcally"

678. I would remove the double negative.

680. Repartition seems like the wrong word here.

684. "Llet"

Works Cited

Perron, J. T., Dietrich, W. E., & Kirchner, J. W. (2008). Controls on the spacing of first-order valleys. *Journal of Geophysical Research: Earth Surface*, *113*(4), 1–21. https://doi.org/10.1029/2007JF000977

Theodoratos, N., Seybold, H., & Kirchner, J. W. (2018). Scaling and similarity of a stream-power incision and linear diffusion landscape evolution model. *Earth Surface Dynamics*, *6*(3), 779–808. https://doi.org/10.5194/esurf-6-779-2018

---

## Author Response (AR2)

**Answers to the review of the article: Large structures simulation for landscape evolution models**

Julien Coatléven [*†], Benoit Chauveau [*]

**To all reviewers**

We first thank the reviewers for their careful reading of the manuscript, their relevant comments and valuable suggestions. We now give here our separate answers to their new remarks.

**1 Reviewer 1**

. Thank you for your comments. To ease the reading, we have tried to reduce the amount of mathematical details, essentially by moving to the appendix some of them also by removing some unnecessary ones inside the appendix sections.

**2 Reviewer 2**

. The main objective of our paper was to explain how to obtain correct numerical results for LEMs, that are not dominated by amplified numerical errors. We might not solve the more general issues of LEMs that the reviewer has in mind, however we strongly believe that it is necessary to start by ensuring that numerical results are reliable for simple models before considering more advanced issues. In particular, we consider and illustrate in this paper that one of the major problems regularly encountered in LEMs is the anomalous mesh dependency and its implication in terms of non-physical results. In this context, we are confident on the added value of our approach that follows well-known principles used by the computational fluid dynamics (CFD) community, since unfortunately the issues of LEMs are very close to those of CFD for turbulent flows. As you mentioned, on observed landscapes channelization can be modeled as occurring "randomly", since very small scales heterogeneity can have a huge impact on the flow. Reproducing this phenomenon would require a detailed knowledge of very small scale details of the landscape (such as boulders, vegetation, etc..). Numerical models that do not incorporate such data or any randomly generated data should not artificially become random out of numerical error amplification. Of course, the approach presented in the paper is only a first step in this direction, this we only filter small scales without adding any "small scale model". This is what we propose as a perspective in our conclusion: to complement the model with sub grid scale modeling.

. The mathematical requirements you insist on (Eq. 8) are introduced as sufficient conditions, not necessary ones. The mathematical references we gave provide well-posedness results under such conditions that essentially consist in requiring the positivity of the zero order operator. Since it is was obviously not clear enough, we have detailed in the new version of the paper (section 1.1) situations for which the system is still likely to be well-posed despite not fulfilling one of those requirements (in particular saddle-points and valleys) and thus not having a comprehensive mathematical theory. We have also explained why if the
* * *
[*]IFP Énergies nouvelles, 1 et 4 avenue de Bois-Préau, 92852 Rueil-Malmaison, France
[†]julien.coatleven@ifpen.fr

topography has some flat or accumulation areas then the system will not be well-posed, in particular by providing an analytic example for the less obvious case of accumulation areas.

. We are quite surprised by the way you mentioned that we replace the grid size dependency by a smoothing parameter dependency. What we have tried to explain is that smoothing allows to get rid of the anomalous grid dependency due to numerical error self-amplification, and that it allows to recover the normal mesh dependency which is the convergence of the solution to the correct result when making the grid and filter size go to zero. This is the best kind of "mesh independency" achievable, since as long as we are discretizing on a mesh, this mesh will have an impact on the results. In particular, in any simulation the mesh will always control the size of the smallest details accurately reproduced in the results (that will ultimately be several cell-size large). When LEMs are designed without any filtering strategy, it corresponds to consider that the cut-off length scale is the mesh size. In other words, the scale you called as an "additional artificial length scale" is already implicitly considered, but it lacks calibration and leads to the recurring problems in LEMs. In our case, since we need a filter to ensure correctness of the results, the resolution of the model will indeed be controlled by the filter size. Here we show that the calibration of this length must simply respect an elementary principle: to be largely lower that the size of the geomorphic structures the LEM aim to reproduce. Based on this calibration, the mesh resolution must be chosen so that the filter is correctly discretized. Thus we have not deteriorated the situation: we still have a unique discretization parameter that governs the resolution of the model. We have added a paragraph at the end of section 3.3 to emphasize this point.

. Asserting that the Gauckler-Manning-Strickler equation (which is the continuous equivalent of corrected MFD algorithms, and thus roughly speaking corresponds using one of the MFD algorithms) is not able to simulate the water flow in valleys would challenge many previous studies. Fortunately, thanks to what has been said on the difference between sufficient and necessary conditions, we hope that it will be clearer that the limitation pointed out by the reviewer is no more relevant.

**3    Editor**

. We hope that we have given satisfactory answers to the remarks of reviewer 2 (see the above section and the new version of the paper).

. We appreciate that despite the abundance of mathematical formalism, you consider that the subject of this manuscript rightfully belongs to the scope of ESURF and have the potential to be of interest for the LEM community. We have tried to reduce the amount of mathematical details, essentially by moving to the appendix the computational steps of section 1.3 devoted to perturbation study and also by removing some unnecessary details inside the appendix sections.

---

## Author Response (AR3)

**Answers to the review of the article: Large structures simulation for landscape evolution models**

Julien Coatléven *†, Benoit Chauveau *

**1   Reviewer 1**

We first thank the reviewer for his very careful reading of the manuscript, his relevant comments and valuable suggestions. We have taken into account all the remarks concerning typos, notations, or rephrasing. We refer to the modified version of the article for those points. We now give more complete answers to your more involved comments.

. *"137-138. This presumes we know what "most classical cell-to-cell MFD algorithms"* you are talking about, and I am unsure." : we have added the corresponding references.

. *"173. You have just said that the specific catchment area (SCA) is an approximation of $q_w$, but now you say that the specific catchment area $a$ is in fact equal to $q_w$? I am confused. Then later you are treating $a$ as interchangeable with $h_w$? And $q_w$ with $Q_w$ ?"*: Our previous notations were indeed quite confusing about discrete and continuous versions of the water discharge and various catchment areas. We believe that this was the origin of the small misunderstanding. In the new version, we use different notations for continuous quantities and their discrete counterparts. It should not be clear that the specific catchment area computed from MFD algorithms was a discrete amount, and thus an approximation of a continuous one. On the other hand, $a$ and $q_w$ are defined at the continuous level. The quantity $a$ can be formally identified with $h_w \eta_w(h_w)$, just by looking at the expressions: this is how the GMS model can be viewed as a generalization of the model underlying $a$. Finally, we used $\mathcal{Q}_w$ as a generic "local discharge of water" to encompass all the possible choices of the literature, without choosing a specific one. At the end, we set $\mathcal{Q}_w = q_w$ to emphasize that we use for the remaining of the paper the consistent $q_w$. We have tried to make it clearer in the new version.

. *"209-225. This whole discussion of flat areas and the bowl is a little obtuse, and seems to me outside the main point of this paper. Can you just say that you will focus on well-posed problems, which are not completely flat or closed depressions, and move much of this to the supplement? (Edit after reading the response to previous reviewer: I am sorry to be asking you to downplay something added to address a past reviewer comment. Do as you please, but I do think that this section is a barrier for readers)"*: we have postponed it to the beginning of the section explaining how to overcome the limitation of the GMS model, i.e. the very last section of the paper. We hope that proceeding this way will be enough to ease the reading.

. *"247-255. This is really hard to follow with the stand-in variables, and the coefficients with plusses and minuses, much of which goes unexplained. I am not even sure why we need to see equations 12 and 13"*: we have removed the stand-in variables and eliminated the technical requirements to ease the reading.
* * *
*IFP Énergies nouvelles, 1 et 4 avenue de Bois-Préau, 92852 Rueil-Malmaison, France
†julien.coatleven@ifpen.fr

. *"324-328. This is a helpful description of the implications of this analysis. However, I do find myself wondering, if the continuous scheme has these self-amplifications, and the numerical scheme results in perturbations that are similar enough to those found in nature (which is full of heterogeneity and general messiness) then should we be so worried about the numerical errors that amplify ? Or is there some difference between the perturbations derived from the numerical scheme? Perhaps I am missing something here, or this is more of a philosophical point, but you might add something about this"* and *"544-547. This seems like an essential point, and possibly an answer to my above comments on the realism of self-amplification. It might be worth highlighting that connection."* : We thank you for this comment, and are happy that the perturbation analysis was helpful. Since the numerical noise grows from the physically based self-amplification mechanisms, it finally has a "realistic" good look. However, the numerical noise arises for instance from linear and non-linear solvers, and is influenced by software or hardware choices, which has obviously no relation with nature. Moreover, numerical noise is generated at each time step, and thus accumulates over time. Its statistical signature will never coincide with the physical observations: it would be like throwing the parts of a puzzle and hoping that they will correctly reconstruct the puzzle when going down. This is in our opinion a methodological key point: a numerical simulation is not supposed to reproduce directly natural observations, but only to compute a correct approximation of a model for a given dataset. This is the model and its data that should reproduce nature, and in this case if solved correctly the numerical solution will be a useful approximation. If we do not willingly add randomness in the model or its data, the numerics should not introduce it out of nowhere and bypass our modeling. It is not reasonable to rely on numerical hazard to recover the missing elements in a model. Worst of all, numerical noise lacks two essential modeling requirements: reproducibility and explainability. The first one since numerical noise depends on the softwares/algorithms used, the number of processors, etc. The second one since it is almost impossible to track how the numerical errors are generated. We have added a paragraph in the discussion section on this point.

. *"387-388. This also makes me think of how steady-state solutions to the imperfect numerical model do generate steady-state topography that appear to satisfy the governing equation. For example, the relationship between incision height and curvature (Figure 7) in (Theodoratos et al., 2018), or slope and area in many other studies. There is sometimes some scatter around the expected relationship though. So does your work suggest this scatter is the result of the error you describe, or are you saying we are even using the wrong measuring stick of success ?"*: It is indeed possible that the amplified numerical noise is reflected in this scatter. However the only way to be sure of it would be to reproduce their experiments with a filtered model.

. *"397. What would it mean for the cartesian mesh to not be symmetric ?"*: we intended this to emphasize the fact there is no symmetry problems coming from the mesh. We have simply removed this confusing comment in the new version.

. *"419-421. I think I understand this from the perspective of reproduceable LEM simulations, but as you have shown, aren't these self-amplifications likely reflecting a natural phenomenon ? I am not familiar with the CFD world to know how they accept or handle such features, but I suspect geomorphologists using these LEMs will wonder about this"*: For CFD, numerical methods are usually tested on analytic solutions and well-established benchmark solutions (such as the Taylor-Green vortex). Symmetry is systematically tested, and numerical methods that do not control efficiently numerical noise are simply discarded. Self-amplification of numerical noise is indeed avoided at all costs, since noisy numerical results are quantitatively useless.

. "429. "refer the reader to a the quite recent review Zhiyin (2015)" Remove "the quite". Here and elsewhere your citation style should be checked too. I would expect (Zhiyin, 2015).": we have removed the quite. As for the citation style, we use the style file provided by the journal, and consequently have no control on the chosen style.

. *"463. Are there existing formulations for steady state shallow water equations that have already used such filtering, or is this also new?"*: When shallow water alone is considered, the topography is in general assumed to be a data without noise. Thus, there is no reason to filter the topography. Notice that the full shallow water system because of the non-linear terms poses some specific but well-known numerical challenges.

. *"Figure 15. Is the x-axis here correct? Everywhere else when you have presented a "convergence curve" the x axis has been grid spacing"*: there was indeed a mistake in the x-axis, it was supposed to be $\ln \Delta_{xy}$ as everywhere else.

. *"495. This makes sense, as $\tau$ is similar to Pe in Perron et al. (2008). The length scale derived from Pe=1 describes the scale at which advection (destabilizing) begins to exceed diffusion (stabilizing). Of course, this is just an analogy, the underlying model is different, as you discuss."* : Indeed, $\tau$ and $Pe$ seems to play similar roles, and we believe that this is how our results could transpose to the model of Perron et al.

. *"575-576. "Undoubtedly the correct solution" Rather than saying this, can you support with other evidence ? For instance, what does Perron et al. (2008) suggest should be the length scale spacing between ridges and valleys, and how does that relate to the filter parameter ?"*: In fact we had conducted a convergence study on this case, drastically reducing both the filter parameter and the mesh size. The solution remains unchanged for the refined situations, which is a strong argument that the uniform solution is the correct one for the large value of $k_g$. We have added this comment in the new version.